# Deciphering splicing heterogeneity at single-cell resolution by SCSES

Xiao Wen [1,2,5], Xuan Lv [1,2,3,5], Dan Guo[1,2], Nan Han[1,2,3], Lei Zhou[1,2,3], Peizhuo Wang [4] & Zhaoqi Liu [1,2] ✉

Alternative splicing (AS) plays a critical role in generating cellular transcriptomic heterogeneity. While single-cell RNA sequencing (scRNA-seq) has become a standard approach for exploring this heterogeneity, it remains challenging to accurately characterize splicing changes at the single-cell level due to high dropout rates, inevitable noise, and limited coverage. To address this, we developed SCSES (Single-Cell Splicing EStimation), a computational framework designed to enhance the AS profiles. SCSES infers and completes the missing splicing changes by sharing information across similar cells and events with data diffusion. Through systematic simulation studies, SCSES outperforms existing algorithms in recovering percent spliced-in (PSI) values and diversity across cell populations. When applied to various datasets, SCSES uncovers substantial splicing heterogeneity and cell subgroups with exclusive splicing patterns, which cannot be captured by conventional single-cell gene expression clustering. Together, our study provides SCSES as a valuable tool in deciphering splicing heterogeneity and is widely capable of handling different biological scenarios, species and sequencing platforms.

Single-cell RNA sequencing (scRNA-seq) techniques make it possible to decipher cell transcriptional heterogeneity on a large scale of cells[1]. While many studies have successfully characterized the gene expression specificity in different single-cell clusters[2–6], the complexity of post-transcriptional regulation has often been overlooked. Alternative splicing (AS), an essential post-transcriptional process during pre-mRNA maturation, substantially contributes to tremendous transcriptional diversity[7–9]. AS events can be classified into five classical types: exon-skipping events (SE), alternative 3' splicing site events (A3SS), alternative 5' splicing site events (A5SS), retention intron events (RI), and mutually exclusive exons events (MXE)[10] (Supplementary Fig. 1). AS generates various isoforms from the same pre-mRNA by jointing exons in different combinations[11], thereby directly determining proteins constitutions[12–15]. In this way, more than 95% of human multiexon genes undergo AS, yielding over 300,000 isoforms derived from ~24,000 protein-coding genes[16,17]. Thus, compared to overall gene expression, AS changes are not only more numerous but

also carry richer information in terms of transcript structure and functional diversity, which expands the dimension of feature space to define cell types and infer cell state trajectories[8,15,16].

A number of computational tools, including MISO[18], rMATS[19], MAJIQ[20], and IRFinder[21], are developed to identify and quantify AS events in bulk sequencing data. However, due to technical limitations of single-cell sequencing such as high dropout rate, high sequencing errors, and biased low coverage[9,22], these tools are poorly adaptive for scRNA-seq datasets[23,24], no matter whether using full-length protocols like smart-seq2[25] or the droplet-based protocols like inDrop-seq[26]. Several methods have been specifically designed to address this challenge. For example, Expedition introduces a computational pipeline for quantifying exon-skipping events by leveraging well-aligned reads and a model to detect differential splicing events[27]. Similarly, BRIE/BRIE2 infers differentially spliced exon-skipping events through Bayes regression models incorporating either event sequence or cell features[23,28]. Psix can identify AS events that are highly correlated with

[1]Department of Computation Biology, China National Center for Bioinformation, Beijing 100101, China. [2]Beijing Institute of Genomics, Chinese Academy of Sciences, Beijing 100101, China. [3]University of Chinese Academy of Sciences, Beijing 100049, China. [4]School of Life Science and Technology, Xidian University, 710071 Xi'an, Shaanxi, China. [5]These authors contributed equally: Xiao Wen, Xuan Lv. ✉e-mail: liuzq@big.ac.cn

cell state transition by a probabilistic model[29]. SCASL concentrates on the junctions with alternative 3'/5' splicing sites and infers the missing values in the AS probability matrix from other cells[30]. Nonetheless, there are still apparent deficiencies in each of the above methods. Firstly, some methods, such as BRIE2, heavily rely on cell type identities, which are unavailable before analysis. Secondly, methods including Psix use global gene expression to measure cell similarities, which may not fully reflect the complexity of heterogeneous splicing patterns among cells. Thirdly, the limited junction read counts often result in unreliable estimation of percent splice-in (PSI) values leading to inaccurate quantification in methods such as Expedition and SCASL (Supplementary Fig. 2). Lastly, most algorithms are limited in scope, primarily focusing on SE and MXE events, while lacking the ability to detect and quantify A3SS, A5SS and RI events. A detailed comparison of these methods is listed in Supplementary Data 1.

Here, we present Single Cell Splicing EStimation algorithm (SCSES), a network diffusion-based imputation method designed to accurately recover splicing changes across main types of splicing events at the single-cell level. SCSES is inspired by the data diffusion technique, which is widely applied in scRNA-seq data analysis, such as MAGIC[31], DTFLOW[32] and PHATE[33]. SCSES infers splicing changes by utilizing both cell and event similarities with different options of imputation strategies. Through extensive simulation studies and applications to various scRNA-seq datasets, we demonstrated the power of SCSES in faithfully recovering the splicing features in individual cells and cell groups. In summary, SCSES is a promising tool to explore and interpret single-cell data from a post-transcriptional perspective concerning the conventional single-cell gene expression analysis.

## Results

### SCSES characterizes splicing changes on single-cell resolution via data diffusion

SCSES is a computational framework designed to identify and quantify alternative splicing events using scRNA-seq data (Fig. 1). It takes scRNA-seq alignments as input and outputs refined PSI values for every detected splicing event in each cell. To define a global set of all splicing events, SCSES firstly merges all aligned reads from every single-cell into a pseudo bulk sequencing file without cell identities and identifies main types of splicing events by conventional algorithms (Fig. 1a). According to this splicing reference, SCSES then counts the raw inclusion/exclusion junction reads in each cell, constructing the raw junction read count matrix (Raw RC) and calculating the raw PSI matrix (Raw PSI) (Fig. 1a). Due to the high dropout rate and technical limitations, such matrices are very sparse with limited read counts, leading to inaccurate PSI estimation (Supplementary Fig. 3, 4). To overcome this sparsity, SCSES uses a diffusion operator that propagates information across similar cells/events over a lower dimensional manifold, restoring missing junctions in this process. The framework is built on the assumption that cells exhibiting similar activity of splicing machinery harbor akin splicing patterns[29] (Supplementary Fig. 5), and that events with similar regulatory features result in comparable splicing outcomes[23].

To build the underlying manifold, SCSES constructs cell and event similarity networks using K-nearest neighbor algorithm (KNN) (Fig. 1b). It uses gene expressions of RNA-binding proteins (RBPs), Raw RC, or Raw PSI matrices to measure pairwise cell similarities and builds the network with adaptively optimized $K$ values for each cell (Methods). Event similarities are defined by the RBP regulatory correlations and an embedding representation by integrating event sequence similarities, which includes the length, motif, conservation and k-mer features from the event sequence (Supplementary Note 1). Once these networks are established, SCSES aggregates the splicing information across highly similar cells or events to impute splicing junctions or PSIs, thereby correcting for dropout and other sources of noise (Fig. 1). We evaluate the degree of difference between two successive imputed matrices and terminate the process once the changes stabilize (Fig. 1c,

Methods). Based on our practice, we recommend three different data imputation strategies, matched to four types of biological scenarios, which are defined by the abundance of alternative junctions in both the target cell and its neighboring cells (Fig. 1c, Supplementary Fig. 6, Supplementary Note 7). Here, for a given AS event, cells with dropout in alternative splicing junctions are defined as WD (with dropout), while others as ND (non dropout). For ND cases, which have reads supporting alternative junctions, we suggest imputing the PSI value directly using a cell similarity network (Supplementary Fig. 7, 8). WD cases are further divided into BD (biological dropout) or TD (technical dropout) based on their read count patterns. In BD cases, both the target cell and its neighbors show abundant reads for one junction type (either inclusion or exclusion) but complete absence of reads for the other type, indicating that only a single isoform is expressed. TD cases, however, are characterized by low read depth in both harboring gene and splicing event in the target cell, likely due to technical limitations rather than biological truth. We further divided TD based on the information available from neighboring cells: TD+Info represents cases where neighboring cells have abundant junction reads that can guide imputation, while TD-Info represents cases where such information is lacking in the local neighborhood. For BD, TD+Info and TD-Info, we recommend data imputation on the raw junction matrix with cell similarity network. And specifically, TD-Info requires an additional round of data diffusion using the event similarity network on the PSI matrix derived from previous round of imputation (Methods, Supplementary Fig. 7-8). To determine the appropriate scenario for each cell-event pair belongs to, SCSES pre-trains a cascade decision model to predict the scenario probabilities (Methods). Lastly, SCSES estimates the final PSI value via a weighted linear combination of predictions from all different strategies, with weights determined by their corresponding probabilities (Methods, Fig. 1c).

### SCSES recapitulates splicing characteristics in individual cells

To assess the accuracy of SCSES, we utilized bulk RNA-seq data from four cell lines (HCT116, HCC1954, HepG2, and HL-60) obtained from Cancer Cell Line Encyclopedia (CCLE)[34]. From these bulk datasets, we selected a set of high-confidence splicing events with PSI values served as the biological truth. We then used real scRNA-seq data for the same cell lines[35,36] to test if SCSES could accurately recapitulate the splicing landscape at the single-cell level (Supplementary Fig. 9a, Supplementary Fig. 10). We compared SCSES against five existing algorithms for PSI estimation in scRNA-seq data: BRIE1, BRIE2 (aggregated model), Expedition, Psix and SCASL, five algorithms designed for inferring PSI values in scRNA-seq data, as well as rMATS, a classical algorithm designed for bulk RNA-seq data. Due to the substantial variability in splicing events identified by different algorithms, we performed pairwise comparisons between SCSES and each of the other tools, focusing only on the overlapping splicing events to ensure a fair evaluation (Supplementary Data 2). Firstly, we evaluated the accuracy of PSI recovered by different methods. We calculated the Spearman correlation coefficients (SCC) between the PSI values estimated by the compared algorithms and the benchmark PSI values (derived from bulk RNA-seq of the matched cell type) across all events within each cell, as well as the root mean squared error (RMSE) between estimated and the benchmark PSI values across all cells for each splicing event. Overall, in real datasets, SCSES consistently outperformed the competing algorithms by achieving higher cell-wise PSI correlations and lower event-wise PSI estimation errors (Fig. 2a, b, Supplementary Figs. 11–13). In all cases, SCSES outperformed BRIE1, Expedition and rMATS, exhibiting a significant increase in median SCC values across cells, ranging from 0.1 - 0.6, and a reduction in median RMSE by > 19.4% across events (Fig. 2a, b, Supplementary Figs. 11–13). In most cases, SCSES achieved improved or comparable performance relative to algorithms relying on inference from similar cells, such as BRIE2, Psix and SCASL (Supplementary Figs. 11–13). Furthermore, we synthesized artificial scRNA-seq data to simulate a

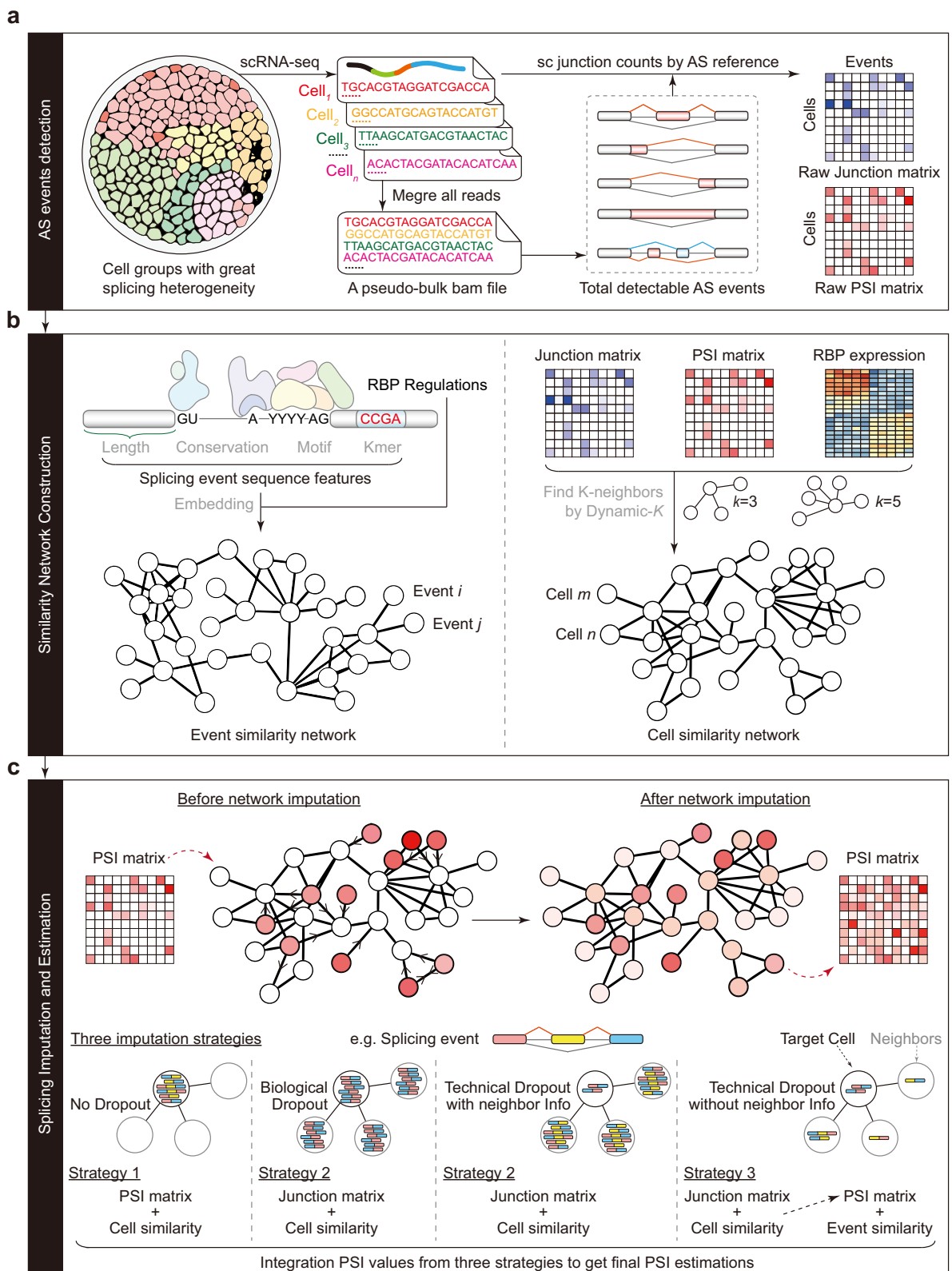

**Fig. 1 | Overview of the SCSES framework. a** SCSES creates a pseudo-bulk bam file by merging all single-cell reads. Conventional AS detection algorithms are used to detect all kinds of AS events. Raw junction read counts can then be retrieved based on this splicing reference. Raw PSI values can be calculated by raw junction read counts. **b** Cell splicing similarities can be built by three types of matrices. AS event similarities are measured by combining the RBP-event regulation relationship and event sequence features. KNN algorithm is used to create cell/event similarity networks. Dynamic-K strategy is introduced to select optimal K for each cell. Random walk with restart (RWR) algorithm is used to capture the global topological similarities in both networks. **c** Event-cell pairs are divided into four types based on the dropout of target cell and neighbor cells. SCSES introduces three imputation strategies for different event-cell groups. Final imputed PSI values are the linear combination of different imputation values with predicted group probabilities as coefficients.

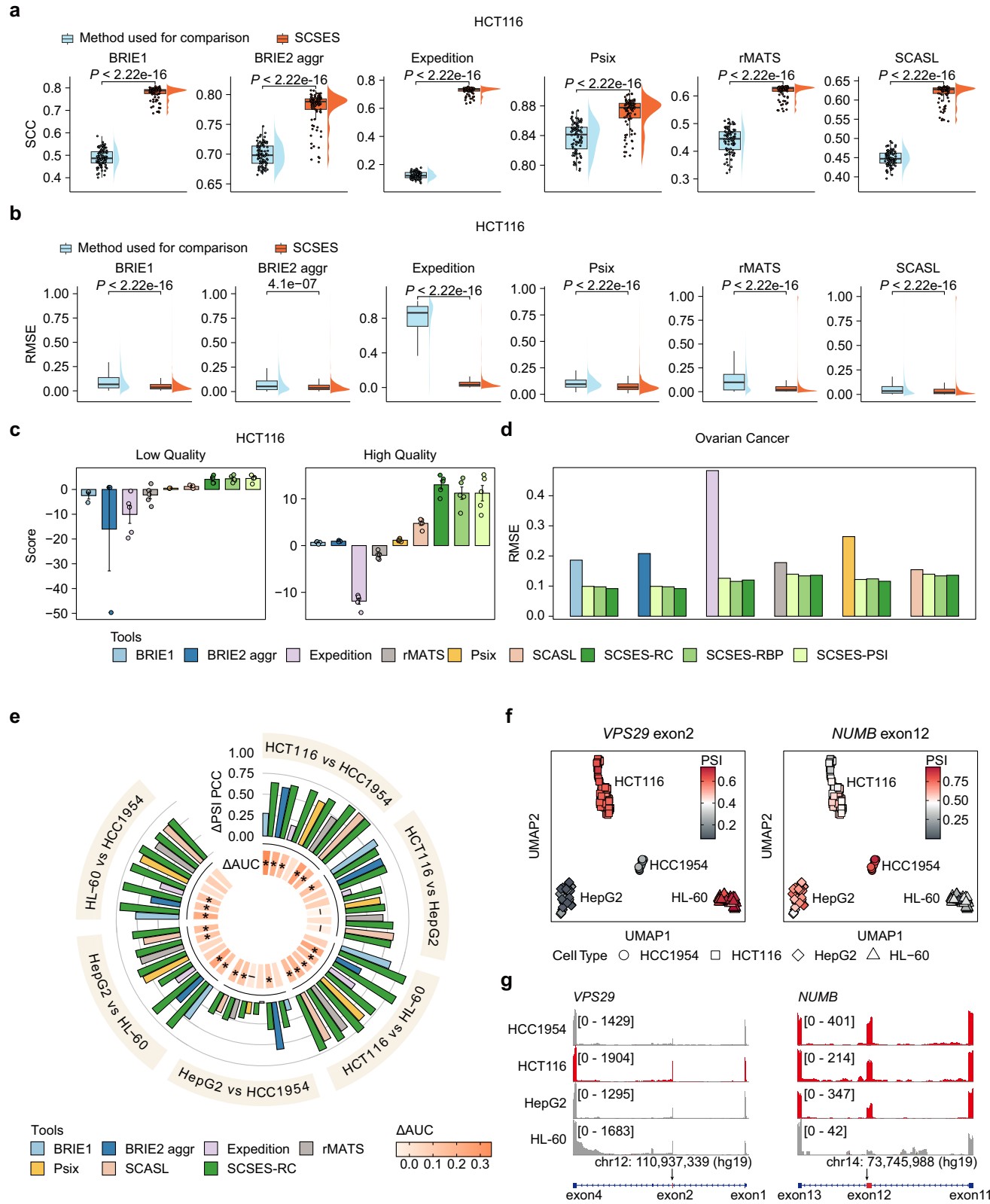

balanced population of the four cell types with varying sequencing qualities via Spanki simulator[37] (Supplementary Fig. 9b, Supplementary Note 2). By testing on these simulated datasets, SCSES obtained the best performance, effectively balancing the accuracy of AS identification and quantification across all cell lines, regardless of the dataset quality (Fig. 2c, Supplementary Fig. 14a–c). To assess the performance of SCSES in real tissues, we applied SCSES and other algorithms to single-cell

datasets from an ovarian cancer sample[38] and the human hippocampus[39], in which each cell was sequenced using paired short-read and long-read technologies (Supplementary Data 3–4, Methods, Supplementary Note 2). The PSI values of high-quality event-cell pairs derived from long-read data were used as benchmarks to calculate the RMSE of PSI estimates from each method (Supplementary Fig. 9d). Considering the substantial differences in detected events by different

**Fig. 2 | SCSES recovers the splicing levels in individual cells. a, b** Raincloud plots showing the performance comparison between SCSES-RC and other algorithms in real scRNA-seq data of HCT116. **a** The SCC value refers to the correlation between the estimated PSI values and benchmarks of all events in each cell. $N_{HCT116} = 91$. **b** The RMSE is calculated between the estimated PSI and benchmarks of an event among all cells. The events detected by both SCSES and the compared algorithm are considered for each comparison. The *P*-values are calculated by the Wilcoxon test (two-sided test) without any adjustments. $N_{BRIE1} = 556$, $N_{BRIE2\ aggr} = 556$, $N_{Expedition} = 1458$, $N_{Psix} = 495$, $N_{rMATS} = 10,313$, $N_{SCASL} = 10,313$. The boxes indicate median (center), Q25, and Q75 (bounds of box), the smallest value within 1.5 times interquartile range below Q25 and the largest value within 1.5 times interquartile range above Q75 (whiskers). **c** Bar plots showing the accuracy scores of different algorithms in HCT116 synthetic dataset. The accuracy score is defined as the product of correlation between inference and benchmarks and AS events recall rate. Error bars represent the standard error of the mean from five independent replicates. **d** Bar plots comparing the performance of different methods on Ovarian Cancer dataset. The RMSE is calculated between the estimated PSI values by different methods and benchmark values averaged on all cell-event pairs. **e** Comparisons of the detected DSEs between SCSES-RC with other algorithms in the real datasets. Bar plot shows the SCC of $\Delta PSI$ in DSEs from the benchmark in each comparison group. Colors represent different algorithms. The inner circle represents the difference of AUC for DSEs identification between SCSES and the compared algorithm. On the circle, *: $AUC_{SCSES} - AUC_{ref} > 0.1$, ·: $AUC_{SCSES} - AUC_{ref} < 0$. **f** Scatter plot showing the inclusion level of *VPS29* exon 2 (left panel) and *NUMB* exon 12 (right panel) between 4 cell lines ($N = 235$). **g** Read coverage showing the inclusion of *VPS29* exon 2 (left panel) and *NUMB* exon 12 (right panel) on the bulk RNA-seq of four cell lines. Alternative exons are highlighted in red in the genome annotation track. Source data of panels is provided as a Source Data file.

methods (Supplementary Fig. 14e–h), we used only the common high-quality event–cell pairs between SCSES and each compared method for the evaluation. SCSES consistently achieved the lowest RMSE across all comparisons. Notably, RMSE was reduced by over 13% in the ovarian cancer dataset and by ~16% in the human hippocampus dataset by SCSES (Fig. 2d, Supplementary Fig. 14d). In general, the algorithms based on inference from similar cells (SCASL, Psix, BRIE2) perform better than others. However, these algorithms require external information to estimate PSI. For instance, BRIE2 incorporates cell identity (e.g., cell type or developmental stage) as a prior feature for Bayesian regression. In contrast, Psix and SCASL require users to predefine the number of nearest neighboring cells, which constrains the learning of splicing information to fixed-size local cell populations. These results suggest that SCSES can recapitulate the splicing features with higher accuracy automatically.

Subsequently, we assessed the capability of detecting differentially spliced events (DSEs) between these four cell types using the receiver operating characteristic curve (Methods, Supplementary Data 5). In general, SCSES demonstrated superior accuracy compared to other algorithms by promoting the area under curve (AUC) > 0.1 in most cases (Fig. 2e). Moreover, the splicing changes detected by SCSES aligned more closely with real cell bulk sequencing results than those reported by other methods (Fig. 2e, Supplementary Fig. 15). These outcomes suggest that SCSES can provide more reliable PSI values for DSE identification. Notably, SCSES could capture a broader range of splicing events and event types than algorithms designed for scRNA-seq data (Supplementary Fig. 16a). For example, SCSES successfully identified the inclusion of exon 2 in *VPS29* transcript in HCT116 cells, which is consistent with observations from bulk RNA-seq data of HCT116 and recent studies[40] (Fig. 2f, g). However, this splicing variant was missed by all other algorithms for scRNA-seq data. Additionally, SCSES was the only method to uncover that *NUMB* exon 12 was more frequently included in three solid tumor cell lines (HCC1954, HCT116 and HepG2) than the blood tumor cells (HL-60), in agreement with recent studies[41–43] (Fig. 2f, g). Moreover, most methods are limited to detect cassette exons, thus cannot infer the splicing levels of other splicing types, including A3SS, A5SS, RI, and MXE. In contrast, SCSES accurately captured these additional event types (Supplementary Fig. 16b–d). Taken together, our evaluation demonstrated that SCSES can provide more reliable and comprehensive splicing changes of all types on single-cell level than other methods.

## SCSES reproduces the splicing status across cell populations

To evaluate the biological implications of SCSES, we assessed the capability of imputed splicing profiles to identify cell types and infer cell pseudo-time trajectories. For this analysis, we collected three public datasets with high sequencing quality, including induced human naïve pluripotent stem cells (nPSC)[44], human early embryos (hEE)[45], and induced human pluripotent stem cells (iPSC)[27] (Methods). We generated lower-quality datasets by down-sampling reads from original BAM files and tested if SCSES could recover the splicing changes from the original scRNA-seq data of relatively high sequencing quality (Supplementary Fig. 9d, Methods). The specific *K* value for each cell was shown in Supplementary Fig. 17.

To evaluate the accuracy of cell clustering based on splicing levels estimated by different algorithms, we calculated the normalized mutual information (NMI) between the *K*-means clustering results derived from the estimated PSI profiles and the cell type annotations provided in the original publications of the test datasets. Overall, SCSES showed the best performance, with an average NMI score improvement of >10% on both the nPSC and hEE datasets (Fig. 3a). For the iPSC dataset, the performance of SCSES was slightly higher than Psix and SCASL, and substantially higher than the remaining methods. Interestingly, low dimensional reduction analysis showed that SCSES divided the motor neurons (MNs) into two more subclusters (MN-C1, MN-C2), a distinction that was also captured by the Psix and SCASL but was not as clearly delineated by other methods or by gene expression values (Fig. 3b, c, Supplementary Fig. 18a, b). These two splicing subclusters could also be detected by using the original splicing profile from MN cells (Fig. 3b right panel). Moreover, a set of genes associated with mRNA splicing and neuron development were differentially expressed between MN-C1 and MN-C2 cells (Fig. 3d). Among them, *PTBP1* emerged as the most significantly upregulated splicing factor in MN-C1. This gene has been extensively characterized for its role in regulating neuronal development in neural stem cells and progenitors[46,47] (Supplementary Fig. 18d). In addition, the pseudotime analysis by Monocle3[48] and CytoTrace[49] demonstrated that MN-C1 exhibited stronger stemness and earlier pseudotime compared to MN-C2 cells (Supplementary Fig. 18c, h right panel). Moreover, multiple genes exhibited changes in splicing patterns ($\Delta PSI > 0.1$ and $FDR < 0.05$, Wilcoxon test) while no changes in expression levels ($\log_2 FC < 0.5$ and $P > 0.05$, Wilcoxon test) (Fig. 3e). For example, expression levels of *VPS29* did not show significant differences between MN-C1 and MN-C2 cells ($P = 0.41$, $\log_2 FC = -0.11$, Fig. 3 f left panel), while MN-C1 cells expressed more isoform including 2nd exon (Fig. 3f right panel, Fig. 3g). *VPS29* has been demonstrated to have important functions in synaptic vesicle recycling and synaptic transmission[50]. Collectively, these observations indicate the presence of splicing heterogeneity within MNs, and we suspect that MN-C1 cells are likely in the early stages of differentiation from iPSCs to MNs. Furthermore, we examined the splicing regulatory relationships between RBPs and splicing events. Previous publications have linked *PTBP1* expression to the mutual exclusivity of exon 9 or exon 10 in *PKM*[51], and exon 4 exclusion of *SRSF3*[52]. The imputed PSI values estimated by SCSES for these events were highly correlated with *PTBP1* expression (Supplementary Fig. 18e-g). Additionally, we assessed whether the predicted PSI values could support infer pseudotime trajectories. The pseudotime inferred from original gene expression using monocle3[48] served as benchmarks (Supplementary Fig. 18h). In terms of individual cell, the pseudotime predicted by SCSES-imputed PSI presented a higher correlation with benchmarks across all datasets (Fig. 3h). From the

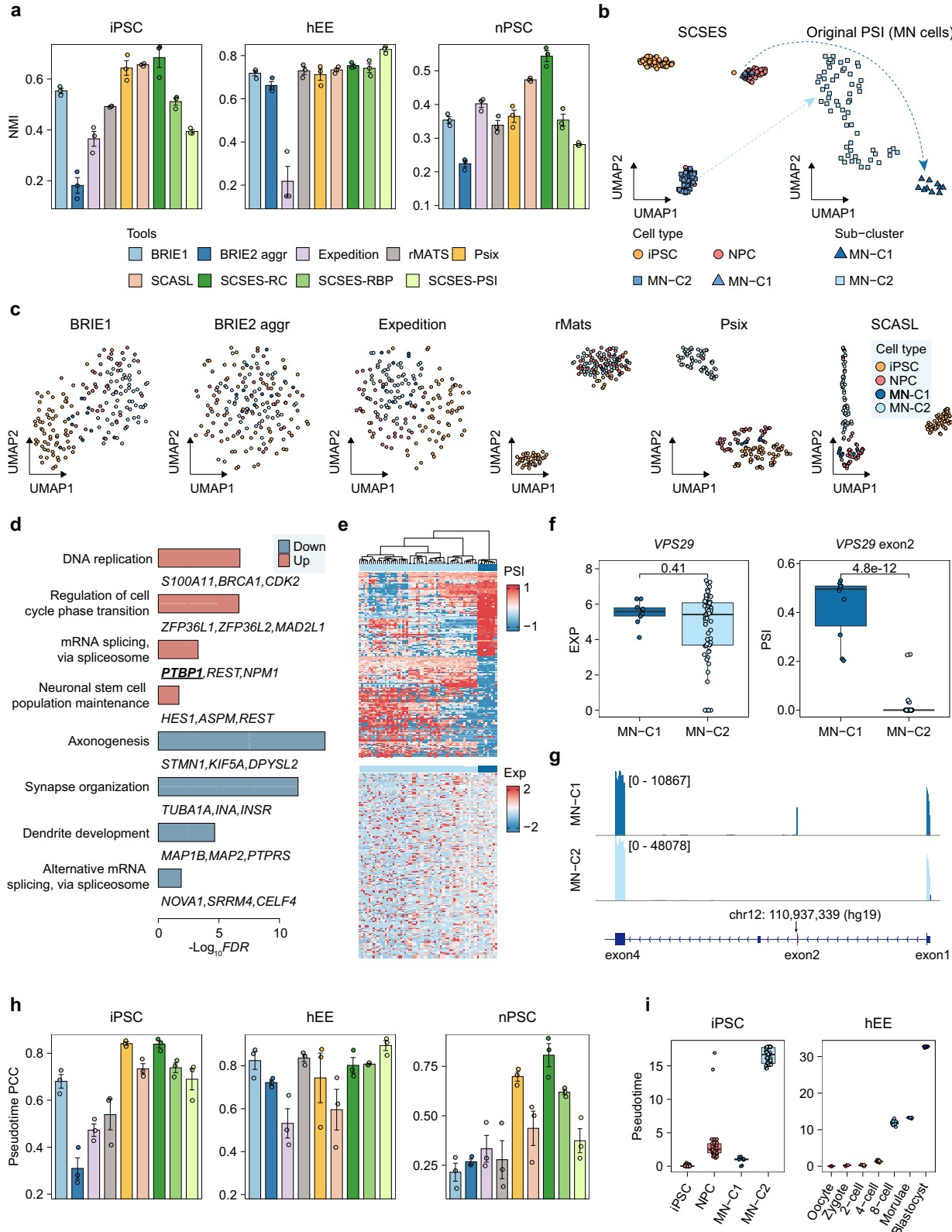

perspective of cell groups, the trajectories were highly consistent with the cell differentiation stages (Fig. 3i, Supplementary Fig. 18i).

## SCSES reveals a cell group at primary diagnosis of multiple myeloma that is associated with potential drug-resistance

Growing evidence indicates that AS contributes to tumor relapse and therapeutic failure in multiple tumors[53]. Here, we investigated the

effect of AS on drug-resistance in multiple myeloma (MM). 127 MM cells by Smart-seq2 sequencing were collected from the same patient before and post treatment with thalidomide and bortezomib (BTZ)[54]. 62/127 cells were collected from bone marrow at diagnosis, and others were collected after metastasizing to ascites dissemination at disease relapse[54]. The conventional gene expression profiles detected two clusters (EC1 and EC2), which matched the cell sampling from initial or

**Fig. 3 | SCSES reproduces the splicing status across cell populations. a** Bar plot of NMI across three test datasets. For each dataset, down-sampling is replicated three times. **b** left panel: UMAP plots of 174 cells from iPSC test dataset by PSI values estimated by SCSES-RC. The colors indicate the cell labels from the previous publication. right panel: UMAP plots show the projection and clusters of MNs using the original PSI profile (non-down sampling data). The shape of points represents subgroups of MN cells. **c** UMAP plots of iPSC test dataset by PSI values estimated by other algorithms. **d** Biological function enrichment analysis of the DEGs in MN-C1 compared to MN-C2. **e** top panel: Heatmap showing PSI profiles of DSEs between MN-C1 and MN-C2 groups, where the corresponding target genes do not exhibit differential expression. bottom panel: Heatmap displaying the expression profiles of the target genes corresponding to the splicing events shown in the top panel. **f** Box plot showing the gene expression (left panel) and PSI (right panel) of *VPS29* between MN-C1 and MN-C2. $N_{MN-C1} = 10$, $N_{MN-C2} = 60$. The *P*-values are calculated by the Wilcoxon test (two-sided test) without any adjustments. **g** Read coverage showing the inclusion of *VPS29* exon 2 on the pseudobulk of MN-C1 and MN-C2. Alternative exons are highlighted in red. **h** Bar plot of PCC between pseudotime inferred by PSI values from test datasets and gene expression benchmark, with calculations on each dataset replicated three times. The color legend is the same with (**a**). **i** Box plot showing the consistency of pseudotime inferred by PSI values and real differentiation stages in iPSC (left panel) and hEE (right panel) dataset. $N_{iPSC} = 63$, $N_{NPC} = 41$, $N_{MN-C1} = 10$, $N_{MN-C2} = 60$. $N_{Oocyte} = 3$, $N_{Zygote} = 3$, $N_{2-cell} = 6$, $N_{4-cell} = 12$, $N_{8-cell} = 20$, $N_{Morulae} = 16$, $N_{Blastocyst} = 30$. Error bars in (**a**) and (**h**) represent the standard error of the mean from three independent replicates. The boxes in (**f**) and (**i**) indicate median (center), Q25, and Q75 (bounds of box), the smallest value within 1.5 times interquartile range below Q25 and the largest value within 1.5 times interquartile range above Q75 (whiskers). Source data of panels is provided as a Source Data file.

recurrent tumor (Supplementary Fig. 19a). In contrast, 12,468 AS events imputed by SCSES detected four clusters, which subdivided each gene expression cluster into two subgroups (EC1 into SC1 and SC2, EC2 into SC3 and SC4), respectively (Fig. 4a, b, Supplementary Fig. 19b). This grouping could not be detected by other splicing inference algorithms (Supplementary Fig. 19c). To investigate the biological relevance of these splicing-defined clusters, we predicted the cell evolution trajectory by pseudotime analysis with monocle3[48] and CytoTrace[49], and both methods reported a highly consistent order from SC1 to SC4 (Fig. 4c, Supplementary Fig. 19d). In addition, we observed that the most variable splicing events by SCSES exhibited a monotonically increasing or decreasing pattern from SC1 to SC4 (Fig. 4d). The previous publication has reported a linear tumor evolution pattern in these cells, with increased copy number variation upon tumor relapse[54]. Similarly, we found increasing copy number changes from SC1 to SC4 (Fig. 4e). Together, these results indicate a continuous cell evolution path by splicing variations. Next, we wished to validate this cell evolution path from the perspective of RNA velocity, which models the transcriptional dynamics using spliced and unspliced reads. We hypothesize that, with the refined splicing information by SCSES, the junction reads supporting alternative intron retentions (IR) can be utilized to better model the RNA dynamics and increase the prediction accuracy. Therefore, we inferred the RNA velocity by scVelo[55] with the imputed IR junctions, and obtained an evolution direction from SC1 to SC4, which was consistent with the above results (Fig. 4f). However, the original scVelo prediction had a total reverse direction from SC4 to SC1, and scVelo using raw IR junctions without imputation also made a disordered prediction (Supplementary Fig. 19e).

Next, we performed marker gene analysis in SC1-SC4 and found that SC1 highly expressed *CD79A*, which is reported to be associated with favorable overall survival[56] (Supplementary Fig. 19f), while SC2-SC4 showed increased expression of genes associated with poor prognosis, including *SRRM2* in SC2[57], *ADAM10* in SC3[58], *AURKA* and *CHEK1* in SC4[59,60] (Supplementary Fig. 19f). Interestingly, SC2-SC4 increased expression of genes associated with BTZ-resistance, including *NFE2L3* and *NOTCH2* in SC2[61–63], *FOXO3* and *MRPL20* in SC3[64], *CDC25B* in SC4[65] (Supplementary Fig. 19f). These results indicate that SC2 cells may already develop BTZ-resistance potential at diagnosis. Thus, we further focused on the transcriptomic changes between SC1 and SC2, and we found that exclusively expressed genes in tumor relapse already demonstrated the same regulation changes in SC2 cells than SC1 (Fig. 4g). Specifically, genes upregulated in SC2 cells were related to cell mitosis, heat shock response and L-glutamine (Gln) process (Supplementary Fig. 19g). Previous studies reported that BTZ resistance can be induced by high levels of heat shock proteins in MM cells[66]. In addition, upregulated Gln metabolism regulators promote proteasome inhibitors resistance in plasma cell myeloma[67]. We also found dysregulation of genes associated with these functions along the cell evolution path from SC1 to SC4 (Supplementary Fig. 19h).

Finally, we analyzed the AS patterns between SC1 and SC2. In total, we detected 248 DSEs between SC1 and SC2, while the overall expression levels of the genes harboring these DSEs were not differentially changed (Supplementary Fig. 20a, Supplementary Data 6). The intensities of splicing changes were highly correlated with the pseudotime prediction from SC1 to SC4 (Supplementary Fig. 20a, b). Genes harboring splicing abnormalities were functionally enriched with protein metabolism and cell death pathways (Fig. 4h). Particularly, 13 genes associated with protein ubiquitination were alternatively spliced (Supplementary Data 6). The mechanism of BTZ treatment involves increasing the endoplasmic reticulum stress and cell apoptosis by inhibiting the activity of proteasome, which leads to the accumulation of ubiquitinated proteins[68]. Here, the aberrant splicing changes in ubiquitination pathway genes may indirectly dysregulate the ubiquitin–proteasome system, which potentially decrease the sensitivity to BTZ. This observation requires experimental supports in further studies. We also found the exclusion of *CADM1* exon 10 and the inclusion of exon 8 and exon9 in SC2 (Supplementary Fig. 20b). *CADM1* isoforms can regulate cell survival and homotypic adhesion in human mast cells[69], but their functions in MM cells have not been investigated. In addition, 19 differentially spliced genes have been reported to be involved in BTZ resistance (Fig. 4i, Supplementary Data 7). For instance, expressions of *EDEM1* and *EPS15* are associated with varying outcomes of BTZ treatment[70,71]. Phosphorylation of *EIF4B* partially leads to the acquisition of BTZ resistance[72]. And a treatment combination of *HUWE1* inhibitors with BTZ can increase the effect of BTZ[73]. These DSEs from BTZ-associated genes persistently presented in SC2 to SC4 cells (Fig. 4i, j, Supplementary Fig. 21a, b), indicating that besides the altered gene expression, AS may also contribute to the BTZ resistance.

To validate the potential BTZ-resistance of SC2 cells, we also analyzed independent scRNA-seq data from a primary MM patient and confirmed that existence of SC2 cells (Supplementary Fig. 22a-d, Supplementary Note 4). Interestingly, by analyzing three additional cohorts of BTZ-treated MM patients, we found that increased activity of SC2 marker genes was associated with worse patient overall survival (Fig. 4k) and disease-free survival (Fig. 4l), and relapsed patients exhibited higher SC2 marker genes activity than non-relapsed group (Supplementary Fig. 22e).

Taken together, with an improved splicing profile by SCSES, we identified a cell subgroup with resistance potential to BTZ treatment at MM diagnosis, which was undetectable by conventional gene expression analysis. Splicing alterations could correctly track the cell evolution trajectory, and reveal the potential mechanism for BTZ resistance.

## SCSES deciphers the splicing dynamics from mesendoderm to definitive endoderm during human embryo development
Previous studies have highlighted the profound impacts of AS during embryonic stem cell (ESC) differentiation with bulk data[74,75]. Here, we used SCSES to evaluate the role of AS during transition from

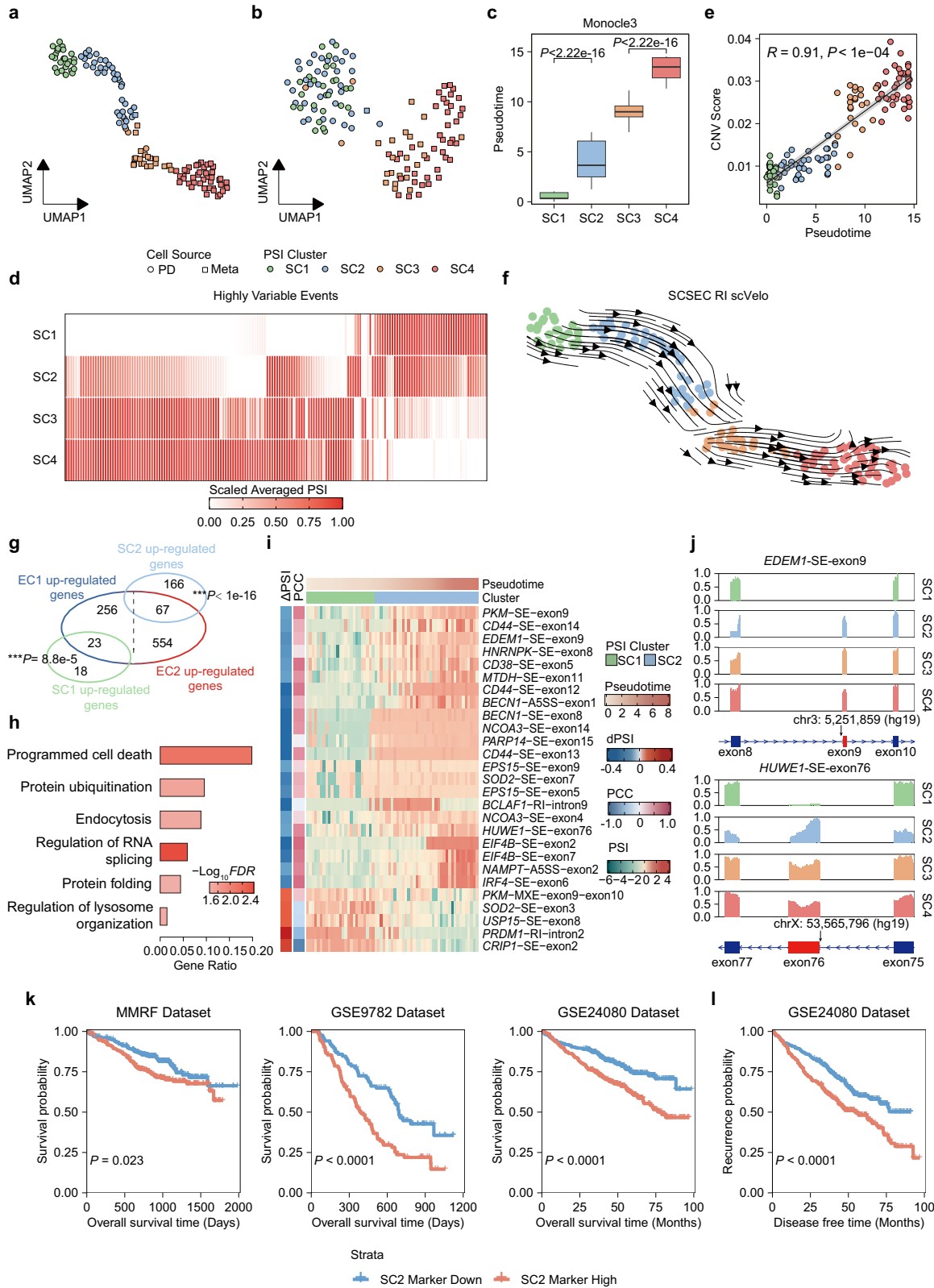

pluripotency maintenance to lineage-specific differentiation on single-cell resolution. To this end, we collected public Smart-seq2 sequenced data on 758 cells from three major differentiation states of H9 hESC (pluripotent state: 0 h, mesendoderm/ME: 12 h and 24 h, definitive endoderm/DE: 36 h, 72 h, 96 h)[76]. Most cells at different time points were clustered into independent groups by SCSES splicing profile, indicating clear splicing dynamics throughout the differentiation course (Fig. 5a), while a small set of cells from 96 h overlapped with cells from 72 h (Fig. 5a). Further analysis demonstrated that the cells appeared to reach a relatively stable state by 72 h, consistent with previous studies[76], and the non-overlapped 96h-cells were in a more mature stage compared to overlapped 96h-cells (Supplementary Fig. 23, Supplementary Note 5). Enrichment analysis on differentially spliced genes (Supplementary Data 8) showed that splicing alterations

**Fig. 4 | SCSES reveals drug resistance profiles in the treatment-refractory multiple myeloma patient. a, b** UMAP plots showing clustering of 127 cells in MM34 patient estimated by splicing profile (**a**) and gene expression profile (**b**). Cell colors represent the clusters, and shapes represent the source of samplings. PD: primary diagnosis, Meta: metastasis. **c** Box plot showing the consistency between PSI clusters and pseudotime estimated by Monocle3. $N_{SC1} = 25$, $N_{SC2} = 38$, $N_{SC3} = 23$, $N_{SC4} = 41$. The *P*-values are calculated by the Wilcoxon test (two-sided test) without any adjustments. The boxes indicate median (center), Q25, and Q75 (bounds of box), the smallest value within 1.5 times interquartile range below Q25 and the largest value within 1.5 times interquartile range above Q75 (whiskers). **d** Heatmap showing the averaged splicing levels of highly variable events in SC1 to SC4. **e** Scatter plot showing the correlation between pseudotime and CNV accumulation. The correlation coefficient (R) and associated *P*-value are calculated by Pearson correlation test ($N = 127$). **f** RNA velocity estimated by imputed raw junction counts of RI events with SCSES. **g** Venn plot showing the overlap of SC1/SC2 DEGs and EC1/

EC2 DEGs. The *P*-values are calculated by the two-sided Fisher's exact test without any adjustments. **h** The gene function enrichment analysis of genes associated with 248 DSEs between SC1 and SC2. **i** Heatmap showing the splicing profiles of DSEs between SC1 and SC2. The associated genes harboring these DSEs were reported to be associated BTZ resistance. $\Delta PSI$ represents the difference of averaged splicing levels in SC1 and SC2. PCC represents the correlation between PSI and pseudotime in all cells. **j** Exon read coverage in two representative DSEs associated with BTZ-resistance genes by merging the reads from the same cell group. Alternative exons are highlighted in red in the genome annotation track. **k** Kaplan-Meier survival curves showing overall survival differences based on SC2 marker gene activities in MMRF, GSE9782, and GSE24080 datasets. Survival differences were evaluated using the two-sided log-rank test. **l** Kaplan-Meier survival curves showing disease-free survival differences based on SC2 marker gene activities in GSE24080 dataset. Survival differences were evaluated using the two-sided log-rank test. Source data of panels is provided as a Source Data file.

were involved in the differentiation progress, such as endodermal cell fate commitment, WNT signaling pathway, and epithelial to mesenchymal transition (EMT) processes (Fig. 5b). By further checking DSEs determining this clustering, we found events that exclusively appeared at unique time points, as well as changes that persistently presented in successive stages (Fig. 5c, d, Supplementary Fig. 24a). For example, *JMJD1C* intron 6 retention specifically appeared at 12 h, while the inclusion level of *DNMT3B* exon 21 underwent a progressive decrease from 0 h to 96 h (Fig. 5d). These results suggest different isoforms are generated at distinct stage of differentiation and their dynamic changes are informative to infer cell state transition. mRNA splicing is orchestrated by numerous RBPs with extensive and dynamic interactions[77]. Next, we constructed a splicing regulatory network between uniquely expressed RBPs at specific time points and regulated DSEs (Fig. 5e, Supplementary Fig. 24b, Supplementary Data 9, Supplementary Note 5). In this network, several RBPs involved in embryonic development presented as hub nodes, which were associated with the large number of DSEs[78–84]. For example, The RBP with the largest number of regulated network targets was *ESRP1*, a critical regulator in epithelial-mesenchymal transition (EMT)[85]. EMT takes place during the ingression of pluripotent epiblast cells through the primitive streak, initiating their differentiation into the mesoderm and DE germ layers[86]. Moreover, we found that *RBM24*, reported to regulate mesenchymal-like splicing patterns[87], was up-regulated in DE stage cells (Supplementary Fig. 24c).

Finally, we focused on the critical transition from ME to DE at 36 h and identified two clusters (EC1, EC2) using overall gene expression without clear demarcation between them. While under the same clustering resolution, the SCSES splicing features yielded three clusters (SC1-SC3) by splitting one gene expression cluster into two subgroups (Fig. 5f, g), which could not be detected by other splicing inference algorithms (Supplementary Fig. 24d). SC1 cells highly expressed DE markers *CXCR4* and *SOX17*, while cells in SC2 and SC3 highly expressed the pluripotency marker *POU5F1*[76] (Fig. 5h), indicating SC1-SC3 corresponding to different differentiation phases. We next inferred the differentiation order of cells by measuring the activity score of differentiation signatures, whose expression monotonically increased from 0 h to 96 h[88] (Supplementary Note 5). In so doing, we found that cells in SC1 had higher differentiation scores than SC2 and SC3 cells (Fig. 5i), which was also confirmed by RNA velocity analysis (Fig. 5j). These results suggest the SC1 cells as a unique cell cluster presenting a late stage of ME to DE transition. The separation of SC1 from SC2 could only be detected by AS changes, rather than overall gene expression. Notably, the PSI changes of *MCM7* 4th intron inclusion, which introduced PTCs in *MCM7* transcripts, was positively correlated with the inferred differentiation order from SC2 to SC1 (Fig. 5k). *MCM7* is reported to be involved in EMT process[89] and exhibited lower expression in SC1 as well as monotonically decreased expression along the differentiation times (Supplementary Fig. 24e, f), suggesting that AS is likely to contribute to

EMT by affecting *MCM7* expression. Moreover, the inclusion levels of *USP8* exon 3 and *ERBB4* exon 26 exhibited a significant negative correlation with the differentiation order without altering gene overall expressions (Supplementary Fig. 24e, f). Based on previous studies, *USP8* preserves stemness of ESCs[90], and the *ERBB4* isoforms lacking exon 26 prevent activation of the PI3K signaling pathway[91], facilitating ME to DE differentiation[92]. These results imply that AS may drive ME to DE transition by modulating transcript compositions, and SCSES helps to reveal critical splicing alterations during this transition.

## SCSES identifies an activated monocyte subtype in induced HSC differentiation

Droplet-based protocols, such as inDrop[26] and 10x Chromium[93], are still the most commonly used technique for single-cell sequencing, with lower cost and higher throughput compared to full-length protocols. However, they introduce more challenges for detecting AS events due to the biased read coverage and low read depth, highlighting the need for SCSES for AS correction. To test SCSES on droplet-based data, we explored the splicing heterogeneity during hematopoietic stem cell (HSC) differentiation with lineage-tracing cells prepared by inDrop-v3[94]. 31,542 mouse cells were collected for 2-, 4-, and 6 day culture in vitro, and 11,192 of them had definite clonal fates. To expand the splicing events pool in the inDrop-seq dataset, we added the alternative last exon events detected by MAJIQ into our analysis (Supplementary Note 6). Totally, 2978 valid AS events were detected and quantified. Compared to the SMART-seq dataset[95], the read coverage in inDrop data was enriched near the 3'end of genes (Supplementary Fig. 25a). This 3' biased read coverage influenced the detection of splicing events towards the 3' ends of genes (Supplementary Fig. 25b). Consequently, AS events in the 3' untranslated regions (3' UTRs) were more frequently detected in inDrop sequencing data compared to SMART-seq data (Supplementary Fig. 25c). 13 cell clusters were detected by combining gene expression profile and AS profile of 2,978 AS events (Supplementary Fig. 26a). Cell clusters were highly consistent with major cell types defined in the original publication (Supplementary Fig. 26b). Moreover, some cell types could be further distinguished into several subgroups by integrating AS profiles after SCSES imputation (Supplementary Fig. 26c), such as neutrophils (cluster1, 2, 3, 5, 7, 9) and monocytes (Mono1, Mono2, Mono3). The three subgroups of monocytes exhibited clearly different UMAP projections by SCSES-imputed splicing changes, which could not be detected by gene expression clustering (Fig. 6a). This analysis demonstrates the capability and application of SCSES in uncovering AS heterogeneity in droplet-based data.

We next sought to understand the difference between the three monocytes AS subgroups (Supplementary Data 10). Comparing the culture time of cells, Mono3 contained more cells cultured for 2 days, and fewer cells cultured for 6 days (Fig. 6b), suggesting cells in different subgroups may have distinct differentiation efficiency. The

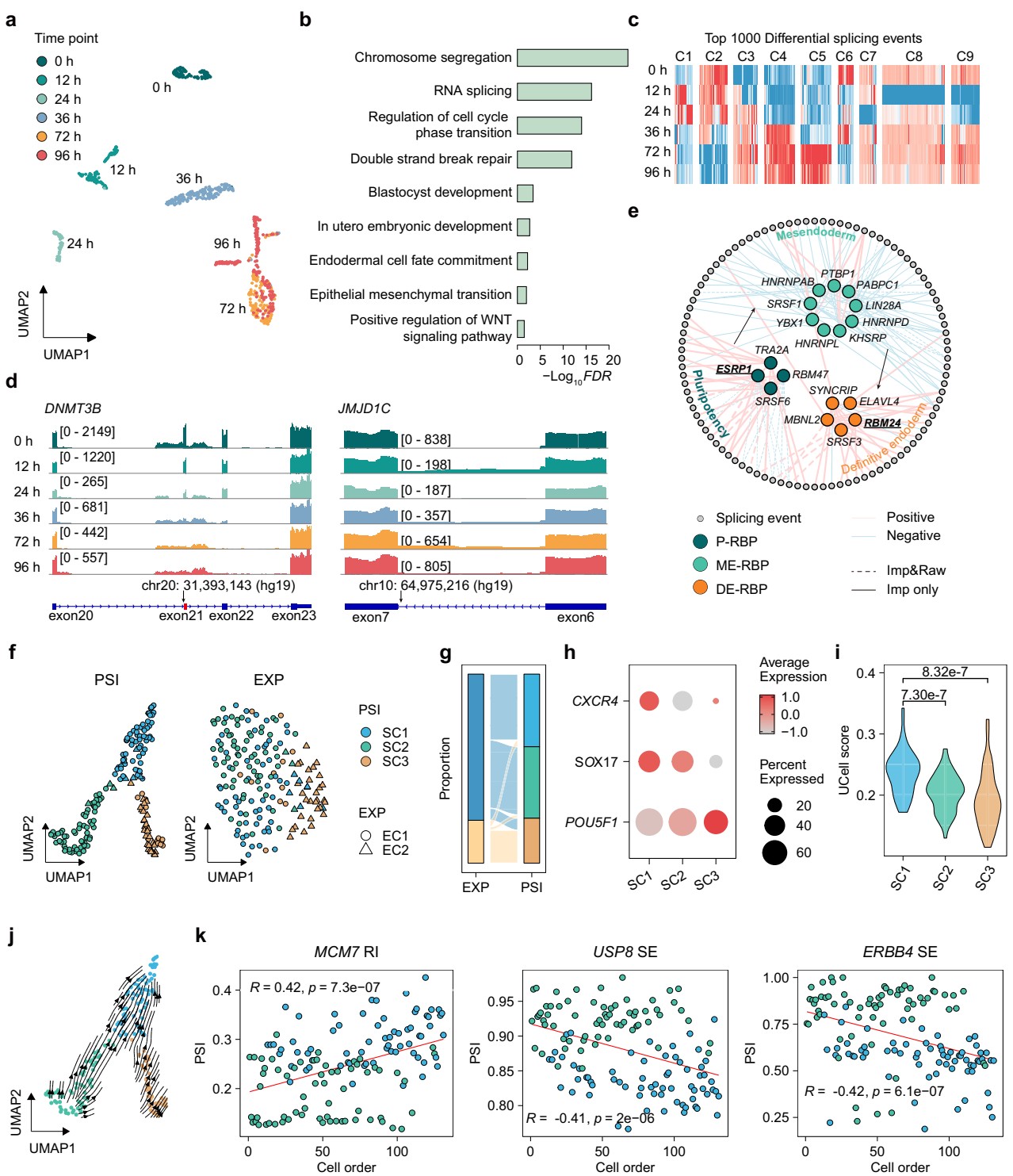

inclusion level of exon 6 in *Runx1* was strikingly elevated in Mono3 (Fig. 6c). *Runx1* is reported to promote the maturity of granulocytic and monocytic cells[96,97]. Moreover, *Runx1* isoform including exon 6 can promote myeloid lineages commitment of multipotent progenitor cells within the mouse fetal liver[98,99]. These results suggest that *Runx1* isoforms with alternative exon 6 may also lead to different differentiation speed in monocytes. In addition, Mono3 exhibited higher expression of the primary granules genes, including *Ctsg*, *Elane* and *Prtn3*, indicating a "neutrophil-like" feature[100] (Supplementary Fig. 26d). It is reported that some monocytes share common progenitors with granulocytes[101], which was validated by the prevalence of

neutrophils within the 6 day cell population derived from the Mono3 specific clones (Supplementary Note 6, Fig. 6d). Besides, it is known that autophagy is pivotal for monocytes survival and differentiation[102]. We identified splicing changes in 19 autophagy-related genes (Fig. 6e). For example, *Lamp2* generates 3 isoforms, *Lamp2a*, *Lamp2b* and *Lamp2c*, by altering the last exon (Supplementary Fig. 26e). Within Mono2, there was a notable enrichment of the *Lamp2a* (Fig. 6f left panel). *Lamp2a* plays an important role in lysosomal docking of HSC70-substrate complexes and substrate translocation into the lumen involved in chaperone-mediated autophagy[103]. *Sh3glb1* displayed a preference towards the transcripts missing exon 11 in Mono2,

**Fig. 5 | SCSES improves the inference of cell state transition in human embryo development. a** UMAP of 758 cells by SCSES splicing profile. Colors indicate cell sampling time points along the differentiation from pluripotent state through mesendoderm to definitive endoderm. **b** Biological function enrichment analysis of the differentially spliced genes. **c** Heatmaps showing the mean PSI value of top 1000 DSEs in each time point. The color shows the *z*-score of the PSI value. Clustering at the event level resulted in nine clusters. **d** Pseudobulk read coverage showing the inclusion of *DNMT3B* exon 21, *JMJD1C* intron 6 at each time point. Alternative exons are highlighted in red in the genome annotation track. **e** Splicing regulatory network between differentially expressed RBPs and DSEs. The RBPs annotated as splicing factors and regulating >70 DSEs are selected. Edges whose correlation coefficient is >0.5 are kept. For each RBP, 10 DSEs with strongest correlation are shown. Colors for RBPs indicate different differentiation stages. Edge Colors indicate a positive or negative correlation between RBPs and splicing events.

Line shapes indicate whether the correlation can be detected by raw PSI value. **f** UMAP of 172 cells at 36 h based on transcript splicing (left panel) or gene expression features (right panel), colored by cell clusters using splicing features, shaped by cell clusters using gene expression features. **g** Sankey diagram showing the changes of Seurat clusters from gene expression to transcript splicing. **h** Bubble heatmap showing the expression of *POU5F1*, *SOX17*, and *CXCR4* in the clusters. **i.** Violin plot showing the differentiation scores by a gene signature, which exhibits a continuous increase in expression from 0 h to 96 h. The *P*-values are calculated by the Wilcoxon test (two-sided test) without any adjustments. $N_{SC1} = 66$, $N_{SC2} = 65$, $N_{SC3} = 41$. **j** The velocity fields estimated by scVelo. **k** Scatter plots showing the order of differentiation scores (x-axis) versus the PSI of *MCM7*, *USP8*, and *ERBB4* of cells in SC1 and SC2. The correlation coefficient (R) and associated *P*-value are calculated by Spearman's correlation analysis ($N = 131$). Source data of panels is provided as a Source Data file.

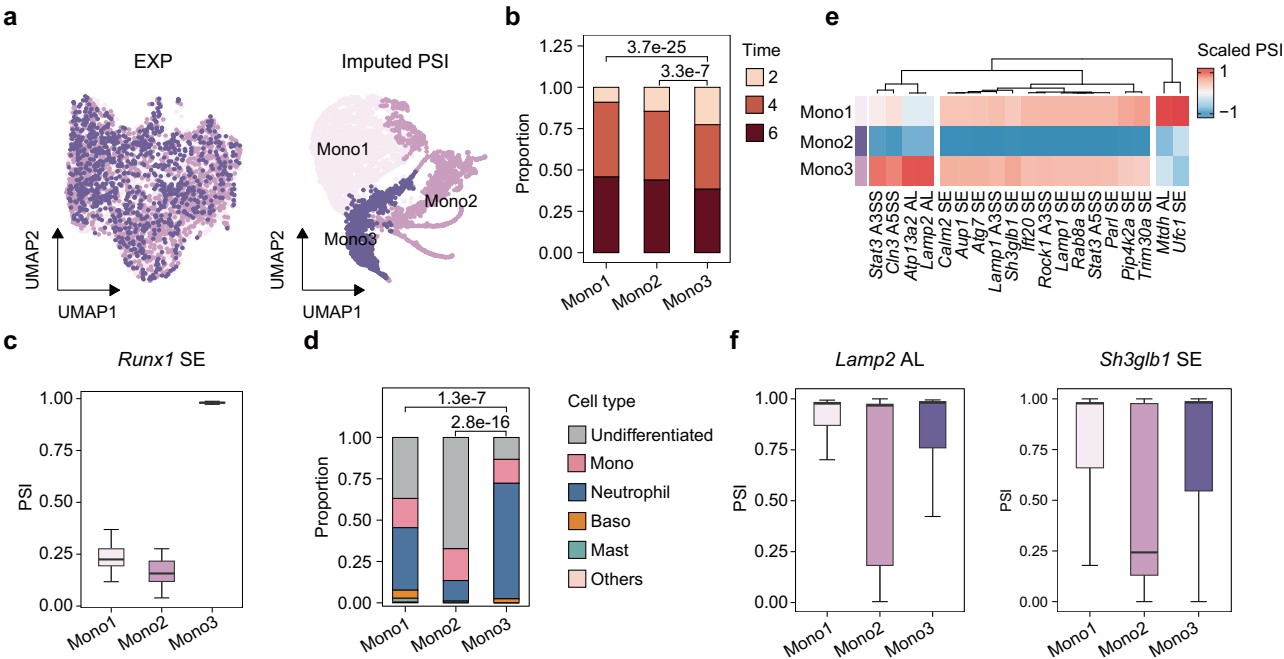

**Fig. 6 | SCSES identifies an activated monocyte subtype in induced HSPC differentiation. a** UMAP plot of 5256 cells from the three monocyte subclasses based on gene expression (left panel) and transcript splicing features (right panel), showing three subclasses: Mono1, Mono2, and Mono3 corresponding to cluster 4, 8, and 10 from Supplementary Fig. 26a. **b** Proportions of the culture time points at which cells were collected in the three monocyte subclasses. The *P*-values are calculated by the one-sided Fisher's exact test without any adjustments. **c** Boxplot of *Runx1* splicing change in the cells of each monocyte subgroup. $N_{Mono1} = 2828$, $N_{Mono2} = 1477$, $N_{Mono3} = 951$. **d** Proportions of the late-stage cell types in each

monocyte subgroup. The *P*-values are calculated by the one-sided Fisher's exact test without any adjustments. **e** Heatmap showing scaled mean PSI among each monocyte subgroup for splicing events associated with autophagy-related genes. AL: alternative last exon event. **f.** Boxplot of splicing events *Lamp2* (left panel) and *Sh3glb1* (right panel) in the cells of each monocyte subgroup. $N_{Mono1} = 2828$, $N_{Mono2} = 1477$, $N_{Mono3} = 951$. The boxes in (**c**) and (**f**) indicate median (center), Q25, and Q75 (bounds of box), the smallest value within 1.5 times interquartile range below Q25 and the largest value within 1.5 times interquartile range above Q75 (whiskers). Source data of panels is provided as a Source Data file.

leading to an incomplete carboxy-terminal Src-homology 3 (SH3) domain (Fig. 6f right panel). The broken SH3 domain may inhibit the autophagosome formation by interrupting the interaction between Sh3glb1, UVRAG, and Beclin1-PI3KC3 complex[104,105]. However, the overall expression levels of *Sh3glb1* and *Lamp2* did not change significantly among the monocyte subgroups (Supplementary Fig. 26f). These results imply that monocytes may regulate the autophagy process via generating splicing variants in associated genes. In summary, these results suggest SCSES can be leveraged to analyze droplet-based scRNA-seq datasets and decipher the transcriptional heterogeneity across large populations of cells.

## Discussion

In this study, we introduced a computational method called SCSES to improve the identification and qualification of splicing alterations at

single-cell level. By simulation studies, SCSES outperformed existing methods in recapitulating splicing changes of individual cells and between cell groups. More intriguingly, SCSES demonstrated the ability to reveal new cell clusters with unique splicing pattern that are biologically relevant, which could not be detected by single-cell gene expressions or current methods on splicing analysis.

Notably, SCSES works on data from various single-cell sequencing platforms including droplet-based protocols, which are incompatible with the majority of current methods. These good performances of SCSES could be explained by the innovative data imputation strategies, which consider the specificity of each cell-event pair based on the abundance of alternative junctions in the target cell and neighbor cells. SCSES firstly aggregates the splicing status from both similar cells and similar events, and then estimates the splicing changes of all cell-event pairs, whether this pair is associated with "dropout" or not. The recent

method SCASL only works on events where both inclusion and exclusion junctions are missed. However, such events may take up only <20% of the total splicing events. Raw PSI values of the other events (>80%) could also be incorrect due to the limited junction reads counts in each single-cell (Supplementary Fig. 27). Finally, SCSES integrates imputation results on each cell-event pair with different biological scenarios, under which alternative choices of similarity and imputation methods have distinct performances (Supplementary Figs. 7, 8). Through systematic comparison of the integrated strategy versus individual strategies, we demonstrated that the integrated approach improved PSI estimation accuracy and enhanced clustering consistency in most cases (Supplementary Fig. 28). Additionally, several carefully designed computational procedures, such as similarity diffusion and fine-tune procedure of the scenario decision model, provided additional performance enhancements, validated by ablation studies (Supplementary Figs. 29–30, Supplementary Note 7). Furthermore, it is well recognized that different splicing event detection tools often produce markedly different outputs, which can lead to conflicting interpretations (Supplementary Fig. 14e–h, 16a)[106]. To address this limitation, SCSES has been designed as an open framework that allows users to incorporate event lists generated by their preferred detection tools. Besides defining novel cell clusters, SCSES may also contribute to infer cell evolution trajectory. We would use SCSES to construct the relationships between gene expression, AS events and cell pseudotime, as well as to detect critical RBPs and splicing events that drive cell evolution.

There are multiple strategies to improve SCSES. In the current SCSES Strategy 1, raw PSI values are used directly. However, the reliability of a PSI value largely depends on its supporting junction read counts. PSI values with higher reliability provide more accurate information for imputing splicing profiles in similar cells. Therefore, incorporating PSI confidence scores into SCSES could enhance the accuracy of the estimated PSI values. SCSES highly relies on the cell splicing similarity network, of which the accuracy may determine the final performance. In SCSES, we used the gene expressions of RBPs, raw PSI, or raw junction read counts to calculate cell splicing similarity in current version. RBPs are crucial regulators of RNA splicing process, which bind directly to specific RNA sequences and control key steps such as splice site selection and spliceosome assembly. However, transcript-level RBP expression may only indirectly reflect the actual functional splicing activity within individual cells. In this context, junction read counts and PSI values provide a more direct representation of cellular splicing states, while their quantitative accuracy is constrained by the inherent sparsity of junction reads in single-cell data. Moreover, RBP expression is only one of many layers involved in splicing regulation, other factors, such as post-transcriptional and post-translational modifications of RBPs, RNA modifications (e.g., m6A), and splicing-associated noncoding RNAs, can also affect the splicing process. Therefore, integrating these diverse regulatory factors into our modeling framework would improve the depiction of cellular splicing heterogeneity. Another way is to implant the cell splicing similarity from some optimization problems instead of defining it directly. For instance, gene expression imputation can be achieved by non-negative matrix factorization (NMF)[107], which inspires us to use multiview NMF algorithms to impute gene expression and splicing changes simultaneously. Similarly, gene expression and splicing similarities can be modeled by searching the optimal cell embeddings that give stable cell clusters in the latent spaces. Here, a number of deep learning algorithms, like autoencoder, can also be applied. Furthermore, despite the good performance of SCSES on droplet-based data, it is still difficult to estimate the intensities of splicing events that are away from the 3′-end of transcripts. To address this, generative artificial intelligence models can be used. For example, we can use large-scale full-length single-cell data or bulk data to pre-train a variational autoencoder, which captures the association between gene expression and splicing events. Then, droplet-based data could be used to fine-tune the model to generate the complete splicing alterations.

## Methods

### SCSES pipeline

**Data processing.** The input data of SCSES is sequence alignment files (SAM/BAM) of scRNA-seq data. For full-length data, gene expressions are extracted by featureCounts[108] and normalized by TPM (Transcripts Per Million). For UMI-base data, gene expressions are extracted by UMI-tools and normalized by library size[109]. Junction read counts are calculated by a JAVA program based on HTSJDK packages (https://github.com/samtools/htsjdk). Junction read counts are normalized by library size to account for sequencing depth variability.

**Splicing event detection.** SCSES employs three widely adopted algorithms to identify different AS types: rMATS[19] for skipped exon (SE) events and mutually exclusive exons (MXE) events, MAJIQ[20] for alternative 3′/5′ splice site (A3SS/A5SS) events, and IRFinder[21] for retained intron (RI) events.

**Cell splicing similarity.** SCSES collects the splicing information from cells with similar splicing patterns for imputation. RNA-binding proteins are critical regulators of RNA splicing process, and the gene expression of RBPs can reflect the cell splicing states. Moreover, the junction read counts of events, as well as PSI values, can directly represent cellular splicing intensity. Hence, SCSES uses these three features to quantify cell-cell splicing similarities. For input feature matrix $\boldsymbol{D}_{M \times N}$ with $M$ cells and $N$ features, SCSES performs dimension reduction with PCA. It keeps the principal components (PC) until the cumulative explained variance exceeds 0.9, resulting in a reduced matrix $\hat{\boldsymbol{D}}_{M \times P}$, where $P$ is the number of retained PCs. The distance between cell $c_1$ and cell $c_2$ is calculated by Euclidean distance:

$$\boldsymbol{Dis}(c_1, c_2) = \sqrt{\sum_i \left( \hat{\boldsymbol{D}}(c_1, i) - \hat{\boldsymbol{D}}(c_2, i) \right)^2} \tag{1}$$

To identify cells with the most similar splicing pattern, SCSES selects the K-nearest neighbor (KNN) cells based on $Dis$. Since cell counts are usually variable for different cell types, a dynamic-$K$ strategy is used to adaptively find the appropriate $k$-value for each cell. Specifically, for a cell $c$, the distances to other cells are sorted in ascending order, denoted as $\boldsymbol{dis}_c^{asc} = (d_1, d_2, \cdots, d_{M-1})$. Then SCSES computes the differences between successive distances, and generate a difference vector $\boldsymbol{df}_c = (d_2 - d_1, d_3 - d_2, \cdots, d_{M-1} - d_{M-2}) = (\Delta d_1, \Delta d_2, \cdots, \Delta d_{M-2})$. Next, SCSES calculates the cumulative average of each position in $\boldsymbol{df}_c$, denoted as $\boldsymbol{cdf}_c = \left( \sum_{i=1}^1 \Delta d_i, \frac{1}{2}\sum_{i=1}^2 \Delta d_i, \cdots, \frac{1}{M-2}\sum_{i=1}^{M-2} \Delta d_i \right)$. SCSES computes the average value of positive difference between $\boldsymbol{df}_c$ and $\boldsymbol{cdf}_c$ across all offset pairs, as $\bar{\Delta}_c = \frac{1}{R}\sum_i^{M-3} \left( relu(\boldsymbol{df}_c(i+1) - \boldsymbol{cdf}_c(i)) \right)$, where $relu$ is the linear rectification function and $R$ represents the number activated by $relu$ function. Finally, SCSES searches the elements in $\boldsymbol{cdf}_c$ from beginning to find the first position $k_c$ that makes $\boldsymbol{df}_c(k_c) - \boldsymbol{cdf}_c(k_c - 1) > \bar{\Delta}_c$, where $k_c$ is the $k$-value for cell $c$. SCSES constructs the KNN distance matrix $\boldsymbol{Dis}_{KNN}$ by retaining the distances to the $k$ nearest neighbors for each cell. $\boldsymbol{Dis}_{KNN}$ is transferred to a similarity matrix by Gaussian kernel function:

$$\boldsymbol{S}_{cell}(c_1, c_2) = e^{-\left( \frac{\boldsymbol{Dis}_{KNN}(c_1, c_2)}{\sigma_{c_1}} \right)^2} \tag{2}$$

where $\sigma_{c_1}$ is a zoom factor for cell $c_1$, set to be $d_{\lceil k_{c_1}/3 \rceil + 1}$[31]. Due to the asymmetricity of $\boldsymbol{S}_{cell}$, SCSES symmetrizes the cell splicing similarity matrix by $\boldsymbol{S}_{cell} \leftarrow \boldsymbol{S}_{cell} + \boldsymbol{S}'_{cell}$.

The RBP list of human was manually curated from RBPbase (https://apps.embl.de/rbpbase/) and a well-used resource[110], while the RBPs for mouse were collected from SFMetaDB[111] and RBPDB[112]. SCSES supports user-defined RBP list for different species.

**Event similarity**. SCSES integrates the event sequence similarity and splicing regulation information to measure the global event-event similarities. Sequence features for SE events are collected from BRIE[23], and similar features for other event types are designed by imitating the characteristics of SE events (Supplementary Note 1, Supplementary Data 11). For each event type, SCSES trains an auto-encoder model to extract features and obtains the latent embedding matrix for further processing. Splicing regulation features are defined by the correlation between RBP expression and events PSI. The Pearson correlation coefficients (PCC) between RBPs and events are computed for further processing.

For an event type $et$ with $E$ events, set $\boldsymbol{Emb}_{E \times L}$ as the sequence feature embedding matrix, and $\boldsymbol{PER}_{E \times B}$ as the PCC matrix between events and RBP, where $L$ is the embedding dimension, and $B$ is the count of RBPs. The distances between two events $e_1$ and $e_2$ in $Emb$ and $PER$ are calculated by the normalized Euclidean distance (NED) respectively, defined as:

$$NED_{Emb}(e_1, e_2) = \sqrt{\sum_i \frac{(Emb(e_1, i) - Emb(e_2, i))^2}{Var(Emb(\cdot, i))}} \tag{3}$$

$$NED_{PER}(e_1, e_2) = \sqrt{\sum_i \frac{(PER(e_1, i) - PER(e_2, i))^2}{Var(PER(\cdot, i))}} \tag{4}$$

where $Var(\cdot)$ is the variance of the feature. To merge $\boldsymbol{NED_{Emb}}$ and $\boldsymbol{NED_{PCC}}$, a combination coefficient $\omega$ is calculated by:

$$\omega(e_1, e_2) = \frac{1}{1 + e^{a \times p(e_1, e_2) + b}} \tag{5}$$

where $p(e_1, e_2)$ is the $P$-value of PCC between $\boldsymbol{PER}(e_1, \cdot)$ and $\boldsymbol{PER}(e_2, \cdot)$, $a = -2\ln(9999)$ and $b = \ln(9999)$ in SCSES. The integrated distance between events is merged by

$$Dis_{event}(e_1, e_2) = (1 - \omega(e_1, e_2)) \times NED_{Emb}(e_1, e_2) + \omega(e_1, e_2) \times NED_{PER}(e_1, e_2) \tag{6}$$

SCSES keeps distances of nearest events with fixed $k_{event}$ (10 by default, Supplementary Fig. 31, Supplementary Note 1), and set others to be infinitive, obtaining matrix $\boldsymbol{Dis_{event\_KNN}}$. $\boldsymbol{Dis_{event\_KNN}}$ is transferred to a similarity matrix by Gaussian kernel function:

$$\boldsymbol{S_{event}}(e_1, e_2) = e^{-\left(\frac{Dis_{event_{KNN}}(e_1, e_2)}{\sigma_e}\right)^2} \tag{7}$$

where $\sigma_e$ is a zoom factor, set to be the $\lceil k_{event}/3 \rceil + 1$ th smallest distance with $e_1$. $\boldsymbol{S_{event}}$ is also symmetrize by $\boldsymbol{S_{event}} \leftarrow \boldsymbol{S_{event}} + \boldsymbol{S'_{event}}$.

**Network Diffusion**. To obtain the network global similarity, we perform the random walk with restart (RWR) algorithm on $\boldsymbol{S_{cell}}$ and $\boldsymbol{S_{event}}$, respectively. $\boldsymbol{S_{cell}}$ and $\boldsymbol{S_{event}}$ are normalized by row to obtain transition probability matrix $\widetilde{\boldsymbol{S}}_{cell}$ and $\widetilde{\boldsymbol{S}}_{event}$. For a cell $c$ (or an event $e$), we define the similarity vector at step $t = 0$ to be $\boldsymbol{v}_c^0 = \widetilde{\boldsymbol{S}}_{cell}(c, \cdot)$ ($\boldsymbol{v}_e^0 = \widetilde{\boldsymbol{S}}_{event}(e, \cdot)$). The similarity information is diffused in the whole network with:

$$\boldsymbol{v}_c^{t+1} \leftarrow (1 - \lambda)\boldsymbol{v}_c^t \times \widetilde{\boldsymbol{S}}_{cell} + \lambda \boldsymbol{v}_c^0 \tag{8}$$

$$\boldsymbol{v}_e^{t+1} \leftarrow (1 - \lambda)\boldsymbol{v}_e^t \times \widetilde{\boldsymbol{S}}_{event} + \lambda \boldsymbol{v}_e^0 \tag{9}$$

where $\lambda$ is the restart probability (0.2 by default, Supplementary Fig. 32, Supplementary Note 7). The random walk is carried out for all cells (and events), and we can obtain the similarity matrix $\widetilde{\boldsymbol{S}}_{cell}^t$ and $\widetilde{\boldsymbol{S}}_{event}^t$ at step $t$. After each iteration, we calculate the change in the

similarity matrix by

$$\Delta_{cell}(t) = \frac{SSE\left(\widetilde{\boldsymbol{S}}_{cell}^t, \widetilde{\boldsymbol{S}}_{cell}^{t-1}\right)}{SST\left(\widetilde{\boldsymbol{S}}_{cell}^t, \widetilde{\boldsymbol{S}}_{cell}^{t-1}\right)} \tag{10}$$

$$\Delta_{event}(t) = \frac{SSE\left(\widetilde{\boldsymbol{S}}_{event}^t, \widetilde{\boldsymbol{S}}_{event}^{t-1}\right)}{SST\left(\widetilde{\boldsymbol{S}}_{event}^t, \widetilde{\boldsymbol{S}}_{event}^{t-1}\right)} \tag{11}$$

where $SSE$ is the sum of squares error and $SST$ is the total sum of squares. The walk is stopped when $\Delta(t)$ is less than a certain threshold, which is 0.05 in SCSES by default (Supplementary Fig. 33, Supplementary Note 7). After obtaining the diffused similarity $\boldsymbol{S_{cell}}$ and $\boldsymbol{S_{event}}$, we still keep the $k$ most similar elements for each row, and others are set to be 0. The $k$ values here are the same as those in $\boldsymbol{S_{cell}}$ and $\boldsymbol{S_{event}}$, respectively.

**Imputation**. The PSI value is defined as the ratio between inclusion read counts and total read counts of the event (Supplementary Fig. 1). SCSES employs multiple strategies for PSI imputation, combining different similarities. For an event $e$ consisted of $IJC = \{ijc_1, ijc_2, \cdots, ijc_I\}$ (for $I$ inclusion junctions) and $EJC = \{ejc_1, ejc_2, \cdots, ejc_J\}$ (for $J$ exclusion junctions), three strategies are used for PSI imputation.

**Strategy 1**: given $\boldsymbol{RC_{IJC}^{raw}}$ and $\boldsymbol{RC_{EJC}^{raw}}$ as the raw normalized read count matrices of $IJC$ and $EJC$, respectively, the raw PSI of event $e$ in cell $c$, $\boldsymbol{PSI^{raw}}(c, e)$, is defined as:

$$\boldsymbol{PSI^{raw}}(c, e) = \frac{\frac{1}{I}\sum_{ijc \in IJC} \boldsymbol{RC_{IJC}^{raw}}(c, ijc)}{\frac{1}{I}\sum_{ijc \in IJC} \boldsymbol{RC_{IJC}^{raw}}(c, ijc) + \frac{1}{J}\sum_{ejc \in EJC} \boldsymbol{RC_{EJC}^{raw}}(c, ejc)} \tag{12}$$

We then impute the raw PSI with cell similarities as follow:

$$\boldsymbol{PSI_{PSI}^{cell}} = \hat{\boldsymbol{S}}_{cell} \times \boldsymbol{PSI^{raw}} \tag{13}$$

**Strategy 2**: we first impute raw inclusion and exclusion read counts $\boldsymbol{RC_{IJC}^{cell}}$ and $\boldsymbol{RC_{EJC}^{cell}}$, respectively, and calculate the imputed PSI of event $e$ in cell $c$ by cell similarities, which is formalized as:

$$\boldsymbol{RC_{IJC}^{cell}} = \hat{\boldsymbol{S}}_{cell} \times \boldsymbol{RC_{IJC}^{raw}} \tag{14}$$

$$\boldsymbol{RC_{EJC}^{cell}} = \hat{\boldsymbol{S}}_{cell} \times \boldsymbol{RC_{EJC}^{raw}} \tag{15}$$

$$\boldsymbol{PSI_{RC}^{cell}}(c, e) = \frac{\frac{1}{I}\sum_{ijc \in IJC} \boldsymbol{RC_{IJC}^{cell}}(c, ijc)}{\frac{1}{I}\sum_{ijc \in IJC} \boldsymbol{RC_{IJC}^{cell}}(c, ijc) + \frac{1}{J}\sum_{ejc \in EJC} \boldsymbol{RC_{EJC}^{cell}}(c, ejc)} \tag{16}$$

**Strategy 3**: we combine cell similarities and event similarities for imputation. Specifically, we first impute the PSI using strategy 2, then further impute the results using event similarities:

$$\boldsymbol{PSI^{both}} = \left(\left(\hat{\boldsymbol{S}}_{event}\right)^T \times \left(\boldsymbol{PSI_{RC}^{cell}}\right)^T\right)^T \tag{17}$$

For each imputation strategy, the imputation procedure is performed iteratively until the change between the imputed PSI matrices of two consecutive steps meets a convergence threshold. The loss function is identical to that used in network diffusion, with a default threshold value of 0.05.

**Prediction of event-cell pair scenarios**. To determine the groups that event-cell pairs belong to, we use cascade decision models, including a well-defined model (Model0) and two logistical regression models (Model1, Model2) for the prediction. Model0 is used to determine if

pairs belong to ND (non dropout) or WD (with dropout). Pairs with PSI values not equal to 0 or 1 are classified as ND, and others are in WD. In SCSES, the PSI values of pairs without any supporting junction reads (neither inclusion nor exclusion junction) are also set to 0. Model1 is used to predict the probabilities of BD (biological dropout) and TD (technical dropout), while Model2 is used to predict probabilities of TD-Info and TD+Info. In Model1, 17 features (Supplementary Data 12, Supplementary Fig. 34) are collected to reflect the distribution of junction read counts and PSI values of an event in the target cell and its neighbor cells. In Model2, two more features are added to reflect the effect of cell-similarity imputation (Supplementary Data 12, Supplementary Fig. 34). All features are normalized by min-max normalization before training.

To train these models, we collected scRNA-seq data from five cell lines[35,36] and corresponding bulk RNA-seq data from NCBI Gene Expression Omnibus (GEO) and CCLE. By comparing PSI values between scRNA-seq data and corresponding bulk data, event-cell pairs in scRNA-seq data are separated into four groups. Two logistical models are pre-trained with corresponding features extracted from these single-cell datasets (Supplementary Fig. 35).

To improve the model adaptability for the new dataset, we also provide a procedure to fine-tune these models. We collected a set of splicing events with conserved splicing levels in different human tissues. First, we collected the housekeeping genes[113], essential genes[114], and genes without tissue specificity (https://zenodo.org/records/6408906) from published database. We selected genes presented in all three databases for further processing. Then, we removed genes associated with development or differentiation in GO:BP terms. Next, we collected all annotated splicing events in these genes from GTF file and kept the events whose variation of PSI values among all tissues in GTEx Splicemap were <$10^{-4}$. Finally, we obtained 344 conserved events from 92 genes. We recorded the splicing levels of these 344 events as a reference. For a new dataset, we compare the splicing status of the conserved events in new data with the reference records and give the group definition to each event-cell pair, which is used to fine-tune the pre-trained model above.

**Estimation.** SCSES employs different imputation strategies based on the event-cell pair scenarios: strategy 1 for pairs in ND are imputed with strategy 1, strategy 2 for pairs in BD and TD+Info, and strategy 3 for pairs in T0-Info. For each event-cell pair, the imputation results from these different strategies are combined, using the probabilities of the pair belonging to different groups as coefficients. The final imputation of PSI values is formalized as:

$$PSI_{impute} = P_{ND} \odot PSI_{PSI}^{cell} + (1 - P_{ND})$$
$$\odot \left( P_{BD} \odot PSI_{RC}^{cell} + (1 - P_{BD}) \right.$$
$$\left. \odot \left( P_{TD+Info} \odot PSI_{RC}^{cell} + \left(1 - P_{TD+Info}\right) \odot PSI^{both} \right) \right) \tag{18}$$

where $\odot$ is the element-wise product of two matrices, $P_{ND}$ is an indicator matrix for ND group, $1$ is a matrix with all elements equal to 1, $P_{BD}$ and $P_{TD+Info}$ are the probability matrices for group BD and TD+Info, respectively.

## Evaluation

**Dataset preparation.** We prepared multiple datasets for the evaluation of SCSES. Bulk sequencing data for four cell lines (HCT116, HCC1954, HepG2, and HL-60) was downloaded from CCLE to serve used as benchmarking datasets. We generated two datasets as test data for evaluation. (1) the real scRNA-seq data of the four corresponding cell lines from GEO (Supplementary Fig. 9a), (2) an artificial synthetic dataset by spanki simulator (Supplementary Fig. 9b). For the latter, we proposed a pipeline to simulate the isoform expression in individual cells from four cell lines by referring isoform expressions of corresponding bulk sequencing data and integrating the noise and low read coverage of

scRNA-seq data. The Spanki simulator[37] was used to generate bam files of different cells according to the isoform expression profiles (Supplementary Note 2). To assess the performance of SCSES in the real heterogeneous single-cell environment, we collected two datasets where paired short-read and long-read sequencing libraries were constructed from individual cells, one of which was derived from high-grade serous ovarian carcinoma tissues[38] (sample P1, 8 cell types), and the other was from adult human hippocampus[39] (sample f1, 16 cell types).

To assess the biological significance, we downloaded three additional scRNA-seq datasets (nPSC, hEE, and iPSC) from GEO. These datasets had average library sizes ranging from 5.6 to 25 million reads per cell (Supplementary Fig. 36a). Down-sampling was performed on these datasets to generate synthetic data with different ratios: 0.15 for the nPSC, 0.03 for the hEE, and 0.1 for the iPSC (Supplementary Fig. 9d). The down-sampling process ensured the resulting data had similar sequencing abundance with standard scRNA-seq data[115] (Supplementary Fig. 36a).

**Evaluation of PSI accuracy.** In the real scRNA-seq dataset, the splicing events detected by different methods were significantly varied, and most methods only considered SE events. Therefore, for a fair comparison, we only used the events identified by SCSES and each individual compared method. Then, the events with high confidence in bulk datasets were selected for evaluation based on the following procedure: (1) the events should be detected by pairwise-comparison algorithms (e.g., SCSES and rMATS) in both single-cell dataset and matched bulk dataset; (2) the junction read counts supporting either inclusion or exclusion exons should exceed 20 in bulk dataset; (3) when read counts of either upstream or downstream inclusion junction have fewer than 5 reads in bulk dataset, the read count fold change between them should be <10; (4) the genes harboring events should be expressed in both single-cell dataset and matched bulk dataset; (5) removing splicing events exhibiting cell-cycle dependence. Specifically, we curated eight cell cycle-synchronized RNA-seq datasets (GSE123958, GSE81485, GSE143275, GSE216497, GSE97774, GSE94479, GSE116131 and PRJEB7566) from GEO and European Nucleotide Archive (ENA). Among these datasets, five contained cell cycle phase annotations in their original publications, while the "CellCycleScoring" function from the "Seurat" package was used to assign cell cycle phases for the remaining three datasets (Supplementary Data 13). Differential splicing analysis was then conducted through pairwise comparisons between all cell cycle phases. Cell cycle-regulated splicing events were defined based on the following criteria: $\Delta PSI > 0.2$ and $P < 0.05$ (Wilcoxon test), with consistent direction of splicing changes across all samples for each phase pair. Subsequently, AS events associated with cell cycle were excluded from evaluation. The event counts for each comparison and each cell are listed in Supplementary Data 2. We computed two metrices: Spearman correlation coefficient (SCC) in cells and root mean squared error (RMSE) in events, to assess the discrepancy between estimated PSI values and the benchmarks. The SCC between the imputed PSI and benchmark PSI in cell $c$ is defined as:

$$SCC_c = 1 - \frac{6\sum_{i=1}^{n}(IPR_c(i) - BPR_c(i))^2}{n(n^2 - 1)} \tag{19}$$

where $IPR_c(i)$ and $BPR_c(i)$ are the imputed PSI rank and benchmark PSI rank of event $i$ in cell $c$, respectively, $n$ is the event count. And the RMSE between imputation PSI $IP_e$ and benchmark PSI $BP_e$ in event $e$ is defined as:

$$RMSE_e = \sqrt{\frac{1}{M}\sum_i (IP_e(i) - BP_e(i))^2} \tag{20}$$

where $IP_e(i)$ and $BP_e(i)$ are the imputed and benchmark PSI values of event $e$ in cell $i$, respectively.

In the synthetic dataset, which had known SE events (detected by SUPPA[116]) as ground truths (Supplementary Note 2), the union of events detected by different methods are used to calculate the accuracy score, which is defined to assessing the accuracy of event identification and PSI estimation. The accuracy score of algorithm *alg* in cell *c* is defined as:

$$ACS_{c,alg} = \begin{cases} SCC_{c,alg} \times rec_{alg} & if\ SCC_{c,alg} > 0 \\ SCC_{c,alg} \times \left(0.5 - rec_{alg}\right) & if\ SCC_{c,alg} \leq 0 \end{cases} \quad (21)$$

where $rec_{alg}$ is the events recall rate of algorithm *alg*, and $SCC_{c,alg}$ is the SCC value between *alg* imputed PSI values and benchmarks in cell *c*. The accuracy score of algorithm *alg* is calculated by the summing accuracy scores of all cells.

In paired long-read and short-read datasets, to ensure a fair comparison, we used only the SE events identified by both SCSES and each individual method under comparison. High-confidence event–cell pairs from the long-read data were used as the benchmark; these pairs were defined as splicing events supported by >10 long-read molecules in a given cell. The reference PSI values were computed based on the ratio of long-read molecules including the alternative exon versus the total number of long-read molecules including/excluding the exon.

**Evaluation of differentially spliced event detection capacity.** Only the events detected by both SCSES and the compared algorithm were used for assessing the capacity of differentially spliced events (DSEs). The events with $\Delta PSI > 0.2$ between two cell lines in bulk sequencing data were designated as the ground-truth DSEs. In the single-cell data, we constructed the receiver operating characteristic (ROC) curves using the *P*-values from the Wilcoxon test as thresholds. The area under the ROC curve (AUC) was used to evaluate the performance of differential splicing event detection. Additionally, we assessed the consistence of dysregulation direction by calculating the SCC of $\Delta PSI$ between imputation results and benchmarks.

**Evaluation of cell clustering and cell trajectory.** The cell clustering was executed in Seurat pipeline. For all compared methods, top 30% highly variable splicing events were selected to execute principal component analysis (PCA), and the PCs with cumulated variance >0.9 were obtained for clustering. *k*-means algorithm was used to detect cell clusters, with *k* set to the number of cell types annotated in the original paper ($K_{nPSC} = 6$, $K_{hEE} = 7$, and $K_{iPSC} = 3$). UMAP was used to visualize cell clusters in each dataset. For SCASL, we used its built-in functions to generate the cell clustering results and UMAP embedding. The normalized mutual information (NMI) was used to evaluate the consistence between splicing-based clusters and reference cell types.

Monocle 3 (v1.3.5) was used to infer cell developmental trajectories based on both gene expression profiles and splicing profiles. The starting cell types for trajectory inference were designated based on biological knowledge, i.e. inner cell mass for the nPSC data, oocyte for the hEE data, and induced pluripotent stem cells for the iPSC data.

**Implementation of SCSES application.** The analysis of MM, hESC and HSC datasets was conducted using the Seurat package in R. In each dataset, dimension reduction and cluster identification were performed with the same parameters for gene expression and splicing profiles (Supplementary Note 4–6). Enrichment analysis was performed by gProfiler (v0.2.3)[117] and clusterProfiler (v4.6.2)[118] packages in R. DEGs and DSEs were identified with Wilcoxon test. For DEGs, the fold change was calculated by the ratio of mean expressions in two group cells. For DSEs, the $\Delta PSI$ was calculated by the difference of mean PSI in two group cells. RNA velocity was performed by scVelo package in Python. Other details are shown in Supplementary Note 4–6.

**Statistics and reproducibility.** The statistical tests have been described in the sections above, Supplementary Note and in the figure legends. No statistical method was used to predetermine sample size. In multiple myeloma dataset, to focus on the cells with BTZ resistance, cells from patient MM16 were excluded in analysis. In inDrop dataset, to remove cells with low quality, cells with <500 expressed genes or library size <1000 were removed. No additional data were excluded from the analyses. The experiments were not randomized. The Investigators were not blinded to allocation during experiments and outcome assessment.

**Reporting summary**

Further information on research design is available in the Nature Portfolio Reporting Summary linked to this article.

## Code availability

The open-source SCSES R package and tutorial are available at GitHub (https://github.com/lvxuan12/SCSES), https://doi.org/10.5281/zenodo.17087025.

## Data availability

All datasets analyzed in this study are publicly available. The bulk RNA-seq data of cell lines were downloaded from CCLE Project from NCBI Sequence Read Archive (SRA) under accession codes SRP186687. The single-cell RNA-seq data HCC1954 and HL60 were downloaded from NCBI SRA under accession codes SRP041736. The single-cell RNA-seq data of HCT116 and HepG2 were downloaded from Gene Expression Omnibus (GEO) under accession codes GSE150993. The paired long-read and short-read of ovarian cancer dataset was downloaded from SRA under accession PRJNA993664. The paired long-read and short-read of human hippocampus dataset was downloaded from The Neuroscience Multi-omic Data Archive (NeMO) under identifier nemo:dat-ho986e6 and nemo:dat-unjyo0u, respectively. The single-cell RNA-seq data of human naïve pluripotent stem cells, human early embryos, and induced human pluripotent stem cells were downloaded from GEO under accession codes GSE171820, GSE36552 and GSE85908, respectively. The cell-cycle synchronized datasets used for removing cell-cycle-dependent splicing events were downloaded from GEO under accession codes GSE123958, GSE81485, GSE143275, GSE216497, GSE97774, GSE94479, GSE116131 and PRJEB7566. The single-cell RNA-seq data of multiple myeloma patients were downloaded from GEO under accession codes GSE110499. The single-cell RNA-seq dataset for validation of multiple myeloma were downloaded from GEO under accession codes GSE118900, GSE9782, GSE24080 and TCGA MMRF project. The single-cell RNA-seq data of H9 human ESC differentiation were downloaded from GEO under accession codes GSE75748. The inDrop-v3 RNA-seq data of mouse hematopoietic stem cell were provided by Caleb Weinreb from Allon Klein Lab in Department of Systems Biology, Harvard Medical School, which can be accessed by Dropbox repository. Source data are provided with this paper.

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

## Acknowledgements

This work is supported by the National Key R&D Program of China (grant 2023YFF0725400, grant 2025YFF1207901 and 2022YFC2704202 to Z.L.), the National Natural Science Foundation of China (grant 32300558 to X.W., 62403390 to P.Z.W.), the Beijing Natural Science Foundation (grant Z220012 to Z.L.) and Chinese Academy of Sciences Hundred Talents Program (to Z.L.). We acknowledge all participants involved in the study.

## Author contributions

Z.L. conceived and supervised the entire project. X.W. and X.L. designed and implemented the method. P.Z.W. contributed to the method development. X.L. performed the evaluation on real datasets. N.H. per-formed the evaluation on synthetic datasets. X.L. and W.X performed the application of MM, hESC, and mHSC. D.G. and L.Z. downloaded and prepared datasets for evaluation and application. X.W. and X.L. wrote the manuscript with the feedback from Z.L. P.Z.W. modified the manu-script. All authors read and approved the final manuscript.

## Competing interests

The authors declare no competing interests
