## [Transparent Peer Review file · Nature Communications]

Deciphering splicing heterogeneity at single-cell resolution by SCSES

Corresponding Author: Dr Zhaoqi Liu

Version 0:

Reviewer comments:

Reviewer #1

(Remarks to the Author)

The authors developed SCSES, an R package to identify and quantify alternative splicing events from single-cell RNA-seq data. To overcome the high dropout rate and improve alternative splicing profiles, SCSES imputes missing alternative splicing alterations via data diffusion, by learning information from similar cells or events. Detecting and analyzing alternative splicing at the single-cell level is undoubtedly a significant challenge. Current single-cell platforms are often hindered by technical limitations, making this task difficult to achieve. The authors attempt to address these limitations by employing data diffusion techniques.

Yet, in the submitted manuscript, it is difficult to comprehend the significance of the tool's results and its overall contribution, due to the insufficient information and the omission of explanations for the decisions made during its development. The article lacks clarity, and no validation is provided to support the novel results obtained by SCSES.

Major points:

- In the abstract, the authors claim that "Systematic simulation studies demonstrate that SCSES outperforms existing algorithms". In the manuscript, the benchmark is performed in cell lines or simulated data. Both are not real scenarios for single-cell datasets. Simulation is not sufficient alone as a benchmark. Benchmarks on real scenarios would prove the added value of SCSES with respect to other tools.
- A table describing and comparing all the existing single-cell methods mentioned in the introduction (BRIE/BRIE2, Psix, SCASL,..) with respect to SCSES would be useful to understand the specificities of SCSES.
- In the discussion and in results, authors state that SCSES can identify novel clusters with unique splicing patterns that could not be detected by single-cell gene expressions or current methods on splicing analysis. Based only on the reported results and without any additional validation, this statement is not strongly supported. Experimental or orthogonal validation would be needed to support this statement and the usefulness of the tool.
- The authors state that cell splicing similarities can be built with three different types of matrices: RNA-binding protein expressions, RAW RC or RAW PSI matrices. These matrices should be introduced/described in the context of splicing. In particular, SCSES can use RNA-binding protein expressions to measure pairwise cell similarities. How is RNA-binding protein information obtained? Can it be used in multiple species? The use of this information and its relevance should be introduced/described/discussed. What are the advantages/disadvantages of using one matrix with respect to the other two? How to best select input data?
- The authors say "due to the high dropout rate and technical limitations, such matrices are very sparse with limited read counts, leading to inaccurate PSI estimation." Is there data or evidence supporting this? Can this sparsity be quantified for AS types?
- SCSES uses three different strategies to estimate the final PSI value. The rationale for the choices is not described.
- What is the biological relevance of identifying two motor neuron subclusters? Additionally, the fact that differentially expressed genes, associated with mRNA splicing and neuron development, were identified between C1 and C2 cells seems to contradict the initial assumption that AS tells a different story with respect to gene expression. If these clusters are different in terms of gene expression, what is the added value of AS?
- SCSES uses three algorithms to detect different AS types: rMATS, MAJIQ and IRFinder. What is the rationale for this choice? Does the tool require a GTF?
- The article could benefit from additional context to make it more comprehensive and easier to understand also for non-specialist readers. For example, in the abstract, the authors mention that SCSES "imputes the junction count matrix and fills

in missing AS alterations”, but these concepts might be difficult to understand for non-specialist readers.

- In all PSI-based UMAPs, the cells are organized into very well-defined trajectories. Could this be an effect of diffusion, which organizes everything into trajectories? A negative example would be useful (for instance, cell lines, where no trajectories should be present).
- How single-cell data obtained from different samples were analyzed together? From the methods, it seems that integration was not performed. Could this introduce a batch effect? In all UMAP visualizations cells cluster together based on batches rather than biological differences.

Minor points:

- SUPPA and SUPPA2 are only mentioned in the methods section without any references and are absent from the benchmarking analysis. Including these tools in the comparison with SCSES would provide a more comprehensive evaluation.
- Is not clear what the meaning of AS intensity is.
- In the introduction, the authors say: “Lastly, due to the limited junction read counts, the calculated percent splice-in (PSI) values are usually not reliable, which makes the results of Expedition and SCASL incorrect.” Is there evidence or literature supporting this statement?
- Splicing event types should be explained in the introduction.
- What are “event sequence similarities”? Are both exons and introns included in this measure?
- In the introduction, the authors categorized cells according to dropout events in alternative splicing junctions (“with dropout”, “biological dropout” or “technical dropout”, “non dropout”, etc..). Is there a visualization that helps to understand this classification? The text could improve in clarity.
- In the first example, the authors say: “We selected a set of high-confidence splicing events.” How many? How were they selected? What is the definition of high-confidence here?
- “SCSES outperformed in terms of higher correlation and lower averaged errors in real datasets”. It is not clear if the higher correlation is based on PSI values, and with respect to what??
- “0.05~0.6 in Spearman correlation confidence (SCC) values among cells, and reduction of more than 0.05 in root mean squared error (RMSE) among events”. The meaning of this sentence is unclear.
- “mimic a “fixed” population of the four cell types”. Is this a typo?
- The authors state that “BRIE2 requires the identity of each cell as prior information, while Psix and SCASL require a fixed number of neighbor cells as input.” This is not clear, could these requirements be better described?
- The limitation that most methods can only detect cassette exons should be introduced earlier, rather than at the end of the first example.
- In the third paragraph of the results section, the authors say: “To evaluate the accuracy of cell clustering by splicing imputation, the normalized mutual information was used to compare with the cell identities from the original publications. Overall, SCSES achieved the highest consistency with an increased score of more than 10% on nPSC and hEE datasets”. The meaning here is not clear.
- At the end of the paragraph “SCSES reveals a drug-resistance cell group at primary diagnosis of multiple myeloma”, the authors state that “this observation requires experimental support in further studies.” Indeed, without further validation, I would not state in the title of the paragraph that a new subpopulation has been identified.
- What does “AS alternations” mean?
- In the last paragraph of the results section, one or a few specific examples are described to demonstrate that the SCSES approach works properly. Out of how many events? How are the example events selected? Would the example events be detected by the other methods?
- “Lamp2 generates 3 isoforms, LAMP-2A, LAMP-2B and LAMP-2C, by altering the last exon.” Is this alternative polyA rather than splicing?
- In the last paragraph of the results I would also include a discussion on the coverage limitation. Quantification of the bias could also be reported.
- Figure 2a and 2b show pairwise plots. SCESC information is displayed multiple times. Could it be only one overall comparison among all tools?
- Figure 2e: how many DSEs are considered?
- Figure 3: Is the UMAP based on gene expression shown?
- Figure 4a and 4b: How many events? How many genes?

The current manuscript's English needs significant editing to reach publication standards. Here I reported some of the problems:

- The title: “Deciphering splicing heterogeneity on single cell resolution by SCSES”, should be “at single cell resolution”. In the introduction section:
 - “Single cell RNA sequencing (scRNA-seq) techniques make it possible to decipher cell transcriptional heterogeneity on large scale of cells”. Correct with “on a large scale of cells”.
 - “The gene expression specificity in different single cell clusters have been decoded in many studies”. Correct with “has been decoded”.
 - “but most of which neglected”. Correct with “but most of them”.
 - “during pre-mRNA maturity”. Correct with “during pre-mRNA maturation”.
 - “no matter on full-length protocols like smart-seq or the droplet-based protocols like inDrop-seq”. Correct with “even with full-length”
 - “which is independent with the conventional single-cell gene expression analysis”. Correct with “with respect to the conventional”
- Typo in Figure 1A “megre” all reads.

In the results section:

- "It takes scRNA-seq alignments as input, and output refined PSI value of every detected splicing event in each cell." Correct with "outputs" if used as a verb.
- "We then used real scRNA-seq data of these four cells." It is more accurate to say "these four cell lines".
- "We then used real scRNA-seq data of these four cells to test if SCSES could recapitulate the splicing landscape on single cell level". Correct "on" with "at single cell level".
- "And the imputed PSI values by SCSES in these events were highly correlated with PTBP1 expression". Sentences should not begin with "and".
- "Specifically, SC2 cells upregulated genes were related to cell mitosis". Correct with "Specifically, genes upregulated in SC2.."
- "In addition, 19 differentially spliced genes have been reported to involve in BTZ resistance". Correct with "have been reported to be involved in BTZ".
- "By further checking DSEs determining this clustering, we found events that exclusively appeared at unique time point". Correct with "at unique time points".
- "Moreover, we found RBM24". Correct with "Moreover, we found that RBM24".

(Remarks on code availability)

The author says that SCSES is an R package but actually, it requires R, Python, Matlab Compiler Runtime, and Java, as also reported in the GitHub page. Additionally, it requires the manual installation of rMats, MAJIQ, IRFinder and STAR. Would it be useful to create a container grouping together the necessary packages to run SCSES? In the end, I was unable to install it on my machine due to discrepancies with the dependencies. However, this may be an issue on my end rather than a problem with the package itself.

Reviewer #2

(Remarks to the Author)

This paper proposes a novel algorithmic framework, SCSES, for analyzing variable splicing at the single-cell level. Overall, the study demonstrates significant innovation and biological relevance, but there are still some details that need further refinement. The specific comments and suggestions are as follows:

1. The description in line 153 of the paper regarding Fig. 2e is not ideal. While emphasizing the improvement in AUC is important, the specific AUC values should also be provided. Improvement is meaningful only when built upon strong baseline values.
2. It is recommended to further label the MN in Fig. 3c as MN-C1 and MN-C2. Since Psix also has a small blue cluster separated out, distinguishing MN-C1 and MN-C2 is crucial.
3. In line 194 of the paper, it is mentioned that the C1 cells are suspected to be in the early stage of differentiation from iPSCs to MNs. This hypothesis could be preliminarily supported by performing pseudotime analysis combined with UMAP.
4. Can the correlation between 72h and 96h in Fig. 5a be further discussed? Why do these time points partially overlap?
5. It is suggested to add experiments related to model performance, such as ablation studies, to discuss the contribution of each component of the model.

(Remarks on code availability)

Reviewer #3

(Remarks to the Author)

In this paper, the authors present a novel data imputation method, SCSES, designed to analyze alternative splicing (AS) quantification at the single-cell level. In addition to the pseudo-bulk approach based on cell similarity, they also propose utilizing splicing event similarity to address challenging cases, such as technical dropouts without sufficient neighboring cell information (TD-info). The authors list several applications of SCSES. However, there are multiple significant concerns about the validity and robustness of the model.

Major Concerns:

1. Potentially false assumption. The model evaluation primarily relies on bulk RNA-seq data from cell lines, assuming that all cells within a cell line are identical. However, using AS quantification from bulk RNA-seq as the "gold standard" may be problematic, as cells within the same cell line can exhibit heterogeneity (e.g., due to different cell cycle stages or other factors). This assumption could introduce biases or inaccuracies in model evaluation.
2. Concerns about the robustness. The results indicate that for the TD+info cases, Strategy 3 (Strategy 2 + even similarity) performs worse than Strategy 2 alone. This outcome is counterintuitive, as one might expect that incorporating additional information would improve the results. This raises concerns about the robustness and reliability of the model, especially when expanding the input information.
3. Furthermore, the results for Strategy 3 in the ND and BD cases are not presented, which leaves a gap in understanding how this strategy behaves across different conditions.
4. The typical final values of K in the dynamic K algorithm are not provided. The typical K values for individual event-cell groups would help clarify how many similar cells are typically required for accurate imputation, particularly for rare cell types. This could give readers a better understanding of the algorithm's performance in different scenarios.
5. It is surprising that using cell similarity based on RBP transcript expression (which btw needs to be clarified as transcript-level expression, not protein expression), raw counts, and PSI values yields similar performance. Have the authors

considered using whole transcriptome expression data or perhaps randomly selected gene expressions for comparison? This could provide experimental justification into the robustness of the method and whether alternative similarity measures might yield improved results.

6. Due to technical limitations, many AS events, particularly those at the 5' ends of genes, are often missed in scRNA-seq data. How does the model perform when imputing AS events that are detected in bulk RNA-seq but are missing in scRNA-seq data? Specifically, how accurate is the imputation for events that are purely inferred from bulk data and not observed in the single-cell context?

7. The Number of Junction Reads for PSI Estimation. For the raw PSI matrix, are the number of junction reads taken into account during PSI estimation? The accuracy of PSI estimation is highly dependent on the number of reads. For example, if we assume that read assignment to inclusion and exclusion follows a binomial distribution (where n is the number of reads and p is the true PSI), then with 10 reads and a true PSI of 0.5, the 95% confidence interval for the estimated PSI would be [0.2, 0.8]. This demonstrates the significant variability in PSI estimation with low read counts. Did the authors incorporate this variability into their model, particularly when imputing PSI for non-dropout events, where the junction matrix is not used (Strategy 1)? Clarification on this aspect would greatly enhance the understanding of the model's reliability and robustness.

8. Clustering Parameters. The same set of parameters was applied to both gene expression and splicing profiles to derive cell clusters. It would be valuable to explore whether tuning the parameters specifically for the gene expression profile could yield cell clusters similar to those identified using the splicing profile, such as SC1-SC4 in the multiple myeloma dataset. Notably, SC1-SC4 exhibit differential expression in certain genes, suggesting that adjustments to clustering parameters might enable the recovery of SC1-SC4 clusters even when using the gene expression profile alone. In such a scenario, the uniqueness of splicing-based clustering in detecting these subgroups would need to be re-evaluated. It would be beneficial for the authors to perform additional clustering experiments with parameter optimization tailored separately for gene expression and splicing profiles. This could provide stronger evidence to support their claim that splicing-based clustering uncovers novel cell subgroups that cannot be detected through gene expression clustering alone.

9. Tool Evaluation and Benchmarking. Please consider testing Psix in scRNA-seq of multiple myeloma (Extended Figure 12C) and hESC (Extended Figure 15D), as it appears to perform well and may provide additional insights or validation.

10. Formula Clarification for SCC. The formula presented in Line 627 appears to compute the Pearson correlation coefficient rather than the Spearman correlation coefficient. As these two measures assess different types of relationships—Pearson for linear relationships and Spearman for monotonic relationships—it is crucial to ensure the correct term is used to avoid confusion. If the formula is indeed calculating the Pearson correlation coefficient, please revise the description accordingly. Alternatively, if the intention was to compute the Spearman correlation coefficient, kindly provide the correct formula and verify whether the calculations throughout the manuscript reflect this adjustment. Clarifying this point will enhance the accuracy and credibility of the reported results.

11. Alternative Last Exon. The alternative last exons of Lamp2 are detected by SCSES (Figure 6F), yet the methods section specifies that only five classical splicing events (SE, A5SS, A3SS, MXE, and RI) are analyzable by SCSES. This raises the question of whether SCSES is capable of detecting alternative last exons as well. Furthermore, does SCSES also support the detection of alternative first exons? If SCSES indeed identifies these additional splicing events, it would be helpful to clarify this capability in the manuscript, including any modifications to the underlying model or algorithms that enable this functionality. Additionally, if SCSES does not currently analyze these events, it would be valuable to explain how the detection of the Lamp2 alternative last exon was achieved and whether this was an exception or a general feature of the method. Providing such clarifications would improve the transparency and completeness of the methodological framework.

Minor Comments:

1. Typographical Errors

Please address the following typos:

- "SCSEC" (Line 988) should be corrected to "SCSES".
- "SCSECI" (Line 429) should be corrected to "SCSES".
- Correct "Spearman correlation confidence" (Lines 135-136) to "Spearman correlation coefficient".

2. Line 103 "four different strategies"

Update the text to "three different strategies" for correctness.

3. Hyperparameter Explanations

Please provide details on how specific hyperparameters were determined, such as the restart probability in the Random Walk with Restart (RWR) algorithm and the k value for diffused similarity matrix. Including these details would greatly enhance the reproducibility and transparency of the methodology.

4. Line 548–549: Imputation Strategy Iterations

The manuscript states: "In SCSES, each imputation strategy is executed multiple times until the difference between two runs is less than 0.05." Could you clarify what is meant by "the difference" in this context? For example, does it refer to the difference in imputed PSI values, convergence of a loss function, or another metric? Providing this information would improve the clarity of the methods description.

5. The concept of information sharing through data diffusion has been reported in multiple publications on scRNA-seq analysis, primarily in the context of gene expression quantification rather than splicing quantification. The authors can cite these publications to enhance the validity of their work.

(Remarks on code availability)

Reviewer #4

(Remarks to the Author)

(Remarks on code availability)

Version 1:

Reviewer comments:

Reviewer #1

(Remarks to the Author)

I thank the authors for conducting a comprehensive revision of the manuscript. However, I still have some concerns:

1. One primary concern is that, in spite of the substantial revisions requested, the authors chose not to modify any panels within the main figures. This absence of visual integration significantly detracts from the revision's perceived strength, inadvertently communicating that the extensive additional analyses, benchmarks, and results are not sufficiently critical or robust to warrant inclusion in the central presentation of their work.
2. Benchmark in real scenarios. The authors included one additional benchmark from 174 iPSC-derived cells, sequenced with C1 + SMARTer-seq. I would argue this is still a scenario where the heterogeneity is limited with respect, for example, to a real tissue. Furthermore, the dataset is from 2016/2017 and limited in the number of cells with respect to standard single-cell experiments. In summary, the additional benchmark dataset fails to be an ideal representative of a single-cell dataset that could be analyzed with the proposed tool.
3. The authors' rationale for employing cell lines or in-vitro systems and bulk RNA-seq as ground truth for their benchmark presents a fundamental contradiction. While they rightly assert the superior utility of single-cell approaches for resolving cellular heterogeneity, they simultaneously claim that this very heterogeneity precludes the testing of alternative splicing (AS) methods on real single-cell data. Consequently, they advocate for benchmarking on datasets amenable to bulk analysis, and indeed utilize bulk RNA-seq as the ground truth. This creates a paradox: a method specifically developed for analyzing AS in single-cell data is benchmarked exclusively against scenarios where its purported advantage (single-cell resolution) offers no real added value. This approach severely limits the generalizability and practical applicability of the developed method to genuine biological contexts, a critical concern also highlighted by Reviewer 3. To overcome this inherent limitation and truly validate the method's utility, a crucial next step would be to incorporate long-read single-cell RNA-seq data into the benchmark, leveraging the long-read detected isoforms as the definitive ground truth. This would directly assess the method's performance in the heterogeneous single-cell environment it aims to address.
4. In different scenarios, SCSES appears to exhibit similar performance to Psix. What specific advantages, if any, can be claimed when asserting that Psix also achieves comparable performance? I would suggest clarifying and strengthening the evidence showing that SCSES outperforms Psix. As it stands, the examples provided, including the real single-cell scenario, do not convincingly demonstrate a significant advantage, which in turn reduces the impact and justification for SCSES. For example, for the iPSC dataset, the author claims that "SCSES divided the motor neurons (MNs) into two more subclusters (MN-C1, MN-C2), a distinction that was not as clearly delineated by other methods or by gene expression values". However, in Figure 3c it is evident that Psix also identifies these two subclusters, yielding results that are comparable to those of SCSES (cluster MN-C2 can also be observed by SCASL).

(Remarks on code availability)

Please see the attached file to see issues about the code.

Reviewer #2

(Remarks to the Author)

The author has thoroughly addressed all of my questions and concerns, providing detailed explanations and clarifications where necessary. I am now satisfied with the revisions and there are no outstanding issues. Based on the improvements made, I recommend that this paper be accepted for publication.

(Remarks on code availability)

I have assessed the code accompanying the paper. The repository includes a comprehensive README.md file that outlines the installation process, required dependencies, and detailed instructions for running the main scripts. The code is well-structured and reasonably documented, making it accessible to users with a background in the relevant tools and

programming languages.

Reviewer #3

(Remarks to the Author)

The authors have addressed some of the original concerns raised in the initial review. However, some have not been adequately addressed. These points still require clarification.

1. Rationale for using cell line datasets

To validate splicing heterogeneity within a cell line, the authors projected cells into a UMAP space using raw PSI values from high-quality events (Response Figure 1). However, a seemingly random distribution in the UMAP plot does not necessarily imply a lack of heterogeneity, as the spatial distribution is highly influenced by parameters such as "n.neighbors" in the Seurat pipeline. A more direct measure of heterogeneity would be the distribution of PSI variances across cells for all events.

Additionally, the criteria used to define high-quality events are not fully described. For example:

- How was the minimum junction read count of 20 determined?
- What quantitative threshold defines a "huge" difference between upstream and downstream splice site coverage?
- Are the authors only examining exon skipping (SE) events, or are other splicing types included?

Besides, in Response Figure 2a, the meaning of "Raw" PSI is unclear. Does it refer to PSI values for individual cells, a pseudo-bulk sample, or averaged values across cells? This should be clarified in the text or figure legend.

2. Splicing events in cell cycle genes

The authors excluded splicing events in genes annotated with cell cycle-related functions in an attempt to remove potential confounding effects. However, this approach is problematic. Cell cycle-regulated splicing events are not necessarily limited to genes labeled as "cell cycle" genes in gene ontology. A more appropriate strategy would be to identify and exclude splicing events that show cell cycle-dependent regulation, for instance by analyzing bulk RNA-seq data from synchronized cells at different stages of the cell cycle.

3. Cell similarity based on RBP transcript expression

The authors state that "splicing-based similarities used in our initial submission, specifically similarities derived from gene expression of RBPs, have better performance compared to the whole-transcriptome or random transcriptomic similarities." However, Response Figure 15 suggests that the improvement is minimal (less than 0.05 difference in Spearman correlation coefficient), and in some cases—such as HL-60 cells—there is virtually no improvement. This raises the question of whether transcript-level RBP expression is truly informative for PSI imputation. Please clarify the rationale for relying on RBP expression and discuss the limitations or possible improvements in this approach.

4. Events detected in bulk but not in scRNA-seq data

The authors state that "some events may completely lose all supporting junction reads in individual cells, even if they can be detected in the pseudo-bulk sample." This explanation is not convincing. If an event is detected in a pseudo-bulk sample—which aggregates reads across all cells—one would expect that at least some individual cells contribute reads supporting that event. Please explain under what circumstances an event could appear in the pseudo-bulk but be entirely absent in all single cells.

In Response Figure 16, the distinction between panels (a) and (b) is unclear. Specifically, what does "in the entire scRNA-seq dataset" mean in this context? Please provide clarification.

5. Extended Data Figure 35 Legend

The legend for Extended Data Figure 35 incorrectly describes the content. It should refer to the restart probability parameter. Please revise the legend accordingly.

(Remarks on code availability)

Reviewer #4

(Remarks to the Author)

(Remarks on code availability)

Version 2:

Reviewer comments:

Reviewer #1

(Remarks to the Author)

We thank the authors for answering our previous points, in particular by considering two additional single-cell benchmarking datasets, one using long-read data and one using full-coverage short-read data as ground truth. These additions potentially provide stronger evidence of the tool's robustness and performance. However, to enable a complete understanding of the results of these benchmarks, it would be important to consider the following points:

1. The use of a pairwise design, in which SCSES is compared to only one tool in each round, represents a non-standard choice and makes the results more difficult to evaluate, particularly in the absence of information on how the AS events differed across each comparison. Moreover, the performances of SCSES change in each comparison, because a different set of AS events is considered each time. A solution could be to use the same set of events, detected by all tools, for the benchmark. Alternatively, the number of AS events detected by each method, along with the number of AS events present in the ground truth, should be included. For each comparison, the number of AS events considered and the number discarded because not detected by the tools should be provided. A discussion on the differences in AS detection observed between the methods with respect to the ground truth should also be included.

2. In the manuscript, it is stated: "The event counts for each comparison and each cell are listed in Supplementary Table 2." However, the table contains only about 200 rows, making it unclear whether the benchmark was conducted at the individual-cell level, as currently stated in the figure captions. At the cell level, there should be thousands of rows, corresponding to the number of cells of the datasets, not only 200. Furthermore, providing details on which genes and which specific events were included in the analysis would improve the clarity and completeness of the reported results.

(Remarks on code availability)

The GitHub page has been clarified and is now easier to follow without errors. However, I recommend adding a note for users who are not familiar with Docker: specifically, that the working directory must have appropriate write permissions for the Docker container to function correctly. In my case, I was working on a remote server without sudo access, and I needed to run `chmod -R 777` on the working directory in order for the container to write files properly.

Reviewer #3

(Remarks to the Author)

The authors have adequately addressed the concerns.

(Remarks on code availability)

Reviewer #4

(Remarks to the Author)

(Remarks on code availability)

Reviewer #5

(Remarks to the Author)

(Remarks on code availability)

Reviewer #6

(Remarks to the Author)

(Remarks on code availability)

Point-by-point response to Reviewers

We thank the reviewers for their constructive and detailed evaluation of our manuscript. Their insightful comments have helped to improve our work. We have addressed each point raised in our revised manuscript as follows:

For a better reading of our responses, we marked the words in different colors as follows:

Reviewer comments are marked in **black font**

Responses are marked in **blue font**

New descriptions added to the revised manuscript are marked in **red font**

The location of added descriptions in the revised manuscript is marked in **green font**

Reviewer #1 (Remarks to the Author):

The authors developed SCSES, an R package to identify and quantify alternative splicing events from single-cell RNA-seq data. To overcome the high dropout rate and improve alternative splicing profiles, SCSES imputes missing alternative splicing alterations via data diffusion, by learning information from similar cells or events. Detecting and analyzing alternative splicing at the single-cell level is undoubtedly a significant challenge. Current single-cell platforms are often hindered by technical limitations, making this task difficult to achieve. The authors attempt to address these limitations by employing data diffusion techniques. Yet, in the submitted manuscript, it is difficult to comprehend the significance of the tool's results and its overall contribution, due to the insufficient information and the omission of explanations for the decisions made during its development. The article lacks clarity, and no validation is provided to support the novel results obtained by SCSES.

REPLY: We thank the Reviewer for the careful consideration of our manuscript and thoughtful comments.

Major points:

1. In the abstract, the authors claim that “Systematic simulation studies demonstrate that SCSES outperforms existing algorithms”. In the manuscript, the benchmark is performed in cell lines or simulated data. Both are not real scenarios for single-cell datasets. Simulation is not sufficient alone as a benchmark. Benchmarks on real scenarios would prove the added value of SCSES with respect to other tools.

REPLY: We thank the Reviewer for this valuable comment. Below, we added an evaluation in real scenarios, and interpreted the rationale for using cell line datasets in our submission.

(1) Evaluation on real scenarios

To evaluate the performance of SCSES in real single-cell scenarios, we analyzed the iPSC dataset (GSE85908), which includes scRNA-seq data from 174 cells spanning three cell types (induced pluripotent stem cells/iPSCs, neural progenitor cells/NPCs, and motor neurons/MNs) along with matched bulk RNA-seq data. Following the same evaluation approach used for cell line data in the initial submission, we conducted pairwise comparison of SCSES to each of the other six methods, including BRIE1, BRIE2 aggr, Expedition, Psix, rMATS and SCASL. For each pair of comparison and cell type, we selected AS events detected by both two methods as the list of AS events to compare. The PSI values of this list of AS events derived from bulk data of matched cell type were served as biological truth. In this way, we calculated the Spearman correlation coefficient (SCC) between the predicted PSI values by each method and benchmark PSI values for each cell and also computed the root mean squared error (RMSE) between the predicted PSI values and benchmark PSI values for each valid event across all cells. As shown in **Extended Data Fig. 15-17**, except Psix, SCSES

exhibited over 30% higher median SCC values across cells by in all comparisons. Similarly, SCSES presented over 0.05 reduction in the median RMSE values in all comparisons, and had the similar performance with SCASL and Psix.

Extended Data Figure 15. Raincloud plots showing the performance comparison between SCSES-RC and other algorithms on iPSC, NPC, and MN cells from iPSC dataset (GSE85908), respectively. The AS events detected by both SCSES and the compared algorithm are considered for each comparison. The SCC value refers to the Spearman correlation coefficients between estimated PSI values and benchmark PSI values (derived from bulk RNA-seq of matched cell type) of all events in each cell. The RMSE is calculated between estimated PSI and benchmark PSI values of an event among all cells. The *P*-values are calculated by the Wilcoxon test. N.S.: *P*-value >0.05, *: *P*-value <0.05, **: *P*-value <0.01, ***: *P*-value <0.001.

Extended Data Figure 16. Raincloud plots showing the performance comparison between SCSES-PSI and other algorithms on iPSC, NPC, and MN cells from iPSC dataset (GSE85908), respectively. The AS events detected by both SCSES and the compared algorithm are considered for each comparison. The SCC value refers to the Spearman correlation coefficients between estimated PSI values and benchmark PSI values (derived from bulk RNA-seq of matched cell type) of all events in each cell. The RMSE is calculated between estimated PSI and benchmark PSI values of an event among all cells. The *P*-values are calculated by the Wilcoxon test. N.S.: *P*-value >0.05, *: *P*-value <0.05, **: *P*-value <0.01, ***: *P*-value <0.001.

Extended Data Figure 17. Raincloud plots showing the performance comparison between SCSES-RBP and other algorithms on iPSC, NPC, and MN cells from iPSC dataset (GSE85908), respectively. The AS events detected by both SCSES and the compared algorithm are considered for each comparison. The SCC value refers to the Spearman correlation coefficients between estimated PSI values and benchmark PSI values (derived from bulk RNA-seq of matched cell type) of all events in each cell. The RMSE is calculated between estimated PSI and benchmark PSI values of an event among all cells. The *P*-values are calculated by the Wilcoxon test. N.S.: *P*-value >0.05, *: *P*-value <0.05, **: *P*-value <0.01, ***: *P*-value <0.001.

In light of the reviewer's suggestion, we have added the above figures into the revised Extended Data file as **Extended Data Fig. 15-17**, and we have added the corresponding results in the revised manuscript:

"To assess the performance of SCSES in real single-cell scenarios, we analyzed the iPSC dataset (GSE85908), which includes scRNA-seq data from 174 cells across three distinct cell types (induced pluripotent stem cells/iPSCs, neural progenitor cells/NPCs, and motor neurons/MNs) along with matched bulk RNA-seq data of each cell type. Following the same evaluation approach used for cell line data, we found that with the exception of Psix, SCSES exhibited over 30% higher median SCC values across cells in all comparisons. Similarly, SCSES presented over 0.05 reduction in the median RMSE values in all comparisons, and performed similarly with SCASL and Psix (**Extended Data Fig. 15-17**)." (Results Section, Page 7, Line 169-177)

(2) Rationale for using cell line datasets in our initial submission

We would also like to explain the reason why we used cell line dataset for evaluation of the performance of SCSES in our initial submission. Due to the cell heterogeneity of real single-cell data and technical limitations of scRNA-seq data (high dropout, high noise, low coverage), it's a great challenge to obtain the accurate composition of cell types along with the exact splicing changes on each cell type from a real single-cell data for method benchmarking. Therefore, in the initial submission, single-cell RNA-seq datasets of cell lines were chosen for benchmarking due to their minimal cellular heterogeneity and the availability of accurate splicing characteristics obtained from the bulk data of matched cell type. Specifically:

a) Minimal cellular heterogeneity of scRNA-seq datasets of cell lines

Cell lines are derived from a single cell or a small group of cells, and are propagated under controlled laboratory conditions. Standardized culture conditions (e.g., growth medium, temperature, pH) further minimize variations in cellular behavior, resulting in a homogeneous cell population. This ensures a highly consistent genetic background across the population.

To validate the minimal splicing heterogeneity, we plotted UMAPs of four cell lines using raw PSI values of high-quality events. Splicing events were considered as high quality if they met the sequencing quality threshold in at least 5 cells. The quality thresholds included: (1) more than 20 junction reads to support either inclusion or exclusion exons; (2) when read counts of either upstream or downstream inclusion junction have fewer than 5 reads, the read count fold change between them should be less than 10. The second threshold is used to exclude events with a huge difference of reads depth between the upstream or downstream splice sites, which may lead to inaccurate estimation of PSI value. As shown in **Response Fig. 1**, cells of each cell line were randomly distributed without obvious clustering or trajectory, confirming the lack of significant heterogeneity.

Response Figure 1. UMAP plots based on the PSI values of high-quality AS events showing the lack of heterogeneity in scRNA-seq data of four cell lines. The event counts used for UMAP plots are 5218 for HCT116, 4377 for HepG2, 3653 for HCC 1954, and 1506 for HL-60.

b) Accurate quantification of splicing events from bulk RNA-seq of matched cell types for method benchmarking:

Accurate quantification of alternative splicing requires high depth of junction reads, which is more feasible with bulk RNA-seq data. However, bulk data from tissue often represent mixed cell types and states, making them unsuitable as benchmarks for splicing changes under specific cell types. Due to the minimal heterogeneity, bulk RNA-seq data from cell lines provides an ideal reference for quantification of splicing events, thus is suitable for method benchmarking of quantifying splicing events from the matched cell type on single-cell level.

To validate this, we calculated the PSI differences of high-quality event-cell pairs (meeting the quality thresholds as described above) between single-cell and matched bulk datasets. As shown in **Response Fig. 2a**, more than 82% of the pairs exhibited differences of less than 0.1. Additionally, the Pearson correlation coefficient (PCC) of PSI values between single cell and bulk samples exceeded 0.9 for more than 80% of cells (**Response Fig. 2b**). These results indicate that the splicing profiles from bulk RNA-seq data of cell lines are highly similar to those from matched scRNA-seq datasets, making them suitable benchmarks for evaluating AS quantification methods.

Response Figure 2. Density plots showing the high splicing consistency between bulk and single cell datasets of cell lines. **(a)** Distribution of PSI differences between bulk datasets (Ref) and corresponding event-cell pairs (Raw). **(b)** Distribution of Pearson correlation coefficients (PCC) between PSI values from bulk data and corresponding single cells.

c) Similar strategies adopted in previous studies:

The very similar strategy of using single-cell and bulk RNA-seq data from the same cell lines has been commonly used to evaluate the method performance of gene expression and alternative splicing estimation at the single-cell level. For example, Hou et al. systematically evaluated 18 scRNA-seq gene expression imputation algorithms by comparing imputed single-cell profiles with bulk RNA-seq profiles from the same cell lines, leveraging their less heterogeneous and well-defined gene expression as benchmarks³. Huang et al. used single-cell and bulk RNA-seq datasets of HCT116 to evaluate the accuracy of estimated PSI by BRIE⁴. Inspired by these studies and many others on the related research topic, we have decided to use single-cell RNA-seq of cell lines as the test sets for method comparisons, and bulk RNA-seq data of the same cell types were used to provide the background truth of the splicing changes for the performance evaluations in our initial submission.

2. A table describing and comparing all the existing single-cell methods mentioned in the introduction (BRIE/BRIE2, Psix, SCASL,...) with respect to SCES would be useful to understand the specificities of SCSES.

REPLY: We thank the Reviewer for this very useful comment. We have added a comparison table as **Supplementary Table 1** as below, and added the corresponding description in Introduction section.

“A detailed comparison of these methods is listed in **Supplementary Table 1.” (Introduction Section, Page 4, Line 69-70)**

Algorithm	Event Type	Basic Idea	Limitation	Reference
Expedition	SE, MXE	1. Merge valid junction reads, create AS annotation 2. Remove junction with non-canonical splice sites 3. PSI calculation based on inclusion/exclusion read counts	Only focus on high coverage events while events with low coverage may be inaccurate	Song Y, et al
BRIE	SE	Inferring the posterior of PSI from event sequence features and observed junction read counts by Bayesian regression and mixture model	Limited information from event sequence features	Huang Y, et al
BRIE2	SE	BRIE2 employs a Bayesian hierarchical model to predict PSI by incorporating cell-type/state features.	Mode 2-quant: imputation using all-cell aggregation; Mode 2-diff: prior cell features dependent, which may be unknown before analysis	Huang Y, et al
Psix	SE	Evaluate the PSI by incorporating both the observed PSI and information derived from similar cells with probabilistic model	Restricted to gene expression-based cell similarity, ignoring splicing heterogeneity.	NajrC F B A, et al
SCASL	-	Impute the dropout junctions with similar cells	Only focus on the imputation of missing values while overlooking uncertainty in low-coverage events	Xiang X, et al
SCSES	SE, MXE, A3SS, A5SS, R	1. Merge all cell reads to create a pseudo-bulk sample 2. Use rMATS, MAJIQ and IRFinder to identify AS events from pseudo-bulk sample 3. Impute PSI or junction read counts with similar cell or similar events according to event-cell pair scenario		-

Supplementary Table 1: Brief summary of current AS quantification algorithms. Each row provides an overview of an AS quantification method, including the algorithm name, detected event type, the main idea behind the algorithm, algorithm limitations, and the reference.

3. In the discussion and in results, authors state that SCSES can identify novel clusters with unique splicing patterns that could not be detected by single-cell gene expressions or current methods on splicing analysis Based only on the reported results and without any additional validation, this statement is not strongly supported. Experimental or orthogonal validation would be needed to support this statement and the usefulness of the tool.

REPLY: We thank the Reviewer for this valuable comment. Below, we provide orthogonal validation using independent datasets and demonstrate the clinical relevance of our identified novel clusters of multiple

myeloma (MM) cells with unique splicing patterns by SCSES (**Figure 4** in our initial submission).

In our initial submission, we applied SCSES to analyze the splicing patterns of cells from patient MM34, collected at both diagnosis and relapse following treatment with thalidomide and bortezomib (BTZ). Based on splicing profiles, we identified two subclusters in the diagnosis-stage cells (SC1 and SC2) and two in the relapse-stage cells (SC3 and SC4). Our analysis revealed a continuous cell evolution trajectory from SC1 to SC4. Further investigation showed that SC2 cells shared similar gene expression and splicing patterns with relapse-stage cells, suggesting that SC2 cells may have had BTZ-resistance potential at primary diagnosis.

To independently validate our findings in patient MM34, we analyzed an additional scRNA-seq data (GSE118900)⁵, which was collected from a primary MM patient (NDMM3). Based on imputed splicing profiles by SCSES, we identified two cell subgroups (SC1 and SC2) that were not detectable by conventional gene expression analysis (**Extended Data Fig. 25a**). Then, we identified the differentially spliced events (DSEs) between SC1 and SC2 ($\Delta\text{PSI}>0.1$, $\text{FDR}<0.05$) and found that 64 events were differentially spliced in both MM34 and NDMM3 samples (**Extended Data Fig. 25b**), 44 of which had a consistent splicing direction (P -value=0.035, Fisher's exact test). Interestingly, by using the splicing intensities of 44 DSEs, most SC1 and SC2 cells from the two patients were clustered together. More intriguingly, the partial of SC2 cells, but not SC1 cells from two patients were clustered together with SC3 and SC4 cells from the relapsed sample (**Extended Data Fig. 25c**). This observation supports our hypothesis in the initial submission that SC2 cells had BTZ-resistance potential at primary diagnosis. Additionally, the activity of MM34-SC2 marker genes estimated by UCell⁶ was significantly higher in NDMM3-SC2 cells than in NDMM3-SC1 cells (**Extended Data Fig. 25d**). These results confirmed that we have successfully recapitulated the existence of a novel cluster SC2 in an independent scRNA-seq data from a primary MM patient. This novel cluster in the primary tumor is potentially associated with tumor relapse upon the BTZ treatment, and could only be detected by SCSE rather than gene expression clustering.

To assess any clinical relevance of SC2 cells, we analyzed three independent bulk RNA-seq datasets (MMRF⁷: $N=566$, GSE9782⁸: $N=188$, and GSE24080⁹: $N=559$) of BTZ-treated cohorts. The activity of MM34-SC2 marker genes was calculated via UCell⁶ or GSVA¹⁰ for each patient, and patients were separated into two equal-sized groups based on the median value of SC2 marker genes' activity. The results showed that the patients with high activities had significantly shorter overall survival time (**Extended Data Fig. 25 e-g**) and shorter disease-free time (**Extended Data Fig. 25h**) (log-rank test, P -value < 0.0001). This implies that SC2 marker genes were associated with patient outcome. Furthermore, primary diagnosis samples from eventually relapsed patients exhibited higher SC2 marker activity and lower SC1 activity compared to non-relapsed patients, suggesting SC2-like cells are associated with higher risk of BTZ resistance and tumor relapse.

In light of the reviewer's suggestion, we have added the above analysis into revised Extended Data file (**Supplementary Note Section, Page 6, Line 110-143**). The **Extended Data Fig. 25** has been added into revised Extended Data file. In the revised manuscript, we have added the related description as:

“To validate the potential BTZ-resistance of SC2 cells, we also analyzed independent scRNA-seq data from a primary MM patient and confirmed that existence of SC2 cells. Interestingly, by analyzing three additional cohorts of BTZ-treated MM patients, we found that increased activity of SC2 marker genes was associated with worse patient overall and disease-free survival, and relapsed patients exhibited higher SC2 marker genes activity than non-relapsed group (**Extended Data Fig. 25, Supplementary Note**).” (**Results Section, Page 13, Line 322-327**)

Extended Data Figure 25. Orthogonal validation of splicing-based subclusters in multiple myeloma patients. (a) UMAP plots showing cell projections for NDMM3 cells based on gene expression profile (left) and splicing profile (right). Colors indicate the two sub-clusters by splicing profile. (b) Scatter plot showing the direction of commonly differentially spliced events (DSEs) in NDMM3 and MM34. Commonly differentially spliced events are highlighted by colors, with shapes indicating whether the corresponding genes were differentially expressed between SC1 and SC2 of MM34. (c) Unsupervised hierarchical clustering and heatmap of all cells from NDMM3 and MM4 by the 44 DSEs with the same splicing direction between SC1 to SC2 in (b). (d) Box plots comparing marker gene activities of MM34 SC1 and SC2 subclusters in subclusters identified in NDMM3 (Wilcoxon test, $N_{SC1}=20$, $N_{SC2}=39$). (e-g) Kaplan-Meier survival curves showing overall survival differences based on SC2 marker gene activities in MMRF (e), GSE9782 (f), and GSE24080 (g) datasets (log-rank test). (h) Kaplan-Meier survival curves showing disease-free survival differences based on SC2 marker gene activities in GSE24080 dataset (log-rank test). (i) Box plots comparing MM34-SC1 and MM34-SC2 marker gene activities in primary diagnosis samples, stratified by patient relapse status.

4. The authors state that cell splicing similarities can be built with three different types of matrices: RNA-binding protein expressions, RAW RC or RAW PSI matrices. These matrices should be introduced/described in the context of splicing. In particular, SCSES can use RNA-binding protein expressions to measure pairwise cell similarities. How is RNA-binding protein information obtained? Can it be used in multiple species? The use of this information and its relevance should be introduced/described/discussed. What are the advantages/disadvantages of using one matrix with respect to the other two? How to best select input data?

REPLY: We thank the Reviewer for this valuable comment. Below we describe the three cellular matrices and compare their respective advantages.

(1) Explanation of three cellular similarity matrices in SCSES

We provide a detailed explanation of these three matrices in the context of splicing. As shown in **Extended Data Fig. 1**, a splicing event involves inclusion junctions and exclusion junctions. Typically, inclusion junctions refer to transcripts including the alternative region, while exclusion junctions refer to transcripts excluding alternative region. To quantify the splicing intensity of a splicing event, PSI values are commonly used, which is defined as the proportion of inclusion junction reads and the total junction reads (inclusion + exclusion). Additionally, RNA-binding proteins (RBPs) are crucial regulators of RNA splicing process, which bind directly to specific RNA sequences and orchestrate various aspects of alternative splicing, such as splice site selection and spliceosome assembly. The gene expression patterns of RBPs can effectively reflect the state of cellular splicing regulation. Therefore, to measure the cellular splicing similarities, SCSES provides three options, including junction read counts, PSI values, and the expression of RBP genes. The junction read counts are generated by counting splice junction reads in each cell, and the PSI value of each event is calculated by corresponding junction read counts with the definition described in **Extended Data Fig. 1**. The RBP gene expression is extracted with featureCounts¹¹ algorithm from scRNA-seq data. The RBP genes of different species are various. In the current version of SCSES, RBP lists for both human and mouse are included, which are downloaded from public datasets. Specifically, the human RBP list is manually curated from RBPbase (<https://apps.embl.de/rbpbases/>) and a well-used resource¹², while the RBP list for mouse is collected from SFMetaDB¹³ and RBPDB¹⁴. For other species, SCSES supports user-defined RBP list.

Extended Data Figure 1: The definition of PSI of different AS events. IJC1/IJC2 represents the inclusion junction read counts, while the EJC1/EJC2 represents the exclusion junction read counts. For RI events, the IJC1 and IJC2 are defined as the reads spanning 5nt upstream to downstream of the cryptic 3'/5' splicing site, respectively. "A" represents the alternative splicing region. "C1/C2" represents the constitutive exon.

(2) Recommendation for cellular similarity matrix selection

Benefiting from the higher coverage in RBP genes than junctions, the cellular similarities based on RBP expression are fitted in most cases. However, RBP expressions cannot comprehensively represent cellular splicing profiles. RNA splicing is a complex process, where RBP expressions are only one of several regulatory factors¹⁵. As a result, junction read counts and PSI values depict cell splicing states more directly. Comparing between junction read counts and PSI values, PSI values are the downstream product of junction read counts and may be affected by the quality of junction read counts.

Regarding the strategy for feature selection, the evaluation in the initial submission indicate that the performances of the three similarity measures are highly dataset-dependent. We recommend that users test and compare the results obtained from different similarity measures to determine the most suitable approach for their specific dataset. Based on the above analysis of advantages and disadvantages, we suggest a prioritized ranking of the measures as follows: RBP expression, junction read counts, and PSI. This recommendation is particularly valuable for datasets with lower coverage, where the expression of RBP genes may provide more robust and reliable insights into splicing regulation.

In light of reviewer's suggestion, we have revised the manuscript and Extended Data file as follows:

"RNA-binding proteins are critical regulators of RNA splicing process, and the gene expression of RBPs can reflect the cell splicing states. Moreover, the junction read counts of events, as well as PSI values, can directly represent cellular splicing intensity. Hence, SCSES uses these three features to quantify cell-cell splicing

similarities.” (Methods Section, Page 20, Line 526-529)

“The RBP list of human was manually curated from RBPbase (<https://apps.embl.de/rbpbases/>) and a well-used resource¹⁰², while the RBPs for mouse were collected from SFMetaDB¹⁰³ and RBPDB¹⁰⁴. SCSES supports user-defined RBP list for different species.” (Methods Section, Page 21, Line 554-557)

“Benefiting from the higher coverage in RBP genes than junctions, the cellular similarities based on RBP expression are fitted in most cases. However, RBP expressions cannot comprehensively represent cellular splicing profiles. RNA splicing is a complex process, where RBP expressions are only one of several regulatory factors. As a result, junction read counts and PSI values depict cell splicing states more directly. Comparing between junction read counts and PSI values, PSI values are the downstream product of junction read counts and may be affected by the quality of junction read counts.

Regarding the strategy for feature selection, the evaluation in the manuscript indicates that the performances of the three similarity measures are highly dataset-dependent. We recommend that users test and compare the results obtained from different similarity measures to determine the most suitable approach for their specific dataset. Based on the above analysis of advantages and disadvantages, we suggest a prioritized ranking of the measures as follows: RBP expression, junction read counts, and PSI. This recommendation is particularly valuable for datasets with lower coverage, where the expression of RBP genes may provide more robust and reliable insights into splicing regulation.” (Extended Data file, Page 9, Line 214-231)

5. The authors say “due to the high dropout rate and technical limitations, such matrices are very sparse with limited read counts, leading to inaccurate PSI estimation.” Is there data or evidence supporting this? Can this sparsity be quantified for AS types?

REPLY: We thank the Reviewer for this valuable comment. The following section provides data and evidence to support our statement that “due to the high dropout rate and technical limitations, such matrices are very sparse with limited read counts, leading to inaccurate PSI estimation”.

To quantify the sparsity of splicing events, we analyzed scRNA-seq data from four cell lines (HCC1954, HCT116, HepG2, and HL-60). Using SCSES, we detected 29,734 alternative splicing events across five types (A3SS, A5SS, RI, MXE, and SE), each of which was supported by at least 5 junction reads in at least 25 cells. Using genes that met the same quality criteria as a reference, we compared the sparsity between the total read count matrix of splicing events and genes.

We calculated the dropout rate of each cell, defined as the proportion of splicing events or genes without any supporting read. As shown in **Extended Data Fig. 3a**, the dropout rate for splicing events was significantly higher than that for gene expression. For example, in HCT116 dataset, the dropout rate for splicing events was 1.91–2.32 times higher than that for gene expression. Furthermore, for non-dropout splicing events, the read counts were significantly lower than those for non-dropout genes within each cell. As shown in **Extended Data Fig. 3b**, the median total read counts for genes were at least 1.3 times higher than that for splicing events. These results demonstrated high sparsity and limited read counts of splicing events.

To evaluate the effect of total read count of splicing events on raw PSI accuracy, we used the PSI values from matched bulk RNA-seq dataset as benchmarks, and categorized splicing events into three groups based on their PSI values in the bulk datasets of four cell lines, including:

- **High Group:** defined as events with $PSI > 0.8$ in all four cell lines.
- **Low Group:** defined as events with $PSI < 0.2$ in all four cell lines.
- **Normal Group:** defined as events that are not classified in either the High or Low Group.

We calculated the PSI differences between raw single-cell PSI values and the benchmark PSI values, then grouped the cell-event pairs based on their total read counts. As shown in **Extended Data Fig. 4**, splicing

events with lower read counts exhibited higher PSI estimation errors across all three event groups, demonstrating the impact of data sparsity on PSI estimation accuracy. For example, for *UBTF* exon 8 skipping, the single-cell data captured only 7 junction reads in the splicing region with no reads supporting the skipping event. Similarly, for *CRK* alternative 5' splice site, insufficient read coverage resulted in the absence of junction reads supporting the upstream splice site. These read coverage limitations led to substantial PSI estimation errors of 0.52 and 0.61 for *UBTF* and *CRK* events, respectively (**Response Fig. 3**).

These findings are further supported by evidence from Buen Abad Najjar et al.¹⁶, which demonstrated that technical limitations significantly impact splicing measurements in single cells by simulation studies.

Response Figure 3. Read coverage showing two splicing events in HCT116: *UBTF* exon 8 skipping and *CRK* exon 2 alternative 5' splice site usage, compared between bulk RNA-seq and scRNA-seq data.

In light of Review's suggestion, we have added **Extended Data Fig. 3** and **Extended Data Fig. 4** into the revised Extended Data file, and revised the manuscript as follows:

“Due to the high dropout rate and technical limitations, such matrices are very sparse with limited read counts, leading to inaccurate PSI estimation (**Extended Data Fig. 3-4**).” (Results Section, Page 5, Line 94-95)

Extended Data Figure 3. Analysis of splicing event sparsity. (a) Box plots showing the distribution of dropout rate per cell across different alternative splicing event types (column 1-5) and gene (column 6). The dropout rate in each cell is defined as the proportion of features (splicing events or genes) without any supporting read. The distribution of dropout gene proportions is shown as a reference. (b) Violin plots showing the distribution of total read counts per cell across different alternative splicing event types (column 1-5) and gene (column 6), only considering non-dropout features (splicing events or genes). The distribution of gene expression read counts is shown as a reference.

Extended Data Figure 4. Analysis of impact of splicing event sparsity on PSI estimation accuracy.

Box plots showing the distribution of PSI differences between raw single-cell data and corresponding benchmarks for each splicing event, grouped by total read counts. (a) High-PSI event group (defined as events with PSI > 0.8 in all four cell lines); (b) Low-PSI event group (defined as events with PSI < 0.2 in all four cell lines); (c) Normal-PSI event group (defined as events that are not classified in either the High or Low Group).

6. SCSES uses three different strategies to estimate the final PSI value. The rationale for the choices is not described.

Reply: We thank the Reviewer for this valuable comment. In SCSES, we use three imputation strategies for four types of biological scenarios of event-cell pairs. The strategies choices for different scenarios were determined based on our experimental results on cell line datasets. As shown in **Extended Data Fig. 7**, compared to other strategies, PSI values estimated by Strategy 1 in single-cell dataset showed higher Spearman correlation coefficients (SCC) with reference PSI values derived from matched bulk data in ND (non-dropout) group, which suggests that Strategy 1 is more suitable for ND group pairs. In the WD (with-dropout) group, we evaluated the percentage of true-positive event-cell pairs (where Δ PSI between reference and imputed PSI was <0.1), because the real PSI values for BD group are either zero or one. The results demonstrated Strategy 2's superior performance for both BD (biological dropout) and TD (technical dropout) groups (**Extended Data Fig. 7**). Furthermore, we observed that some event-cell pairs could not obtain information from neighbor cells when those neighbors also belonged to the TD group. We therefore tested Strategy 3's performance separately in TD+Info and TD-Info groups. The results showed Strategy 2 achieved minimal RMSE in TD+Info cases, while Strategy 3 performed better in TD-Info cases. Hence, Strategy 3 is used to PSI imputation in TD-Info group pairs. To validate the robustness of these scenario-dependent patterns, we replicated the analysis in down-sampled datasets of iPSC, hEE and nPSC, using raw PSI values as reference. Similar patterns can be found in these datasets (**Extended Data Fig. 8**).

Biologically, this observation makes sense: ND events have detectable reads supporting both inclusion and exclusion junctions, thereby ensuring the inherent reliability of their PSI values. In contrast, WD events suffer from uncertainty about whether missing reads reflect true biological absence or technical limitations, making junction read imputation (Strategy 2) a more robust approach. Event similarity (Strategy 3) provides supplementary information only when neighboring cells are unable to provide sufficient read coverage for reliable inference.

In light of the reviewer's suggestion, we have added the description about rationale for strategy choices and **Extended Data Fig. 7-8** into the revise Extended Data file (Supplementary Note Section, Page 10, Line 233-259) and revised the manuscripts as follows:

“Based on our practice, we recommend three different data imputation strategies, matched to four types of biological scenarios, which are defined by the abundance of alternative junctions in both the target cell and its neighboring cells (**Fig. 1c, Extended Data Fig. 6, Supplementary Note**)” (Results Section, Page 5, Line 112-115)

“For ND cases, which have reads supporting alternative junctions, we suggest imputing the PSI value directly using a cell similarity network (**Extended Data Fig. 7-8**).” (Results Section, Page 5, Line 117-118)

“And specifically, TD-Info requires an additional round of data diffusion using the event similarity network on the PSI matrix derived from previous round of imputation (**Methods, Extended Data Fig. 7-8**).” (Results Section, Page 6, Line 129-131)

Extended Data Figure 7. Performance of different imputation strategies for different event-pair scenarios with cell similarities computed by RBP (a), PSI (b), RC (c). SCC represents the SCC between imputed PSI and benchmark PSI in cells. TP means the event-cell pairs, whose absolute differences between imputed PSI and benchmark PSI is less 0.1. The TP percentage indicates the proportion of TP pairs among BD or TD pairs. RMSE shows the root mean square error between imputed PSI and benchmark PSI in cells. Colors represent different imputation strategies. ND: non-dropout group, BD: biological dropout group, TD: technical dropout group, TD+info: technical dropout group that could obtain information from neighbor cells, TD-info: technical dropout group that could not obtain information from neighbor cells.

Extended Data Figure 8. Performance of different imputation strategies for different event-pair scenarios with cell similarities computed by RBP in iPSC, hEE, and nPSC down-sampling datasets. SCC represents the SCC between imputed PSI and benchmark PSI in cells. TP means the event-cell pairs, whose absolute differences between imputed PSI and benchmark PSI is less 0.1. The TP percentage indicates the proportion of TP pairs among BD or TD pairs. RMSE shows the root mean square error between imputed PSI and benchmark PSI in cells. Colors represent different imputation strategies. The red arrow indicates the strategy used in SCSES. ND: non-dropout group, BD: biological dropout group, TD: technical dropout group, TD+info: technical dropout group that could obtain information from neighbor cells, TD-info: technical dropout group that could not obtain information from neighbor cells.

7. What is the biological relevance of identifying two motor neuron subclusters? Additionally, the fact that differentially expressed genes, associated with mRNA splicing and neuron development, were identified between C1 and C2 cells seems to contradict the initial assumption that AS tells a different story with respect to gene expression. If these clusters are different in terms of gene expression, what is the added value of AS?

Reply: We thank the Reviewer for this valuable comment. Below, we discussed the biological relevance of identifying two motor neuron subclusters (MN-C1 and MN-C2) and clarified the added value of alternative splicing (AS) analysis in this context.

The two subclusters of motor neurons were identified unsupervised based on SCSES-estimated splicing profile (**Response Fig. 4a**), while the difference of overall gene expression profiles between these subclusters was not as large as that of splicing profile (**Response Fig. 4b**). To investigate the biological meaning of these two subclusters, we performed a supervised analysis between them on purpose and identified the differentially expressed genes (DEGs) between MN-C1 and MN-C2. This supervised comparison is performed by given the cell identities of the two groups, which however, do not necessary means that the overall gene expression profile could naturally distinguish them. Our results indicated that these DEGs were associated with mRNA splicing and neuron development. UMAP plot showed that MN-C1 cells located closer to neural progenitor cells (NPCs) and farther from MN-C2 cells (**Response Fig. 4a**). The motor neurons in this study were induced from iPSCs, and recent studies have shown that NPCs represent an intermediate phase during iPSC differentiation into motor neurons¹⁷⁻¹⁹. This suggested that MN-C1 cells may represent an earlier stage of motor neuron maturation. To validate this hypothesis, we used CytoTrace²⁰ and Monocle²¹ to assess their differentiation state. The results showed that MN-C1 cells had greater developmental potential and earlier pseudotime compared to MN-C2 cells (**Response Fig. 5**), further supporting the view that MN-C1 cells correspond to an earlier stage of motor neuron maturation.

Response Figure 4. (a). UMAP plots of iPSC dataset by PSI values estimated by SCSES **(b).** UMAP plots of iPSC dataset by gene expression. The colors indicate the cell clusters.

Response Figure 5. (a). Boxplot showing the pseudotime distribution of different cell groups estimated by Monocle algorithm based on expression. (b). Boxplot showing the distribution of stemness scores of in different cell groups.

Moreover, multiple genes exhibited changes in splicing patterns without corresponding changes in expression levels (**Response Fig. 6a, b**), where some genes were associated with neuron differentiation. For example, expression levels of *DDX6* did not exhibit significant difference between MN-C1 and MN-C2 cells (P -value=0.79, Log_2FC =-0.23, **Response Fig. 6c**), while MN-C1 cells expressed more isoform including 6th exon (**Response Fig. 6d, e**). *DDX6* is reported to be an essential factor for neuron differentiation²². These findings highlight the ability of AS to uncover regulatory mechanisms that are not detectable at the gene expression level.

Response Figure 6. (a). Heatmap presents the PSI profiles of 191 differential splicing events between MN-C1 and MN-C2 groups without differential expression of their target genes. (b). Heatmap presents the expression profiles of genes from (a). (c). Box plot showing the gene expression of *DDX6* between MN-C1 and MN-C2. (d). Box plot showing the PSI of *DDX6* between MN-C1 and MN-C2. (e). Read coverage showing the inclusion of *DDX6* exon 6 on the pseudobulk of MN-C1 and MN-C2. Alternative exons are highlighted in red.

8. SCSES uses three algorithms to detect different AS types: rMATS, MAJIQ and IRFinder. What is the rationale for this choice? Does the tool require a GTF?

Reply: We thank the Reviewer for this valuable comment. Due to low coverage and high noise of scRNA-seq data, it's a great challenge to identify splicing events in individual cells. In SCSES, to identify potential splicing events, we perform classical event detection algorithms in a pseudo bulk sample, which is created by merging total reads of all cells. In bulk datasets, different algorithms are suitable in detecting different splicing event types. rMATS is an algorithm based on MATS²³, which is developed to detect exon skipping events. MAJIQ detects events by identifying alternative 3'/5' splicing site, which specialize in identifying A3SS and A5SS events. IRFinder is developed to identify intron-retention events. Furthermore, these algorithms are all widely used in AS event detection, as indicated by their citation numbers on Google Scholar: 2304 for rMATS, 487 for MAJIQ, and 267 for IRFinder. Hence, we selected these three algorithms to identify different types of splicing events in SCSES.

All these algorithms need gene annotation files, where rMATS and IRFinder require gene annotation in GTF format, and MAJIQ require gene annotation in GFF format.

Finally, we want to note that SCSES is an open framework in detecting splicing events. Users can select their preferred algorithms to detect splicing events and provide an event list in a SCSES-defined format. With the event structure described in **Extended Data Fig. 1**, the event format can be described in Response Table 1 below:

Extended Data Figure 1: The definition of PSI of different AS events. IJC1/IJC2 represents the inclusion junction read counts, while the EJC1/EJC2 represents the exclusion junction read counts. For RI events, the IJC1 and IJC2 are defined as the reads spanning 5nt upstream to downstream of the cryptic 3'/5' splicing site, respectively. "A" represents the alternative splicing region. "C1/C2" represents the constitutive exon.

Event type	Format
A3SS	isoform1=exon:[C1 coord]@junction:[EJ1 coord]@[C2 coord] isoform2=exon:[C1 coord]@junction:[IJ1 coord]@exon:[A coord] [gene name] A3SS
A5SS	isoform1=exon:[C1 coord]@junction:[EJ1 coord]@[C2 coord] isoform2=exon:[A coord]@junction:[IJ1 coord]@exon:[C2 coord] [gene name] A5SS
SE	isoform1=exon:[C1 coord]@junction:[EJ1 coord]@exon:[C2 coord] isoform2=junction:[IJ1 coord]@exon:[A coord]@junction:[IJ2 coord] [gene name] SE
MXE	isoform1=exon:[C1 coord]@junction:[EJ1 coord]@exon:[A1 coord]@junction:[EJ2 coord] Isoform2=exon:[C1 coord]@junction:[IJ1 coord]@exon:[A2 coord]@junction:[IJ2 coord] [gene name] MXE
RI	isoform1=exon:[C1 coord]@junction:[EJ1 coord]@exon:[C2 coord] isoform2=retention:[IJ1 coord]@retentio:[IJ2 coord] [gene name] RI

Response Table 1. Format used by SCSES for naming splicing events. Each row defines the naming format of a splicing event type used in SCSES. A splicing event is described using four components, separated by the "|" symbol: **[isoform 1] | [isoform 2] | [gene name] | [event type]**. [isoform 1] and [isoform 2] represent the local exon-junction structures of the inclusion and exclusion isoforms, respectively. Local exon-junction structure consists of genomic coordinates for exons and splice junctions, separated by "@". IJ1/IJ2 represents the inclusion junction, while the EJ1/EJ2 represents the exclusion junction. Genomic coordinates (coord) follow the format: **[chromosome]:[start base]-[end base]:[strand]**.

9. The article could benefit from additional context to make it more comprehensive and easier to understand also for non-specialist readers. For example, in the abstract, the authors mention that SCSES “imputes the junction count matrix and fills in missing AS alterations”, but these concepts might be difficult to understand for non-specialist readers.

Reply: We thank the Reviewer for this valuable comment. In light of the reviewer’s suggestion, we have modified the description in the Abstract Section as follows:

“To address this, we developed SCSES (Single-Cell Splicing ESTimation), a computational framework designed to enhance the AS profiles. SCSES infers and completes the missing splicing changes by sharing information across similar cells and events with data diffusion.” (Abstract Section, Page 2, Line 20-23)

10. In all PSI-based UMAPs, the cells are organized into very well-defined trajectories. Could this be an effect of diffusion, which organizes everything into trajectories? A negative example would be useful (for instance, cell lines, where no trajectories should be present).

Reply: We thank the Reviewer for this valuable comment. To address this concern, we performed additional analysis using scRNA-seq data from four different cell lines, where developmental trajectories are not expected to be present. We analyzed the PSI values both before and after imputation using the top 20% highly variable splicing events and consistent UMAP parameters. The UMAP visualizations of these cell lines showed no apparent trajectory-like structures either before or after imputation, which serves as a negative control for our analysis (**Response Fig. 7**).

Response Figure 7. (a). UMAP plots by SCSES-RBP imputed PSI profiles of four cell line datasets (HCC1954, HCT116, HepG2, and HL-60). (b). UMAP plots by raw PSI profiles of four cell line datasets (HCC1954, HCT116, HepG2, and HL-60).

11. How single-cell data obtained from different samples were analyzed together? From the methods, it seems that integration was not performed. Could this introduce a batch effect? In all UMAP visualizations cells cluster together based on batches rather than biological differences.

Reply: We thank the reviewer for this valuable comment. In our initial submission, only one analysis may potentially involve with batch effects. This analysis combined two single-cell RNA-seq datasets including four cell lines (**Fig 2f** in the initial submission). All the other cells with UMAP visualizations were collected from the same dataset or from a single study, which indicated no technical batch effect. Therefore, in the initial submission, no additional procedure was conducted for batch correction.

To address the reviewer's concern about potential batch effects in our analysis of cell line datasets (**Fig. 2f** in the initial submission), we adopted an approach²⁴ for batch correction. We firstly validate its validity by using two independent datasets where batch effects exist, and then apply this method onto our own case of cell lines datasets to test whether batch effects exist or not. More specifically, to remove the batch effect of estimated splicing profile, we first applied the "ComBat" function from the R package "sva" version 3.46.0 to the original junction read count matrix, and used corrected junction read counts for SCSES estimation.

To validate the effectiveness of this approach, we introduced a new HCT116 single-cell dataset (GSE51254), and combined it with GSE150993 dataset used in the initial submission, which contained HCT116 cells and HepG2 cells. We checked the existence of batch effect on HCT116 cells between the two datasets by UMAP plot. As shown in **Response Fig. 8a**, an obvious batch effect was observed, with HCT116 cells from different datasets forming separate clusters. Then, we estimated the PSI values by batch-corrected junction read counts using "ComBat" in the combined dataset. The UMAP plot showed the HCT116 cells from different datasets were clustered together (**Response Fig. 8b**), demonstrating the effectiveness of "ComBat" on removing batch differences of splicing profile.

Finally, we used this batch correction approach on our analysis of four cell lines (**Fig. 2f** in the initial submission). The results showed no significant differences in UMAP plots before and after batch correction by "ComBat", indicating that batch effects were minimal in our cases.

Response Figure 8. UMAP plots showing the cell clusters before (a) and after (b) PSI batch correction in the new and existing HCT116 dataset. UMAP plots showing the cell clusters before (c) and after (d) PSI batch correction in the cell line dataset used in our initial submission.

Minor points:

1. SUPPA and SUPPA2 are only mentioned in the methods section without any references and are absent from the benchmarking analysis. Including these tools in the comparison with SCSES would provide a more comprehensive evaluation.

Reply: We apologize for the careless absence of the reference. We have added the reference of SUPPA in the revised manuscript as follows:

“In the synthetic dataset, which had known SE events (detected by SUPPA¹¹²) as ground truths (**Supplementary Note**), the union of events detected by different methods are used to calculate the accuracy score, which is defined to assessing the accuracy of event identification and PSI estimation.” (Methods Section, Page 27-28, Line 720-723).

In our evaluation, SUPPA/SUPPA2 was not included in the comparison for two main reasons. First, SUPPA/SUPPA2 estimates PSI values based on isoform-level expression, meaning that its accuracy is heavily dependent on the performance of upstream isoform quantification algorithms. In contrast, all algorithms evaluated in our initial submission calculate PSI values directly from alignment data (BAM files), independent of additional quantification tools. Therefore, comparing SUPPA/SUPPA2 with these algorithms would not be appropriate. Second, we evaluated the power of events identification across different algorithms with synthetic datasets. SUPPA identifies splicing events solely from gene annotation files, without requiring sequencing reads. This characteristic makes SUPPA more suitable as a third-party tool for generating benchmark event pools. For these reasons, we did not include SUPPA/SUPPA2 in the comparative analysis.

2. Is not clear what the meaning of AS intensity is.

Reply: We thank the Reviewer for this valuable comment. The AS intensity represents the splicing level of a splicing event measured by percent splice-in (PSI) value, which is defined as the proportion of inclusion junction reads and the total junction reads (inclusion + exclusion). In light of the reviewer’s suggestion, we have revised manuscript as follows:

“Through systematic simulation studies, SCSES outperforms existing algorithms in recovering percent spliced-in (PSI) values and diversity across cell populations.” (Abstract Section, Page 2, Line 23-25)

3. In the introduction, the authors say: “Lastly, due to the limited junction read counts, the calculated percent splice-in (PSI) values are usually not reliable, which makes the results of Expedition and SCASL incorrect.”

Is there evidence or literature supporting this statement?

Reply: We thank the Reviewer for this valuable comment. Below, we provide evidence and analysis to support our statement that “Lastly, due to the limited junction read counts, the calculated percent splice-in (PSI) values are usually not reliable, which makes the results of Expedition and SCASL incorrect”.

(1). Evidence for Expedition

To assess the relationship between junction read counts and the accuracy of PSI estimation by Expedition, we applied the tool to scRNA-seq datasets from four cell lines using default parameters, treating all NA values as zeros. PSI values derived from matched bulk RNA-seq data served as the benchmark. We calculated the absolute PSI differences between the benchmark and Expedition-estimated PSI values. Each event-cell pair was stratified into six groups based on total junction read counts, and the PSI differences were binned into six intervals ranging from 0 to 1. As shown in **Extended Data Fig 2a, 2b**, the proportion of event-cell pairs with lower PSI differences (e.g., 0–0.1) increased as the total junction read counts increased, while the proportion of event-cell pairs with higher PSI differences (e.g., 0.9–1) decreased. Based on the algorithm’s design, Expedition only processes splicing events with adequate read support for both inclusion and exclusion junctions, and directly calculates PSI values without performing imputation or correction. As discussed in Major Point 5 of Reviewer 1, the accuracy of raw PSI values is strongly influenced by the total number of junction reads. These results highlight that the performance of Expedition is highly dependent on junction read counts.

Extended Data Figure 2. Bar plots showing the distribution of absolute difference between the benchmark and the estimated splicing levels for Expedition (**a**, **b**) and SCASL (**c**) across four cell lines. The splicing levels in matched bulk data were calculated as benchmark splicing levels. Each event-cell pair was stratified into six groups based on total junction read counts (x-axis), and the PSI differences were binned into six intervals ranging from 0 to 1 (y-axis). Points within each bar indicate the count of event-cell pairs. (**a**) SE events analyzed by Expedition; (**b**) MXE events analyzed by Expedition; (**c**) All events analyzed by SCASL.

(2). Evidence for SCASL

To assess the relationship between junction read counts and the accuracy of splicing level estimation by SCASL, we applied the tool to scRNA-seq datasets from four cell lines using default parameters. We calculated the benchmark splicing levels in matched bulk data. The absolute difference between the benchmark and the SCASL-estimated splicing levels were calculated. Each event-cell pair was stratified into six groups based on total junction read counts, and the differences of splicing levels were binned into six intervals ranging from 0 to 1. As shown in **Extended Data Fig 2c**, the proportion of event-cell pairs with lower splicing level differences (e.g., 0–0.1) increased as the total junction read counts increased, while the proportion of event-cell pairs with larger errors (e.g., 0.9–1) decreased. Based on the algorithm's design, SCASL only imputes junctions that are completely missing, while ignoring those with low read counts, which also exhibit large estimation errors. As a result, its accuracy is highly sensitive to the total junction read counts of the events.

Our analyses demonstrated that the accuracy of both Expedition and SCASL is significantly influenced by junction read counts. Low read counts lead to unreliable splicing level estimates, which is consistent with our statement in the introduction.

In light of the reviewer's comments, we have added the **Extended Data Fig 2** into the revised Extended Data file and revised the manuscript as follows:

“Thirdly, the limited junction read counts often result in unreliable estimation of percent splice-in (PSI) values leading to inaccurate quantification in methods such as Expedition and SCASL (**Extended Data Fig. 2**).” (Introduction Section, Page 4, Line 65-67).

4. Splicing event types should be explained in the introduction.

Reply: We thank the Reviewer for this valuable comment. We have added the description of splicing event types in the introduction section and revised the manuscript as follows:

“AS events can be classified into five classical types: exon-skipping events (SE), alternative 3' splicing site events (A3SS), alternative 5' splicing site events (A5SS), retention intron events (RI), and mutually exclusive exons events (MXE)¹⁰ (**Extended Data Fig. 1**)” (Introduction Section, Page 3, Line 37-40)

5. What are “event sequence similarities”? Are both exons and introns included in this measure?

Reply: We thank the Reviewer for this valuable comment. Event sequence similarity is based on the event sequence features of both introns and exons. Event sequence features for SE events are collected from BRIE⁴, and similar features for other event types are designed by imitating the characteristics of SE events. Specifically, for SE events, the sequence features contain four categories: length, motif, conservation and k-mer. As shown in **Extended Data Fig. 40**, an SE event is associated with five regions: the first exon (C1), the first intron (I1), the alternative exon (A), the last intron (I2), and the last exon (C2). The length features include the length of region C1, C2 and A, as well as the length ratio of A/C1, A/C2 and C1/C2. The motif features include the scores of average position weight matrix (PWM) from regions near 3'/5' splicing site of I1 and I2. The nearby regions are defined between 16nt upstream to 4nt downstream for 3' splicing site, and

between 4nt upstream to 6nt downstream for 5' splicing site. The conservation features include the conservation scores of region C1, A, and C2, as well as the regions near 3'/5' splicing site of region I1 and I2, calculated by phastCons. The k-mer features include the 1-3mers of region C1, C2, nearby regions of I1 3' splicing site and I2 5' splicing site, 1-2mers of nearby region of I1 5' splicing site and I2 3' splicing site, and 1-4mers of region A. The features of an MXE event are the combination features of two related SE events (**Extended Data Fig. 40**). Features for A3SS, A5SS and RI events are defined similarly (**Supplementary Table 9, Extended Data Fig. 40**). Furthermore, we define an adenine ratio feature to account for alternative branch point selection for A3SS, A5SS, and RI events. Specifically, for an alternative region in an event (**Extended Data Fig. 40**), we divide the region into 100 bins with equal length, and calculate the cumulative adenine ratio for each bin as the adenine ratio features.

To compute the event sequence similarity, SCSES trains an auto-encoder model with event sequence features for each event type, and obtains the latent embedding matrix. Then the embedding matrix are used to calculate pair-wised normalized Euclidean distance of events.

All description above were contained in the Methods Section and Extended Data file.

Extended Data Figure 40: Region definition in different AS types. I1-5' and I1-3' indicates the 5' and 3' splicing site of intron 1, respectively. I2-5' and I2-3' indicates the 5' and 3' splicing site of intron 2, respectively. C1 and C2 represent the two exon regions, and A represent the alternative region.

Event Types	Category	Description
A3SS/A5SS	Length	Length of region C1, C2, A, and I
		length ratio of A/C1, A/C2, A/I
	Motif	Regions near 3'/5' splicing site of region A and I, 16nt upstream to 4nt downstream for 3' region, 4nt upstream to 6nt downstream for 5' region
	Conservation	Conservation scores of regions near 3'/5' of region A and I
	k-mer	1-3mer for region C1, C2, and regions near 3'/5' splicing site 1-4mer for region A
	Adenine ratio	Adenine ratio in region A and I
RI	Length	Length of region C1, C2, and A
		length ratio of A/C1, A/C2
	Motif	Regions near 3'/5' splicing site of region A, 16nt upstream to 4nt downstream for 3' region, 4nt upstream to 6nt downstream for 5' region
	Conservation	Conservation scores of regions near 3'/5' of region A
	k-mer	1-4mer for region C1, C2 and A
	Adenine ratio	Adenine ratio in region A

Supplementary Table 9: Event features of A3SS, A5SS and RI events to calculate event similarity network

6. In the introduction, the authors categorized cells according to dropout events in alternative splicing junctions (“with dropout”, “biological dropout” or “technical dropout”, “non dropout”, etc.). Is there a visualization that helps to understand this classification? The text could improve in clarity.

Reply: We thank the Reviewer for this valuable comment. In light of the reviewer’s suggestion, we have added the hierarchical structure of cell-event categories as **Extended Data Fig. 6** in revised Extended Data file and revised the manuscripts as follows:

“Based on our practice, we recommend three different data imputation strategies, matched to four types of biological scenarios, which are defined by the abundance of alternative junctions in both the target cell and its neighboring cells (**Fig. 1c, Extended Data Fig. 6, Supplementary Note**). Here, for a given AS event, cells with dropout in alternative splicing junctions are defined as WD (with dropout), while others as ND (non dropout). For ND cases, which have reads supporting alternative junctions, we suggest imputing the PSI value directly using a cell similarity network (**Extended Data Fig. 7-8**). WD cases are further divided into BD (biological dropout) or TD (technical dropout) based on their read count patterns. In BD cases, both the target cell and its neighbors show abundant reads for one junction type (either inclusion or exclusion) but complete absence of reads for the other type, indicating that only a single isoform is expressed. TD cases, however, are characterized by low read depth in both harboring gene and splicing event in the target cell, likely due to technical limitations rather than biological truth. We further divided TD based on the information available from neighboring cells: TD+Info represents cases where neighboring cells have abundant junction reads that can guide imputation, while TD-Info represents cases where such information is lacking in the local neighborhood. For BD, TD+Info and TD-Info, we recommend data imputation on the raw junction matrix with cell similarity network. And specifically, TD-Info requires an additional round of data diffusion using the event similarity network on the PSI matrix derived from previous round of imputation (**Methods, Extended Data Fig. 7-8**).” (Results Section, Page 5-6, Line 112-131)

Extended Data Figure 6. The hierarchical structure of four categories of cell-event pairs. Event-cell pairs with dropout in either inclusion or exclusion junctions are defined as WD (with dropout), while others are defined as ND (non dropout). In WD, event-cell pairs are categorized into BD for biological dropout or TD for technical dropout. In BD, both target and neighbor cells harbor good reads depth but lacking alternative junction reads, which imply a fixed isoform for this event. In contrast, the target cell of TD gets both low reads depth and limited alternative junction reads. Based on the abundance of alternative reads in neighbor cells, TD could be further classified into TD+Info that knowledge could be learned from local neighbors with rich splicing information, while others as TD-Info.

7. In the first example, the authors say:” We selected a set of high-confidence splicing events.” How many? How were they selected? What is the definition of high-confidence here?

Reply: We thank the Reviewer for this valuable comment. We define high-confident splicing events for the evaluation of different algorithms in cell line datasets. The high-confident splicing events for each comparison across four cell lines are selected by the following procedure: (1) the events should be detected by pairwise-comparison algorithms (e.g., SCSES and rMATS) in both single cell dataset and matched bulk dataset; (2) the junction read counts supporting either inclusion or exclusion exons should exceed 20 in bulk dataset; (3) when read counts of either upstream or downstream inclusion junction have fewer than 5 reads in bulk dataset, the read count fold change between them should be less than 10; (4) the genes harboring events should be expressed in both single cell dataset and matched bulk dataset; (5) to address Reviewer3's concern about cell cycle heterogeneity in cell line cells (Major point 1), AS events are filtered out if their harboring genes are annotated with the Gene Ontology Biological Process term 'cell cycle' (GO:0007049).

Due to differences of detectable genes in different cells, the number of high-confidence AS events varies across individual cells. We listed a part of event counts for each pairwise algorithm comparison in four cell lines in **Response Table 2**. The complete table has been added into the Supplementary Table files as **Supplementary Table 2**.

Cell type	Cell ID	BRIE	Outrigger	Psix	rMATS	SCASL
HCC1954	SRR1274192	893	1966	799	13851	13851
HCC1954	SRR1274194	875	1920	774	13445	13445
HCC1954	SRR1274196	839	1860	749	13092	13092
HCT116	SRR11826274	962	2140	856	14955	14955
HCT116	SRR11826275	958	2127	850	14825	14825
HCT116	SRR11826276	969	2134	860	14940	14940
HepG2	SRR11826359	944	2047	830	14444	14444
HepG2	SRR11826360	953	2074	835	14602	14602
HepG2	SRR11826361	942	2061	825	14557	14557
HL-60	SRR1275136	588	1502	533	10558	10558
HL-60	SRR1275138	577	1417	507	10032	10032
HL-60	SRR1275140	608	1469	531	10396	10396

Response Table 2: Part of high-confident splicing event counts for performance evaluation across different algorithms and cell types. Each row represents the splicing event counts for a single cell. The columns indicate the cell type, cell ID, and the overlapped high-confident splicing event counts between SCSES and the following algorithms: BRIE, Outrigger, Psix, rMATS, and SCASL.

In light of the reviewer’s suggestion, we have revised the manuscripts as follows:

“From these bulk datasets, we selected a set of high-confidence splicing events with PSI values served as the biological truth.” (Results Section, Page 6, Line 141-142)

“Then, the events with high confidence in bulk datasets were selected for evaluation based on the following procedure: (1) the events should be detected by pairwise-comparison algorithms (e.g., SCSES and rMATS) in both single cell dataset and matched bulk dataset; (2) the junction read counts supporting either inclusion or exclusion exons should exceed 20 in bulk dataset; (3) when read counts of either upstream or downstream inclusion junction have fewer than 5 reads in bulk dataset, the read count fold change between them should be less than 10; (4) the genes harboring events should be expressed in both single cell dataset and matched bulk dataset; (5) AS events are filtered out if their harboring genes are annotated with the Gene Ontology Biological Process term 'cell cycle' (GO:0007049). The event counts for each comparison and each cell are

listed in **Supplementary Table 2.**” (Methods Section, Page 27, Line 699-709)

8. “SCSES outperformed in terms of higher correlation and lower averaged errors in real datasets”. It is not clear if the higher correlation is based on PSI values, and with respect to what??

Reply: We thank the Reviewer for this valuable comment and apologize for the unclear statement.

We would like to clarify that the correlation refers to the Spearman correlation coefficients (SCC) between the PSI values estimated by the compared algorithms (e.g. SCSES and rMATS) and the benchmark PSI values (derived from bulk RNA-seq of the matched cell type) across all events within each cell. The errors refer to the root mean squared error (RMSE) between PSI values estimated by the compared algorithms (e.g. SCSES and rMATS) and the benchmark PSI values (derived from bulk RNA-seq of the matched cell type), calculated across all cells for each splicing event. The higher correlation and lower averaged errors indicate that, in each pairwise comparison, SCSES has higher SCC and lower averaged RMSE than other compared algorithms.

In light of Review’s suggestion, we have revised the manuscript as follows:

“We calculated the Spearman correlation coefficients (SCC) between the PSI values estimated by the compared algorithms and the benchmark PSI values (derived from bulk RNA-seq of the matched cell type) across all events within each cell, as well as the root mean squared error (RMSE) between estimated and the benchmark PSI values across all cells for each splicing event. Overall, in real datasets, SCSES consistently outperformed the competing algorithms by achieving higher cell-wise PSI correlations and lower event-wise PSI estimation errors (**Fig. 2a, 2b, Extended Data Fig. 11-13**).” (Results Section, Page 6, Line 151-158)

9. “0.05~0.6 in Spearman correlation confidence (SCC) values among cells, and reduction of more than 0.05 in root mean squared error (RMSE) among events”. The meaning of this sentence is unclear.

Reply: We thank the Reviewer for this valuable comment, and apologize for the lack of clarity in the initial submission. The meaning of the sentence refers to when compared with BRIE1, Expedition and rMATS, SCSES showed a 0.05-0.6 increase in the median Spearman correlation coefficient (SCC) between the PSI values estimated by the compared algorithms and the benchmark PSI values (derived from bulk RNA-seq of the matched cell type) across all events within each cell, and at least a 0.05 reduction in the median root mean squared error (RMSE) between PSI values estimated by the compared algorithms (e.g. SCSES and rMATS) and the benchmark PSI values (derived from bulk RNA-seq of the matched cell type), calculated across all cells for each splicing event.

In light of Review’s suggestion, we have revised the manuscript as follows:

“In all cases, SCSES outperformed BRIE1, Expedition and rMATS, exhibiting an extreme increase in median SCC values across cells, ranging from 0.05~0.6, and a reduction in median RMSE by more than 0.05 across events.” (Results Section, Page 7, Line 158-161)

10. “mimic a “fixed” population of the four cell types”. Is this a typo?

Reply: We thank the Reviewer for this valuable comment and sincerely apologize for the unclear description. “fixed” is not a typo. Here, we would like to express that synthesized cell counts of four cell lines were the same. To clarify our description, we have revised the manuscript as follows:

“Furthermore, we synthesized artificial scRNA-seq data to simulate a balanced population of the four cell types with varying sequencing qualities via Spanki simulator³⁷ (**Extended Data Fig. 9b, Supplementary Note**).” (Results Section, Page 7, Line 164-166)

11. The authors state that “BRIE2 requires the identity of each cell as prior information, while Psix and SCASL

require a fixed number of neighbor cells as input.” This is not clear, could these requirements be better described?

Reply: We thank the reviewer for pointing out the lack of clarity in our description. BRIE2 requires a pre-defined cell-specific feature (e.g., cell type or developmental stage) as prior information, which is used as regressor features of Bayesian regression. In contrast, Psix and SCASL require users to specify the number of nearest neighboring cells in advance, which constrains the learning of splicing information to fixed-size local cell populations. To address the reviewer's concern, we have revised the manuscript as follows:

“However, these algorithms require external information to estimate PSI. For instance, BRIE2 incorporates cell identity (e.g., cell type or developmental stage) as a prior feature for Bayesian regression. In contrast, Psix and SCASL require users to predefine the number of nearest neighboring cells, which constrains the learning of splicing information to fixed-size local cell populations.” (Results Section, Page 7, Line 178-182)

12. The limitation that most methods can only detect cassette exons should be introduced earlier, rather than at the end of the first example.

Reply: We thank the Reviewer for this valuable suggestion. We agree that introducing this limitation earlier would provide readers with a clearer understanding about technical limitations of current algorithms. We have added this limitation in the Introduction section, where we described the deficiencies of current algorithms. Specifically, we have revised the manuscript as follows:

“Nonetheless, there are still apparent deficiencies in each of the above methods. Firstly, some methods, such as BRIE2, heavily rely on cell type identities, which are unavailable before analysis. Secondly, methods including Psix use global gene expression to measure cell similarities, which may not fully reflect the complexity of heterogeneous splicing patterns among cells. Thirdly, the limited junction read counts often result in unreliable estimation of percent splice-in (PSI) values leading to inaccurate quantification in methods such as Expedition and SCASL (**Extended Data Fig. 2**). Lastly, most algorithms are limited in scope, primarily focusing on SE and MXE events, while lacking the ability to detect and quantify A3SS, A5SS and RI events. A detailed comparison of these methods is listed in **Supplementary Table 1**.” (Introduction Section, Page 4, Line 61-70)

13. In the third paragraph of the results section, the authors say: “To evaluate the accuracy of cell clustering by splicing imputation, the normalized mutual information was used to compare with the cell identities from the original publications. Overall, SCSES achieved the highest consistency with an increased score of more than 10% on nPSC and hEE datasets”. The meaning here is not clear.

Reply: We thank the Reviewer for pointing out this unclear description. To evaluate the accuracy of cell clustering based on splicing levels estimated by different algorithms, we calculated the normalized mutual information (NMI) between the *K*-means clustering results derived from the estimated PSI profiles and the cell type annotations provided in the original publications of the test datasets. Overall, SCSES showed the best performance, with an average NMI score improvement of more than 10% on both the nPSC and hEE datasets (**Fig. 3a**).

In light of Review’s suggestion, we have revised the manuscript to better explain how we evaluated the cell clustering accuracy and the performance improvement of SCSES as follows:

“To evaluate the accuracy of cell clustering based on splicing levels estimated by different algorithms, we calculated the normalized mutual information (NMI) between the *K*-means clustering results derived from the estimated PSI profiles and the cell type annotations provided in the original publications of the test datasets. Overall, SCSES showed the best performance, with an average NMI score improvement of more than 10% on both the nPSC and hEE datasets (**Fig. 3a**).” (Results Section, Page 9, Line 217-222)

14. At the end of the paragraph “SCSES reveals a drug-resistance cell group at primary diagnosis of multiple myeloma”, the authors state that “this observation requires experimental support in further studies.” Indeed, without further validation, I would not state in the title of the paragraph that a new subpopulation has been identified.

Reply: We thank the Reviewer for this valuable comment. We agree that without experimental validation, our statement about revealing a drug-resistance cell group was definitive. We have revised the title of this part as follows:

“SCSES reveals a cell group at primary diagnosis of multiple myeloma that is associated with potential drug-resistance” (Results Section, Page 10, Line 251-252)

15. What does “AS alternations” mean?

Reply: We thank the Reviewer for pointing out this typo, and apologize for our carelessness. We have corrected “AS alternations” to “Splicing alterations” in the revised manuscript.

“Splicing alterations could correctly track the cell evolution trajectory, and reveal the potential mechanism for BTZ resistance.” (Results Section, Page 13, Line 330-331)

“Enrichment analysis on differentially spliced genes (Supplementary Table 6) showed that splicing alterations were involved in the differentiation progress, such as endodermal cell fate commitment, WNT signaling pathway, and epithelial to mesenchymal transition (EMT) processes (Fig. 5b).” (Results Section, Page 13, Line 346-349)

“In this study, we introduced a novel computational method called SCSES to improve the identification and qualification of splicing alterations at single cell level.” (Discussion Section, Page 17, Line 455-456)

“Then, droplet-based data could be used to fine-tune the model to generate the complete splicing alterations.” (Discussion Section, Page 19, Line 507-508)

16. In the last paragraph of the results section, one or a few specific examples are described to demonstrate that the SCSES approach works properly. Out of how many events? How are the example events selected? Would the example events be detected by the other methods?

Reply: We thank the Reviewer for the valuable comment. In total, 2,978 AS events were detected in this dataset, and three monocyte subgroups (Mono1, Mono2 and Mono3) were detected. The example events (inclusion of *Runx1* 6th exon in Mono3, alternative last exon of *Lamp2* and exclusion of *Sh3glb1* 11th exon in Mono2) were selected through following analysis procedures:

First, to identify marker splicing events for each subgroup, we conducted differential splicing analysis by comparing splicing levels between each subgroup and all others using the Wilcoxon test. Marker events were defined by a $FDR < 0.05$, an absolute PSI difference $|\Delta PSI| > 0.1$, and a mean normalized expression level of the corresponding gene > 0.3 in at least one of the compared groups. This analysis revealed 445 and 347 marker splicing events in Mono2 and Mono3, respectively. Then, gene function enrichment analysis was performed on genes harboring these marker splicing events. Next, we focused on events from genes associated with function terms we were interested in (differentiation terms in Mono3, autophagy in Mono2). Finally, we selected events with previously reported functions, which led to our example events: inclusion of *Runx1* 6th exon in Mono3, alternative last exon of *Lamp2* and exclusion of *Sh3glb1* 11th exon in Mono2.

Regarding other methods, BRIE, Psix, Expedition and rMATS are able to detect inclusion of *Runx1* 6th exon and exclusion of *Sh3glb1* 11th exon, both of which are annotated in gene annotation file. However, none of these methods detects the alternative last exon of *Lamp2*. For SCASL, the splicing events are defined as junction groups sharing common splice sites. The unique definition of splicing events of SCASL prevents it from detecting all these three example events. In addition, these methods are not designed to process inDrop-

seq dataset, as they do not include the procedure of removing duplicated reads.

17. “Lamp2 generates 3 isoforms, LAMP-2A, LAMP-2B and LAMP-2C, by altering the last exon.” Is this alternative polyA rather than splicing?

Reply: We thank the Reviewer for this valuable comment. The three isoforms of *Lamp2* have been characterized as alternative last exon events in previous studies^{25,26}. Alternative polyadenylation (APA) refers to the use of multiple polyadenylation sites within the 3' untranslated region (3' UTR) of a gene. APA can be categorized into four subclasses (**Response Fig. 9**), among which tandem 3' UTR APA (**Response Fig.9A**) and splicing-APA (also known as alternative terminal exon polyadenylation, **Response Fig. 9B**) are the most common types^{2,27}. Splicing-APA can lead to transcripts with entirely distinct 3'UTR sequences as well as different C-terminal amino acids of the encoded protein². As shown in **Response Fig. 10**, the coding sequence of C-terminal of three *Lamp2* isoforms are completely non-overlapping, as well as 3' UTRs, supporting the notion that these isoforms are products of splicing-APA. Notably, splicing-APA often occurs in conjunction with splicing of an overlapping intron, suggesting that splicing regulation may also influence APA²⁸. For example, the expression levels of splicing factors from the MBNL family can influence the selection of APA sites in *Papola* and *Tpm1*²⁹. Deletion of MBNL proteins leads to a preferential usage of proximal APA sites in both genes²⁹. Therefore, it is not contradictory that *Lamp2a*, *Lamp2b* and *Lamp2c* are considered as products of both APA and alternative splicing.

Response Figure 9. Four types of alternative polyadenylation events. **(A)** Tandem 3'UTR APA occurs within transcripts that possess two or more cleavage/poly(A) sites within their 3' UTR, resulting in 3'UTR length differences between APA isoforms that code for identical gene products. **(B)** Alternative terminal exon APA is caused by alternative splicing, which results in internal exon skipping and use of a distinct poly(A) site contained within the new terminal exon. **(C)** Intronic APA involves the use of cryptic alternative poly(A) sites found within introns. **(D)** Internal exon APA features Poly(A) site usage within upstream exons, resulting in a truncated mRNA isoform lacking both a stop codon and a 3'UTR. This figure is adapted from Yuan F. et al (*Genes & Diseases*, 2021)² under a CC BY-NC-ND 4.0 license

Response Figure 10. Genomic structure of *Lamp2a*, *Lamp2b* and *Lamp2c*.

18. In the last paragraph of the results I would also include a discussion on the coverage limitation. Quantification of the bias could also be reported.

Reply: We thank the Reviewer for this valuable suggestion. In response, we have added a detailed comparison between inDrop-seq and SMART-seq data in the revised manuscript, focusing on three key aspects: read coverage distribution, alternative splicing (AS) event positions, and the gene regions affected by AS. Compared to the SMART-seq dataset³⁰, reads in the inDrop data were enriched near the 3' end of genes (**Extended Data Fig. 28a**). This 3' biased read coverage influenced the detection of splicing events towards the 3' ends of genes (**Extended Data Fig. 28b**). Consequently, AS events in the 3' untranslated regions (3' UTRs) were more frequently detected in inDrop sequencing data compared to SMART-seq data (**Extended Data Fig. 28c**).

Extended Data Figure 28. (a) Density plot showing the distribution of sequencing reads along gene bodies, from transcript start sites to transcript termination sites. (b) Density plot showing the relative distance of AS regions to the transcript start site, stratified by AS event types. ks: Kolmogorov-Smirnov Test. (c) Bar plots showing the percentage of AS events in each gene region (ncRNA, 5'UTR, CDS, 3'UTR), stratified by AS types. ncRNA: the genes harboring the events is non-coding genes. 3'UTR: either the upstream or downstream exon of the alternative region contains 3'UTR. 5'UTR: either the upstream or downstream exon of the alternative region contains 5'UTR. CDS: the upstream and downstream exon of the alternative region contain only coding sequences.

In light of Review's suggestion, we have added the **Extended Data Fig. 28** in the revised Extended Data file, and revised the manuscript as follows:

“Compared to the SMART-seq dataset⁹², the read coverage in inDrop data was enriched near the 3' end of genes (**Extended Data Fig. 28a**). This 3' biased read coverage influenced the detection of splicing events towards the 3' ends of genes (**Extended Data Fig. 28b**). Consequently, AS events in the 3' untranslated regions (3' UTRs) were more frequently detected in inDrop sequencing data compared to SMART-seq data (**Extended Data Fig. 28c**)” (Results Section, Page 15, Line 408-413)

19. Figure 2a and 2b show pairwise plots. SCESC information is displayed multiple times. Could it be only one overall comparison among all tools?

Reply: We thank the reviewer for this valuable comment. As shown in **Extended Data Fig. 19a** of our initial submission, the splicing events detected by different methods exhibit substantial variability. To ensure a fair and comprehensive comparison, we conducted pairwise comparisons between SCSES and each of the other algorithms. This approach enables a direct evaluation of SCSES's performance relative to individual tools, taking into account the inherent differences in event detection capabilities and methodological frameworks. While an all-in-one comparison using only the events commonly detected by all methods might appear more concise, it would overlook a substantial number of tool-specific events, failing to capture the whole performance of each algorithm. Therefore, we believe that pairwise comparisons provide a more detailed and informative assessment of SCSES's performance relative to existing methods.

To clarify of our description, we have revised the manuscript as follows:

“Due to the substantial variability in splicing events identified by different algorithms, we performed pairwise comparisons between SCSES and each of the other tools, focusing only on the overlapping splicing events to ensure a fair evaluation.” (Results Section, Page 7, Line 148-150)

Extended Data Figure 19a: Venn diagram showing the event counts identified by different methods in real cell line sc-RNAseq datasets.

20. Figure 2e: how many DSEs are considered?

Reply: We thank the reviewer for this valuable comment. The number of DSEs are listed in **Supplementary Table 3**. Comparisons between different methods and SCSES are presented in rows, while differential splicing analysis combinations based on four cell lines are divided into six groups in columns. For example, in the comparison of rMATS and SCSES for differential splicing analysis between HCT116 and HepG2 cell lines, 16,197 splicing events were detected by both rMATS and SCSES. Among these, 196 events were identified as true positives, which were defined as events with $|\Delta PSI| > 0.2$ between two matched bulk data.

In light of the Reviewer's suggestion, we have added the **Supplementary Table 3** into revised Supplementary Tables, and revised the manuscript as follows:

“Subsequently, we assessed the capability of detecting differentially spliced events (DSEs) between these four cell types using the receiver operating characteristic curve (**Methods, Supplementary Table 3**).” (Results Section, Page 8, Line 185-187)

Methods	HCT116 vs HepG2			HCT116 vs HL-60			HCT116 vs HCC1954			HepG2 vs HL-60			HepG2 vs HCC1954			HL-60 vs HCC1954			
	AUC	TP Events	Total Events	AUC	TP Events	Total Events	AUC	TP Events	Total Events	AUC	TP Events	Total Events	AUC	TP Events	Total Events	AUC	TP Events	Total Events	
rMATS	Ref			0.32			0.39			0.36			0.43			0.50			
	SCSES-RBP	0.66	196	16197	0.50	268	13948	0.60	214	16725	0.45	213	13692	0.57	175	16198	0.54	267	13855
	SCSES-PSI	0.68			0.48			0.62			0.43			0.53			0.50		
	SCSES-RC	0.65			0.53			0.67			0.50			0.57			0.59		
Expedition	Ref	0.53			0.46			0.54			0.46			0.50			0.50		
	SCSES-RBP	0.61	53	2402	0.58	67	2072	0.63	54	2462	0.49	56	2007	0.55	46	2366	0.57	67	2022
	SCSES-PSI	0.62			0.54			0.65			0.47			0.48			0.55		
	SCSES-RC	0.59			0.62			0.67			0.52			0.55			0.61		
BRIE1	Ref	0.54			0.50			0.49			0.46			0.55			0.37		
	SCSES-RBP	0.62	30	1110	0.60	35	873	0.60	24	1139	0.56	33	853	0.61	21	1093	0.54	33	864
	SCSES-PSI	0.67			0.57			0.63			0.52			0.57			0.50		
	SCSES-RC	0.65			0.60			0.71			0.58			0.56			0.61		
BRIE2 aggr	Ref	0.58			0.57			0.51			0.35			0.46			0.46		
	SCSES-RBP	0.62	30	1110	0.60	35	873	0.60	24	1139	0.56	33	853	0.61	21	1093	0.54	33	864
	SCSES-PSI	0.67			0.57			0.63			0.52			0.57			0.50		
	SCSES-RC	0.65			0.60			0.71			0.58			0.56			0.61		
Psix	Ref	0.66			0.49			0.64			0.52			0.57			0.57		
	SCSES-RBP	0.62	90	970	0.65	97	770	0.62	86	1009	0.54	79	751	0.58	79	961	0.58	104	761
	SCSES-PSI	0.61			0.63			0.64			0.57			0.52			0.57		
	SCSES-RC	0.60			0.65			0.65			0.57			0.56			0.60		
SCASL	Ref	0.67			0.36			0.49			0.33			0.43			0.54		
	SCSES-RBP	0.66	196	16197	0.50	268	13948	0.60	214	16725	0.45	213	13692	0.57	175	16198	0.54	267	13855
	SCSES-PSI	0.68			0.48			0.62			0.43			0.53			0.50		
	SCSES-RC	0.65			0.53			0.67			0.50			0.57			0.59		

Supplementary Table 3: Evaluation Summary of Differential splicing event detection. Each row shows the comparison between SCSES and a reference method ("Ref"), where "SCSES-RBP", "SCSES-PSI", and "SCSES-RC" indicate SCSES results using different cell similarity features. The columns represent six groups of differential splicing analysis combinations based on four cell lines. Three values are reported for each comparison: Area Under the Curve (AUC), True Positive events (TP Events, number of differential splicing events with $|\Delta PSI| > 0.2$ between two cell lines on matched bulk data), and Total Events (number of splicing events identified by both methods).

21. Figure 3: Is the UMAP based on gene expression shown?

Reply: We thank the Reviewer for this valuable comment. We have added the UMAP plots based on gene expression values as **Extended Data Fig. 21b** in the revised Extended Data file, and revised the manuscript as follows:

“Interestingly, low dimensional reduction analysis showed that SCSES divided the motor neurons (MNs) into two more subclusters (MN-C1, MN-C2), a distinction that was not as clearly delineated by other methods or by gene expression values (**Fig. 3b, 3c, Extended Data Fig. 21a, b**).” (Results Section, Page 9, Line 224-227)

22. Figure 4a and 4b: How many events? How many genes?

Reply: We thank the reviewer for this valuable comment. For **Figure 4a** and **4b**, top 5,000 highly variable features (genes or splicing events) were selected for PCA analysis. Subsequently, the principal components with cumulative explained variance exceeding 90% were used to identify neighboring cells and perform UMAP projection. Relative description was included in the Supplementary Note Section of initial submitted Extended Data file.

Figure 4a, 4b in the initial submission. a, b. UMAP plots showing cell projections in MM34 patient estimated by splicing profile (a) and gene expression profile (b). Cell colors represent the clusters, and shapes represent the source of samplings. PD: primary diagnosis, Meta: metastasis.

Manuscript Modification

The current manuscript's English needs significant editing to reach publication standards. Here I reported some of the problems:

Reply: We thank the Reviewer for this valuable comment. In response to the reviewer's comments about language, we have carefully revised the entire manuscript to improve its grammar and clarity. All changes are marked in red font for easy review.

1. The title: "Deciphering splicing heterogeneity on single cell resolution by SCSES", should be "at single cell resolution".

Reply: We thank the Reviewer for this valuable comment. We have modified the manuscript title in the revised manuscript as follows:

"Deciphering splicing heterogeneity at single cell resolution by SCSES" (Title, Page 1, Line 1)

In the introduction section:

2. "Single cell RNA sequencing (scRNA-seq) techniques make it possible to decipher cell transcriptional heterogeneity on large scale of cells". Correct with "on a large scale of cells".
3. "The gene expression specificity in different single cell clusters have been decoded in many studies". Correct with "has been decoded".
4. "but most of which neglected". Correct with "but most of them".
5. "during pre-mRNA maturity". Correct with "during pre-mRNA maturation".
6. "no matter on full-length protocols like smart-seq or the droplet-based protocols like inDrop-seq". Correct with "even with full-length"
7. "which is independent with the conventional single-cell gene expression analysis". Correct with "with respect to the conventional"
8. Typo in Figure 1A "megre" all reads.

Reply: We thank the Reviewer for all the valuable comments in Introduction section. We have revised the manuscript as follows:

“Single cell RNA sequencing (scRNA-seq) techniques make it possible to decipher cell transcriptional heterogeneity on a large scale of cells¹” (Introduction Section, Page 3, Line 32-33)

“While many studies have successfully characterized the gene expression specificity in different single cell clusters²⁻⁶, the complexity of post-transcriptional regulation has often been overlooked.” (Introduction Section, Page 3, Line 33-35)

“Alternative splicing (AS), an essential post-transcriptional process during pre-mRNA maturation, substantially contributes to tremendous transcriptional diversity⁷⁻⁹” (Introduction Section, Page 3, Line 35-37)

“However, due to technical limitations of single-cell sequencing such as high dropout rate, high sequencing errors, and biased low coverage^{9,22}, these tools are poorly adaptive for scRNA-seq datasets^{23,24}, even when using full-length protocols like smart-seq²⁵ or the droplet-based protocols like inDrop-seq²⁶.” (Introduction Section, Page 3, Line 49-53)

“In summary, SCSES is a promising tool to explore and interpret single cell data from a post-transcriptional perspective concerning the conventional single-cell gene expression analysis.” (Introduction Section, Page 4, Line 78-81)

In the results section:

9. “It takes scRNA-seq alignments as input, and output refined PSI value of every detected splicing event in each cell.” Correct with “outputs” if used as a verb.
10. “We then used real scRNA-seq data of these four cells.” It is more accurate to say “these four cell lines”.
11. “We then used real scRNA-seq data of these four cells to test if SCSES could recapitulate the splicing landscape on single cell level”. Correct “on” with “at single cell level”.
12. “And the imputed PSI values by SCSES in these events were highly correlated with PTBP1 expression”. Sentences should not begin with “and”.
13. “Specifically, SC2 cells upregulated genes were related to cell mitosis”. Correct with “Specifically, genes upregulated in SC2..”
14. “In addition, 19 differentially spliced genes have been reported to involve in BTZ resistance”. Correct with “have been reported to be involved in BTZ”.
15. “By further checking DSEs determining this clustering, we found events that exclusively appeared at unique time point”. Correct with “at unique time points”.
16. “Moreover, we found RBM24”. Correct with “Moreover, we found that RBM24”.

Reply: We thank the Reviewer for all the valuable comments in Results section. We have revised the manuscript as follows:

“It takes scRNA-seq alignments as input and outputs refined PSI values for every detected splicing event in each cell.” (Results Section, Page 4, Line 87-88)

“We then used real scRNA-seq data for the same cell lines to test if SCSES could accurately recapitulate the splicing landscape at the single cell level (**Extended Data Fig. 9a, Extended Data Fig. 10**).” (Results Section, Page 6, Line 142-144)

“The imputed PSI values estimated by SCSES for these events were highly correlated with PTBP1 expression.” (Results Section, Page 9, Line 242-243)

“Specifically, genes upregulated in SC2 cells were related to cell mitosis, heat shock response and L-glutamine (Gln) process.” (Results Section, Page 11, Line 292-294)

“In addition, 19 differentially spliced genes have been reported to be involved in BTZ resistance.” (Results

Section, Page 12, Line 314-315)

“By further checking DSEs determining this clustering, we found events that exclusively appeared at unique time points, as well as changes that persistently presented in successive stages.” (Results Section, Page 13, Line 349-352)

“Moreover, we found that *RBM24*, reported to regulate mesenchymal-like splicing patterns⁸⁴, was up-regulated in DE stage cells.” (Results Section, Page 14, Line 365-367)

Reviewer #1 (Remarks on code availability):

The author says that SCSES is an R package but actually, it requires R, Python, Matlab Compiler Runtime, and Java, as also reported in the GitHub page. Additionally, it requires the manual installation of rMats, MAJIQ, IRFinder and STAR. Would it be useful to create a container grouping together the necessary packages to run SCSES? In the end, I was unable to install it on my machine due to discrepancies with the dependencies. However, this may be an issue on my end rather than a problem with the package itself.

Reply: We thank the reviewer for the valuable comment and apologize for the inconvenience caused by the installation process. To address these challenges and facilitate user experience, we have created a Dockerfile that integrates all the environmental dependencies required for SCSES. Below, we provide a step-by-step guide for setting up SCSES using Docker:

1. **Install Docker Client:** Users should first install the Docker client on their host machine.
2. **Download Dockerfile:** The Dockerfile can be downloaded from: <https://github.com/lvxuan12/SCSES/blob/main/SCSES.dockerfile>.
3. **Build Docker Image:** Users can build the SCSES Docker image using the command:
“docker build -t scses -f SCSES.dockerfile .”
4. **Create Docker Container:** After building the image, users can create a Docker container with the following command:
“docker run -d -p [exported port]:8787 -e PASSWORD=[user password] -v [local directory]:/data --name test scses”

[exported port]: The port on the host machine to access the container.

[user password]: A user-defined password for logging into the RStudio server.

[local directory]: A local directory mapped to the container for data storage and sharing.
5. **Access RStudio Server:** Users can access the RStudio server by opening a web browser and navigating to “[host IP]:[exported port]”. Use the default username “rstudio” and the user-defined password to log in.

In this pre-configured RStudio server environment, SCSES and all its dependencies are correctly installed and ready for use. For more details, please refer to the GitHub page: <https://github.com/lvxuan12/SCSES>.

We hope this Docker solution resolves the installation issues and improves the accessibility of SCSES. If further assistance is needed, please do not hesitate to contact us.

Reviewer #2 (Remarks to the Author):

This paper proposes a novel algorithmic framework, SCSES, for analyzing variable splicing at the single-cell level. Overall, the study demonstrates significant innovation and biological relevance, but there are still some details that need further refinement. The specific comments and suggestions are as follows:

REPLY: We thank the Reviewer for the kind compliments that “the study demonstrates significant innovation and biological relevance.”

1. The description in line 153 of the paper regarding Fig. 2e is not ideal. While emphasizing the improvement in AUC is important, the specific AUC values should also be provided. Improvement is meaningful only when built upon strong baseline values.

Reply: We thank the reviewer for this valuable comment. The AUC values are listed in **Supplementary Table 3**. Comparisons between different methods and SCSES are presented in rows, while differential splicing analysis combinations based on four cell lines are divided into six groups in columns. For example, in the comparison of rMATS and SCSES for differential splicing analysis between HCT116 and HepG2 cell lines, rMATS showed an AUC of 0.63 (“Ref”), while SCSES exhibited better performance with AUC values of 0.66, 0.68, and 0.65 for SCSES-RBP, SCSES-PSI, and SCSES-RC, respectively.

In light of the Reviewer’s suggestion, we have added the **Supplementary Table 3** into revised Supplementary Tables, and revised the manuscript as follows:

“Subsequently, we assessed the capability of detecting differentially spliced events (DSEs) between these four cell types using the receiver operating characteristic curve (**Methods, Supplementary Table 3**).” (Results Section, Page 8, Line 185-187)

Methods	HCT116 vs HepG2			HCT116 vs HL-60			HCT116 vs HCC1954			HepG2 vs HL-60			HepG2 vs HCC1954			HL-60 vs HCC1954			
	AUC	TP Events	Total Events	AUC	TP Events	Total Events	AUC	TP Events	Total Events	AUC	TP Events	Total Events	AUC	TP Events	Total Events	AUC	TP Events	Total Events	
rMATS	Ref	0.63		0.32			0.39			0.36			0.43			0.50			
	SCSES-RBP	0.66	196	16197	0.50	268	13948	0.60	214	16725	0.45	213	13692	0.57	175	16198	0.54	267	13855
	SCSES-PSI	0.68			0.48			0.62			0.43			0.53			0.50		
	SCSES-RC	0.65			0.53			0.67			0.50			0.57			0.59		
Expedition	Ref	0.53		0.46			0.54			0.46			0.50			0.50			
	SCSES-RBP	0.61	53	2402	0.58	67	2072	0.63	54	2462	0.49	56	2007	0.55	46	2366	0.57	67	2022
	SCSES-PSI	0.62			0.54			0.65			0.47			0.48			0.55		
	SCSES-RC	0.59			0.62			0.67			0.52			0.55			0.61		
BRIE1	Ref	0.54		0.50			0.49			0.46			0.55			0.37			
	SCSES-RBP	0.62	30	1110	0.60	35	873	0.60	24	1139	0.52	33	853	0.61	21	1093	0.54	33	864
	SCSES-PSI	0.67			0.57			0.63			0.52			0.57			0.50		
	SCSES-RC	0.65			0.60			0.71			0.58			0.56			0.61		
BRIE2 aggr	Ref	0.58		0.37			0.51			0.35			0.46			0.46			
	SCSES-RBP	0.62	30	1110	0.60	35	873	0.60	24	1139	0.56	33	853	0.61	21	1093	0.54	33	864
	SCSES-PSI	0.67			0.57			0.63			0.52			0.57			0.50		
	SCSES-RC	0.65			0.60			0.71			0.58			0.56			0.61		
Psix	Ref	0.66		0.49			0.62			0.52			0.57			0.57			
	SCSES-RBP	0.62	90	970	0.65	97	770	0.62	86	1009	0.54	79	751	0.56	79	961	0.58	104	761
	SCSES-PSI	0.61			0.63			0.64			0.57			0.52			0.57		
	SCSES-RC	0.60			0.65			0.65			0.57			0.56			0.60		
SCASL	Ref	0.67		0.36			0.49			0.33			0.43			0.54			
	SCSES-RBP	0.66	196	16197	0.50	268	13948	0.60	214	16725	0.45	213	13692	0.57	175	16198	0.54	267	13855
	SCSES-PSI	0.68			0.48			0.62			0.43			0.53			0.50		
	SCSES-RC	0.65			0.53			0.67			0.50			0.57			0.59		

Supplementary Table 3: Evaluation Summary of Differential splicing event detection. Each row shows the comparison between SCSES and a reference method (“Ref”), where “SCSES-RBP”, “SCSES-PSI”, and “SCSES-RC” indicate SCSES results using different cell similarity features. The columns represent six groups of differential splicing analysis combinations based on four cell lines. Three values are reported for each comparison: Area Under the Curve (AUC), True Positive events (TP Events, number of differential splicing events with $|\Delta\text{PSI}|>0.2$ between two cell lines on matched bulk data), and Total Events (number of splicing events identified by both methods).

2. It is recommended to further label the MN in Fig. 3c as MN-C1 and MN-C2. Since Psix also has a small blue cluster separated out, distinguishing MN-C1 and MN-C2 is crucial.

Reply: We thank the reviewer for the valuable comment. We have revised the **Fig. 3c**, and highlighted MN-C1 and MN-C2 using different colors. The revised **Fig. 3c** is shown as below:

Figure 3c. UMAP plots of iPSC test dataset by PSI values estimated by other algorithms.

- In line 194 of the paper, it is mentioned that the C1 cells are suspected to be in the early stage of differentiation from iPSCs to MNs. This hypothesis could be preliminarily supported by performing pseudotime analysis combined with UMAP.

Reply: We thank the reviewer for this valuable comment. To address this problem, we used CytoTrace²⁰ and Monocle²¹ to assess their differentiation state. The results showed that MN-C1 exhibited stronger stemness and earlier pseudotime compared to MN-C2 (**Response Fig. 5**), suggesting that MN-C1 corresponds to an earlier stage of motor neuron maturation.

Response Figure 5. (a). Boxplot showing the pseudotime distribution of different cell groups estimated by Monocle algorithm based on expression. (b). Boxplot showing the distribution of stemness scores of in different cell groups.

In light of the Reviewer's suggestion, we have added the **Response Fig. 5** into revised Extended Data file as **Extended Data Fig. 21c, f (right)** and we have revised the manuscript as follows:

"In addition, the pseudotime analysis by Monocle⁴⁶ and CytoTrace⁴⁷ demonstrated that MN-C1 exhibited stronger stemness and earlier pseudotime compared to MN-C2 cells (**Extended Data Fig. 21c-f**). Consistently, MN-C1 cells presented a more similar gene expression pattern with the neural progenitor cells (**Fig. 3e**)." (Results Section, Page 10, Line 233-237)

- Can the correlation between 72h and 96h in Fig. 5a be further discussed? Why do these time points partially overlap?

Reply: We thank the Reviewer for this valuable comment. Below we provide several pieces of evidence to explain this observation.

First, we observed the similar overlap between 72h- and 96h-cells using different dimension reduction methods (PCA, tSNE), applied to both entire dataset and the subset of 72h- and 96h-cells (**Response Fig. 11, 12**). The results indicated that the observed overlap was not an artifact of a particular algorithm, or the result of absence of time-specific features.

Response Figure 11. Dimensionality reduction analysis of the whole dataset. Visualization of cells using PCA (left), tSNE (middle), UMAP (right) based on PSI value. Each point represents a single cell, colored by time points.

Response Figure 12. Dimensionality reduction analysis of cells from 72 and 96 hours. Visualization of cells using PCA (left), tSNE (middle), UMAP (right) based on PSI value. Each point represents a single cell, colored by time points.

Furthermore, most of 72h- and 96h-cells were overlapped in the PCA-reduction plot based on gene expression in the previous study where the dataset was generated¹ (**Response Fig. 13**). The authors stated that with a general decrease of differentially expressed genes from 0h to 96h, “cells could gradually transition into a relatively ‘stable’ state at 72h of differentiation”. To validate this statement, we compared the expression levels of *CXCR4* and *SOX17*, both of which are definitive endoderm marker genes³¹, and found no significant changes between 72h and 96h (**Extended Data Fig. 26a, b**). Additionally, we also found that the number of differentially spliced events also gradually decreased after 72h (**Extended Data Fig. 26c**). These findings support the authors' conclusion that cells reach a relatively stable differentiation state by 72h.

Response Figure 13. Fig. 3A adapted from Chu et. al¹. Upper panel, schematics of experimental strategy illustrating time points of scRNA-seq sampling along the differentiation from pluripotent state through mesendoderm to DE cells. Lower panel, PCA of scRNA-seq data, shown is PC1 vs. PC2.

Extended Data Figure 26. (a). Violin plot showing the expression of CXCR4 among time points. (b). Violin plot showing the expression of SOX17 among time points. (c). Bar plot showing number of differentially spliced events between each pair of neighboring time points. (d). UMAP plots of cells from 72 and 96 hours, colored by cell subpopulations. (e). Distribution plots showing pairwise cell distances for three comparisons: intrapopulation distances within 72h cells, interpopulation distances between 72h and 96h-specific cells, and between 72h and 96h-overlapped cells. (f). CytoTrace score for cells in the three subpopulations. (g). Scatter plot comparing splicing changes of differentially spliced events identified between 96h and other time points in two contexts: 96h cells versus other cells (x-axis) and 96h-specific cells versus 96h-overlapped cells (y-axis).

Finally, to further investigate the heterogeneity among 96h-cells, we stratified the 96h-cells into two subpopulations, based on their distances to 72h-cells (**Extended Data Fig. 26d, e**). The 96h-cells

overlapping with 72h-cells were labeled as “96-overlap”, and others were labeled as “96h-specific”. CytoTrace-based analysis of cellular stemness revealed that the 96-overlap cells showed greater developmental potential (**Extended Data Fig. 26f**), implying an incomplete differentiation state. Furthermore, we detected 253 differentially spliced events between 96h-cells and other cells, and 245 of which were detected with the same alteration direction between 96-specific cells and 96-overlapped cells (**Extended Data Fig. 26g**). All these results indicate that 96-specific cells were in a more mature stage compared to 96-overlap cells.

In light of Review’s comment, we have added the discussion above into the revised Extended Data file (**Supplementary Note Section, Page 7-8, Line 162-185**), and revised the manuscript as follows:

“Most cells at different time points were clustered into independent groups by SCSES splicing profile, indicating clear splicing dynamics throughout the differentiation course (**Fig. 5a**), while a small set of cells from 96h overlapped with cells from 72h (**Fig. 5a**). Further analysis demonstrated that the cells appeared to reach a relatively stable state by 72 hours, consistent with previous studies⁷³, and the non-overlapped 96h-cells were in a more mature stage compared to overlapped 96h-cells (**Extended Data Fig. 26, Supplementary Note**).” (**Results Section, Page 13, Line 340-346**)

5. It is suggested to add experiments related to model performance, such as ablation studies, to discuss the contribution of each component of the model.

Reply: We thank the Reviewer for this valuable comment. We added ablation experiments to test the contribution of imputation strategy integration, similarity diffusion and fine-tune procedure of the scenario decision model of event-cell pairs.

Firstly, we compared the performance of the integrated strategy with that of individual strategies, which included: Strategy 1 (imputing raw PSI using cell similarity), Strategy 2 (imputing raw junction read counts using cell similarity), and Strategy 3 (imputing the results from Strategy 2 using event similarity). We demonstrated that the integrated approach outperformed the individual strategies in most cases, particularly in terms of PSI accuracy and cell clustering (**Extended Data Fig. 31**). For example, in the HCT116 dataset, the integrated strategies showed higher SCC values between the predicted PSI values and benchmark PSI values compared to the individual strategies, indicating more accurate PSI estimation. Additionally, in the nPSC dataset, the NMI values for cell clustering using the integrated strategy were consistently higher than those from the individual strategies, suggesting more consistent clustering with the real cell types.

Secondly, we evaluated the contribution of similarity diffusion to the overall performance of the algorithm. As shown in **Extended Data Fig. 32**, the diffusion procedure improved the accuracy of both PSI estimation and cell clustering in most cases. For instance, in the HCT116 dataset, applying diffusion increased the SCC values between predicted and benchmark PSI values by at least 0.05. In the iPSC dataset, the diffusion step led to a 70% improvement in NMI for cell clustering. These results highlight the importance of similarity diffusion in enhancing the final splicing profile.

Finally, we evaluated the contribution of the fine-tuning procedure in the scenario decision model. The pre-trained model was originally trained on single-cell cell line datasets. To assess the benefit of fine-tuning, we used down-sampled datasets from iPSC, hEE, and nPSC. As shown in **Extended Data Fig. 33**, the fine-tuned model consistently outperformed the pre-trained model across all cases. For example, in the iPSC dataset, fine-tuning increased the SCC between predicted and benchmark PSI values by more than 0.1, and improved the NMI for cell clustering by over 30%. These results demonstrate that the fine-tuning procedure enables the model to better adapt to specific datasets.

In light of the reviewer’s suggestion, we have added the corresponding results into the revised manuscript. The related figures and description have been added into the revised Extended Data file as **Extended Data Fig. 31-33** (**Discussion Section, Page 12, Line 294-325**). The revised context in revised manuscript is as

follows:

“Through systematic comparison of the integrated strategy versus individual strategies, we demonstrated that the integrated approach improved PSI estimation accuracy and enhanced clustering consistency in most cases (**Extended Data Fig. 31**). Furthermore, several carefully designed computational procedures, such as similarity diffusion and fine-tune procedure of the scenario decision model, provided additional performance enhancements, validated by ablation studies (**Extended Data Fig. 32-33, Supplementary Note**).” (Discussion Section, Page 18, Line 475-481)

Extended Data Figure 31. Ablation study evaluating the contribution of different imputation strategy integrations. (a) Spearman correlation coefficients (SCC) between estimated and benchmark PSI values for all events across cells, comparing different imputation strategies. (b) Normalized Mutual Information (NMI) scores assessing clustering performance in down-sampling datasets of iPSC, hEE, and nPSC for each strategy. “Strategy 1” refers to imputing raw PSI using cell similarity, “Strategy 2” refers to imputing raw junction read counts using cell similarity, and “Strategy 3” refers to imputing the results from Strategy 2 using event similarity. “SCSES” indicates that the integrated strategy was applied.

Extended Data Figure 32. Ablation study evaluating the impact of similarity diffusion on imputation performance. (a) Comparison of PSI estimation accuracy (Spearman correlation coefficients, SCC) between estimated and benchmark values for raw versus diffused similarity networks. (b) Clustering performance (Normalized Mutual Information, NMI) in down-sampling datasets (iPSC, hEE, nPSC) using raw versus diffused similarity approaches. Raw similarity indicates that no diffusion procedure was applied, whereas diffused similarity indicates that diffusion procedure was applied.

Extended Data Figure 33. Ablation study evaluating the impact of model fine-tuning on splicing imputation performance. (a) Comparison of PSI estimation accuracy (Spearman correlation coefficients, SCC) between gene-expression-derived pseudotime and PSI-derived pseudotime for pre-trained versus fine-tuned models. (b) Clustering performance (Normalized Mutual Information, NMI) in down-sampling stem cell datasets (iPSC, hEE, nPSC) comparing pre-trained and fine-tuned models. “Basic” indicates that no fine-tuning procedure was applied, whereas “Fine-tune” indicates that fine-tuning procedure was applied.

Reviewer #3 (Remarks to the Author):

In this paper, the authors present a novel data imputation method, SCSES, designed to analyze alternative splicing (AS) quantification at the single-cell level. In addition to the pseudo-bulk approach based on cell similarity, they also propose utilizing splicing event similarity to address challenging cases, such as technical dropouts without sufficient neighboring cell information (TD-info). The authors list several applications of SCSES. However, there are multiple significant concerns about the validity and robustness of the model.

REPLY: We thank the Reviewer for the careful consideration of our manuscript and thoughtful comments.

Major Concerns:

1. Potentially false assumption. The model evaluation primarily relies on bulk RNA-seq data from cell lines, assuming that all cells within a cell line are identical. However, using AS quantification from bulk RNA-seq as the "gold standard" may be problematic, as cells within the same cell line can exhibit heterogeneity (e.g., due to different cell cycle stages or other factors). This assumption could introduce biases or inaccuracies in model evaluation.

Reply: We thank the Reviewer for this valuable comment. Below, we address the rationale for using bulk RNA-seq data of cell lines as benchmark, and modified the event list for benchmark.

(1) Rationale for using cell line datasets in our initial submission

We would like to explain the reason why we used cell line dataset for evaluation of the performance of SCSES in our initial submission. Due to the cell heterogeneity of real single-cell data and technical limitations of scRNA-seq data (high dropout, high noise, low coverage), it's a great challenge to obtain the accurate composition of cell types along with the exact splicing changes on each cell type from a real single-cell data for method benchmarking. Therefore, in the initial submission, single-cell RNA-seq datasets of cell lines were chosen for benchmarking due to their minimal cellular heterogeneity and the availability of accurate splicing characteristics obtained from the bulk data of matched cell type. Specifically:

- a) Minimal cellular heterogeneity of scRNA-seq datasets of cell lines

Cell lines are derived from a single cell or a small group of cells, and are propagated under controlled laboratory conditions. Standardized culture conditions (e.g., growth medium, temperature, pH) further minimize variations in cellular behavior, resulting in a homogeneous cell population. This ensures a highly consistent genetic background across the population.

To validate the minimal splicing heterogeneity, we plotted UMAPs of four cell lines using raw PSI values of high-quality events. Splicing events were considered as high quality if they met the sequencing quality threshold in at least 5 cells. The quality thresholds included: (1) more than 20 junction reads to support either inclusion or exclusion exons; (2) when read counts of either upstream or downstream inclusion junction have fewer than 5 reads, the read count fold change between them should be less than 10. The second threshold is used to exclude events with a huge difference of reads depth between the upstream or downstream splice sites, which may lead to inaccurate estimation of PSI value. As shown in **Response Fig. 1**, cells of each cell line were randomly distributed without obvious clustering or trajectory, confirming the lack of significant heterogeneity.

Response Figure 1. UMAP plots based on the PSI values of high-quality AS events showing the lack of heterogeneity in scRNA-seq data of four cell lines. The event counts used for UMAP plots are 5218 for HCT116, 4377 for HepG2, 3653 for HCC 1954, and 1506 for HL-60.

b) Accurate quantification of splicing events from bulk RNA-seq of matched cell types for method benchmarking:

Accurate quantification of alternative splicing requires high depth of junction reads, which is more feasible with bulk RNA-seq data. However, bulk data from tissue often represent mixed cell types and states, making them unsuitable as benchmarks for splicing changes under specific cell types. Due to the minimal heterogeneity, bulk RNA-seq data from cell lines provides an ideal reference for quantification of splicing events, thus is suitable for method benchmarking of quantifying splicing events from the matched cell type on single-cell level.

To validate this, we calculated the PSI differences of high-quality event-cell pairs (meeting the quality thresholds as described above) between single-cell and matched bulk datasets. As shown in **Response Fig. 2a**, more than 82% of the pairs exhibited differences of less than 0.1. Additionally, the Pearson correlation coefficient (PCC) of PSI values between single cell and bulk samples exceeded 0.9 for more than 80% of cells (**Response Fig. 2b**). These results indicate that the splicing profiles from bulk RNA-seq data of cell lines are highly similar to those from matched scRNA-seq datasets, making them suitable benchmarks for evaluating AS quantification methods.

Response Figure 2. Density plots showing the high splicing consistency between bulk and single cell datasets of cell lines. (a) Distribution of PSI differences between bulk datasets (Ref) and corresponding event-cell pairs (Raw). (b) Distribution of Pearson correlation coefficients (PCC) between PSI values from bulk data and corresponding single cells.

c) Similar strategies adopted in previous studies:

The very similar strategy of using single-cell and bulk RNA-seq data from the same cell lines has been commonly used to evaluate the method performance of gene expression and alternative splicing estimation at the single-cell level. For example, Hou et al. systematically evaluated 18 scRNA-seq gene expression imputation algorithms by comparing imputed single-cell profiles with bulk RNA-seq profiles from the same cell lines, leveraging their less heterogeneous and well-defined gene expression as benchmarks³. Huang et al. used single-cell and bulk RNA-seq datasets of HCT116 to evaluate the accuracy of estimated PSI by BRIE⁴. Inspired by these studies and many others on the related research topic, we have decided to use single-cell RNA-seq of cell lines as the test sets for method comparisons, and bulk RNA-seq data of the same cell types were used to provide the background truth of the splicing changes for the performance evaluations in our initial submission. Huang et al. used single-cell and bulk RNA-seq datasets of HCT116 to evaluate the accuracy of estimated PSI by BRIE⁴. Inspired by these studies and many others on the related research topic, we have decided to use single-cell RNA-seq of cell lines as the test sets for method comparisons, and bulk RNA-seq data of the same cell types were used to provide the background truth of the splicing changes for the performance evaluations in our initial submission.

(2) Modification of event list for benchmark

To account for potential cell cycle-related confounding effects in cell line datasets, we excluded splicing events associated with cell cycle-regulated genes from our benchmarking analysis. Specifically, we removed all events, whose harboring genes were annotated with the Gene Ontology Biological Process term 'cell cycle' (GO:0007049). The results of this refined analysis have been incorporated into the revised manuscript as **Fig. 2** and **Extended Data Fig. 11-13, 19**.

To maintain consistency with this methodological adjustment, we have correspondingly updated all relevant contextual descriptions throughout the manuscript. Specifically:

“AS events are filtered out if their harboring genes are annotated with the Gene Ontology Biological Process term 'cell cycle' (GO:0007049).” (Methods Section, Page 27, Line 706-708)

“In all cases, SCSES outperformed BRIE1, Expedition and rMATS, exhibiting an extreme increase in median SCC values across cells, ranging from 0.05~0.6, and a reduction in median RMSE by more than 0.05 across events (**Fig. 2a, 2b, Extended Data Fig. 11-13**)” (Results Section, Page 7, Line 158-161)

Modified Fig. 2. SCSES recovers the splicing levels in individual cells. **a, b.** Raincloud plots showing the performance comparison between SCSES-RC and other algorithms in real scRNA-seq data of HCT116. **(a)** The SCC value refers to the correlation between the estimated PSI values and benchmarks of all events in each cell. **(b)** The RMSE is calculated between the estimated PSI and benchmarks of an event among all cells. The events detected by both SCSES and the compared algorithm are considered for each comparison. The P -values are calculated by the Wilcoxon test. *** for P -value <0.001 . **c, d.** Bar plots showing the accuracy scores of different algorithms in HepG2 **(c)** and HCT116 **(d)** synthetic datasets. The accuracy score is defined as the product of correlation between inference and benchmarks and AS events recall rate. **e.** Comparisons of the detected DSEs between SCSES-RC with other algorithms in the real datasets. Bar plot shows the SCC of ΔPSI in DSEs from the benchmark in each comparison group. Colors represent different algorithms. The inner circle represents the difference of AUC for DSEs identification between SCSES and the compared algorithm. On the circle, *: $AUC_{SCSES} - AUC_{ref} > 0.1$, -: $AUC_{SCSES} - AUC_{ref} < 0$. **f.** Scatter plot showing the inclusion level of VPS39 exon 2 **(left panel)** and NUMB exon 12 **(right panel)** between 4 cell lines. **g.** Read coverage showing the inclusion of VPS39 exon 2 **(left panel)** and NUMB exon 12 **(right panel)** on the bulk RNA-seq of four cell lines. Alternative exons are highlighted in red.

2. Concerns about the robustness. The results indicate that for the TD+info cases, Strategy 3 (Strategy 2 + even similarity) performs worse than Strategy 2 alone. This outcome is counterintuitive, as one might expect that incorporating additional information would improve the results. This raises concerns about the robustness and reliability of the model, especially when expanding the input information.

Reply: We thank the Reviewer for this valuable comment. In SCSES, we use three imputation strategies for four types of biological scenarios of event-cell pairs. The strategies choices for different scenarios were determined based on our experimental results on cell line datasets. As shown in **Extended Data Fig. 7**, compared to other strategies, PSI values estimated by Strategy 1 in single-cell dataset showed higher Spearman correlation coefficients (SCC) with reference PSI values derived from matched bulk data in ND (non-dropout) group, which suggests that Strategy 1 is more suitable for ND group pairs. In the WD (with-dropout) group, we evaluated the percentage of true-positive event-cell pairs (where Δ PSI between reference and imputed PSI was <0.1), because the real PSI values for BD group are either zero or one. The results demonstrated Strategy 2's superior performance for both BD (biological dropout) and TD (technical dropout) groups (**Extended Data Fig. 7**). Furthermore, we observed that some event-cell pairs could not obtain information from neighbor cells when those neighbors also belonged to the TD group. We therefore tested Strategy 3's performance separately in TD+Info and TD-Info groups. The results showed Strategy 2 achieved minimal RMSE in TD+Info cases, while Strategy 3 performed better in TD-Info cases. Hence, Strategy 3 is used to PSI imputation in TD-Info group pairs. To validate the robustness of these scenario-dependent patterns, we replicated the analysis in down-sampled datasets of iPSC, hEE and nPSC, using raw PSI values as reference. Similar patterns can be found in these datasets (**Extended Data Fig. 8**).

For event-cell pairs in the TD+Info group, we investigated potential factors affecting the differential performance between Strategy 2 and Strategy 3. We stratified the event-cell pairs from single-cell datasets of four cell lines into two groups based on the PSI accuracy of their neighboring events. Neighboring events were classified as poor-quality when the absolute difference between their raw PSI in single-cell datasets and benchmark PSI from matched bulk data exceeded 0.5. TD+Info pairs with more than 20% poor-quality neighboring events were designated as "High Noise" (indicating less reliable information from neighbors), while the remaining pairs were classified as "Low Noise" (indicating more reliable neighboring information). Comparative analysis of Strategy 3's estimation errors between these groups revealed that pairs with more accurate neighboring events (Low Noise) consistently demonstrated superior performance compared to the other pairs (**Response Fig. 14a, 14c, 14e**). This finding indicates that Strategy 3's performance is associated with the quality of neighboring events, where poor-quality neighbors may introduce additional uncertainty and lead to increased imputation errors. Notably, we observed that more than half of the TD+Info pairs fell into the High Noise group across all cell line datasets (**Response Fig. 14b, 14d, 14f**), suggesting that neighboring events introduce interference in PSI estimation. In contrast, for TD-Info pairs, while limited information is available from neighboring cells, the information derived from neighboring events proves beneficial for estimation (**Extended Data Fig. 7-8**). Therefore, SCSES employs event similarity as a supplementary knowledge for TD-Info group estimation, rather than applying it to the TD+Info group where it might introduce additional noise.

In light of the reviewer's suggestion, we have added **Extended Data Fig. 7-8** into the revise Extended Data file and revised the manuscripts as follows:

"For ND cases, which have reads supporting alternative junctions, we suggest imputing the PSI value directly using a cell similarity network (**Extended Data Fig. 7-8**)."
(Results Section, Page 5-6, Line 117-118)

"And specifically, TD-Info requires an additional round of data diffusion using the event similarity network on the PSI matrix derived from previous round of imputation (**Methods, Extended Data Fig. 7-8**)."
(Results Section, Page 6, Line 129-131)

Extended Data Figure 7. Performance of different imputation strategies for different event-pair scenarios with cell similarities computed by RBP (a), PSI (b), RC (c). SCC represents the SCC between imputed PSI and benchmark PSI in cells. TP means the event-cell pairs, whose absolute differences between imputed PSI and benchmark PSI is less 0.1. The TP percentage indicates the proportion of TP pairs among BD or TD pairs. RMSE shows the root mean square error between imputed PSI and benchmark PSI in cells. Colors represent different imputation strategies. ND: non-dropout group, BD: biological dropout group, TD: technical dropout group, TD+info: technical dropout group that could obtain information from neighbor cells, TD-info: technical dropout group that could not obtain information from neighbor cells.

Extended Data Figure 8. Performance of different imputation strategies for different event-pair scenarios with cell similarities computed by RBP in iPSC, hEE, and nPSC down-sampling datasets. SCC represents the SCC between imputed PSI and benchmark PSI in cells. TP means the event-cell pairs, whose absolute differences between imputed PSI and benchmark PSI is less 0.1. The TP percentage indicates the proportion of TP pairs among BD or TD pairs. RMSE shows the root mean square error between imputed PSI and benchmark PSI in cells. Colors represent different imputation strategies. The red arrow indicates the strategy used in SCSES. ND: non-dropout group, BD: biological dropout group, TD: technical dropout group, TD+info: technical dropout group that could obtain information from neighbor cells, TD-info: technical dropout group that could not obtain information from neighbor cells.

Response Figure 14. Comparison of imputation error of Strategy 3 and the quality of neighboring events with cell similarities computed by PSI (a, b), RC (c,d), RBP (e,f). (a, c, e) The boxplot shows the PSI imputation errors of Strategy 3 for event-cell pairs in TD+Info group, grouped by the ratio of poor-quality neighboring events. Neighboring events are considered poor-quality when the absolute difference between their raw PSI in single cell dataset and benchmark PSI from matched bulk data exceeds 0.5. Groups are defined as "Low Noise" (ratio < 0.2) and "High Noise" (ratio > 0.2) based on the ratio of poor-quality neighbors. (b, d, f) The histogram shows the distribution of the ratio of poor-quality neighboring events for each event-cell pair in TD+Info group. The red dashed line indicates a ratio threshold of 0.2, which was used to classify event-cell pairs into Low and High Noise groups.

3. Furthermore, the results for Strategy 3 in the ND and BD cases are not presented, which leaves a gap in understanding how this strategy behaves across different conditions.

Reply: We thank the Reviewer for this valuable comment. We have added the performance of Strategy3 in ND and BD groups using the scRNA-seq dataset of cell lines. Compared to other strategies, PSI values estimated by Strategy 3 showed the lowest Spearman correlation coefficients (SCC) with reference PSI values derived from matched bulk data in ND (non-dropout) group. Similarly, Strategy 3 showed the lowest the percentage of true-positive event-cell pairs in the BD (biological dropout) group (**Extended Data Fig. 7**). To validate the performance of Strategy 3 in the ND and BD cases, we replicated the analysis in down-sampled datasets of iPSC, hEE and nPSC, obtaining the similar results (**Extended Data Fig. 8**).

In light of the reviewer's suggestion, **Extended Data Fig. 7-8** have been added in the revise Extended Data file.

Extended Data Figure 7. Performance of different imputation strategies for different event-pair scenarios with cell similarities computed by RBP (a), PSI (b), RC (c). SCC represents the SCC between imputed PSI and benchmark PSI in cells. TP means the event-cell pairs, whose absolute differences between imputed PSI and benchmark PSI is less 0.1. The TP percentage indicates the proportion of TP pairs among BD or TD pairs. RMSE shows the root mean square error between imputed PSI and benchmark PSI in cells. Colors represent different imputation strategies. ND: non-dropout group, BD: biological dropout group, TD: technical dropout group, TD+info: technical dropout group that could obtain information from neighbor cells, TD-info: technical dropout group that could not obtain information from neighbor cells.

Extended Data Figure 8. Performance of different imputation strategies for different event-pair scenarios with cell similarities computed by RBP in iPSC, hEE, and nPSC down-sampling datasets. SCC represents the SCC between imputed PSI and benchmark PSI in cells. TP means the event-cell pairs, whose absolute differences between imputed PSI and benchmark PSI is less 0.1. The TP percentage indicates the proportion of TP pairs among BD or TD pairs. RMSE shows the root mean square error between imputed PSI and benchmark PSI in cells. Colors represent different imputation strategies. The red arrow indicates the strategy used in SCSES. ND: non-dropout group, BD: biological dropout group, TD: technical dropout group, TD+info: technical dropout group that could obtain information from neighbor cells, TD-info: technical dropout group that could not obtain information from neighbor cells.

4. The typical final values of K in the dynamic K algorithm are not provided. The typical K values for individual event-cell groups would help clarify how many similar cells are typically required for accurate imputation, particularly for rare cell types. This could give readers a better understanding of the algorithm's performance in different scenarios.

Reply: We thank the Reviewer for this valuable comment, and apologize for the unclear description of the dynamic K algorithm. To address this concern, we would like to clarify that the K values are determined adaptively rather than predefined by the user. Moreover, the dynamic K algorithm operates at the individual cell level, assigning a specific K value to each cell based on its similarity to other cells, instead of applying a uniform K value across an entire cell type. Specifically, the dynamic K algorithm includes the following procedures: for a cell c ,

(1) sort the distances to other cells in ascending order, denoted as $dis_c^{asc} = (d_1, d_2, \dots, d_{M-1})$.

(2) compute the differences between successive distances, and generates a difference vector $df_c = (d_2 - d_1, d_3 - d_2, \dots, d_{M-1} - d_{M-2}) = (\Delta d_1, \Delta d_2, \dots, \Delta d_{M-2})$.

(3) calculate the cumulative average of each position in df_c , denoted as $cdf_c = (\frac{1}{2} \sum_{i=1}^2 \Delta d_i, \frac{1}{3} \sum_{i=1}^3 \Delta d_i, \dots, \frac{1}{M-2} \sum_{i=1}^{M-2} \Delta d_i)$.

(4) compute the average value of positive difference between df_c and cdf_c across all offset pairs, as $\bar{\Delta}_c =$

$\frac{1}{R} \sum_i^{M-3} \left(\text{relu}(\mathbf{df}_c(i+1) - \mathbf{cdf}_c(i)) \right)$, where relu is the linear rectification function and R represents the number activated by relu function.

(5) search the elements in \mathbf{cdf}_c from beginning to find the first position k_c that makes $\mathbf{df}_c(k_c) - \mathbf{cdf}_c(k_c - 1) > \bar{\Delta}_c$, where k_c is the k -value for cell c .

To characterize the typical K values in our approach, in **Extended Data Fig. 20 a, c, e**, we presented the specific K values for individual cells in the down-sampled iPSC, hEE and nPSC datasets used for evaluating the cell clustering capacity of SCSES. Notably, even cells within the same cell type exhibit varying K values. We further visualized the cell similarities between cells and their selected neighbors (**Extended Data Fig. 20 b, d, f**). The results demonstrated that the majority of neighboring cells share the same cell type as the original cell.

Extended Data Figure 20. (a, c, e) The final K value of each cell determined by dynamic K algorithm in hEE (a), iPSC(c) and nPSC (e) down-sampling datasets. Colors of point represent different cell types. (b, d, f) Heatmap showing similarity scores between cells and their selected neighbors in hEE (b), iPSC (d) and nPSC (f) down-sampling datasets. Rows represent the target cells, and columns represent their selected neighbors. The similarities with non-selected cells are set to 0.

Particularly, the dynamic K algorithm can effectively adapt to rare cell clusters. For example, there are only three oocytes in the hEE dataset, whose neighboring cells by the dynamic K algorithm are all oocytes (**Extended Data Fig. 20a, b**). Given the algorithm's ability to adaptively determine K values based on cell-specific similarity context, users do not need to predefine typical K values. This data-driven approach enhances the algorithm's robustness across diverse biological contexts.

In light of the reviewer's suggestion, we have added **Extended Data Fig. 20** into the revise Extended Data file and revised the manuscripts as follows:

"The specific K value for each cell was shown in **Extended Data Fig. 20.**" (Results section, Page 9, Line 215-216)

5. It is surprising that using cell similarity based on RBP transcript expression (which btw needs to be clarified as transcript-level expression, not protein expression), raw counts, and PSI values yields similar performance. Have the authors considered using whole transcriptome expression data or perhaps randomly selected gene expressions for comparison? This could provide experimental justification into the robustness of the method and whether alternative similarity measures might yield improved results.

Reply: We thank the Reviewer for this valuable comment. Indeed, we used transcript-level expression data of RBPs, not protein expression, for calculating cell similarity. To evaluate the robustness of our method and explore alternative similarity measures, we performed additional comparisons using whole transcriptome data and randomly selected genes matching the number of RBPs. For fair comparison, we used the top 500 most variable genes in both analyses, consistent with our approach of using top 500 RBPs in the initial submission. Following the same evaluation approach used in the initial submission, we assessed the performance of these different cell similarity measures in SCSES framework using single-cell cell line datasets. As shown in **Response Fig. 15**, splicing-based similarities used in our initial submission, specifically similarities derived from gene expression of RBPs, have better performance compared to the whole-transcriptome or random transcriptomic similarities with respect to the SCC between the predicted PSI values and benchmark PSI values of all events in cells across cell line datasets. And the RMSE between predicted PSI values and benchmark PSI values have no obvious difference among different cell similarities. In conclusion, our experiments demonstrate that cell splicing similarities in our initial submission outperformed general transcriptomic similarities in the SCSES framework. The enhanced performance of RBP-based similarity measures is consistent with their biological roles, as RBPs are critical regulators of alternative splicing. Their expression patterns closely reflect the splicing state of individual cells. In contrast, whole-transcriptome or randomly selected gene expression profiles predominantly capture general cellular states, potentially masking specific signals related to splicing regulation. By focusing on RBP gene expression, our framework leverages more splicing-relevant information, thereby improving the accuracy of PSI estimation.

In light of Review's suggestion, we have revised the manuscript as follows:

"It uses gene expressions of RNA-binding proteins (RBPs), Raw RC, or Raw PSI matrices to measure pairwise cell similarities and builds the network with adaptively optimized K values for each cell (**Methods**)."

(Results Section, Page 5, Line 102-105)

"We only used the gene expressions of RBPs, raw PSI, or raw junction read counts to calculate cell splicing similarity in current version." (Discussion Section, Page 18, Line 491-492)

Response Figure 15. (a, b) Raincloud plots showing the performance comparison between alternative similarity measures in real scRNA-seq data of cell lines. (a) The SCC value refers to the correlation between the estimated PSI values and benchmarks of all events in each cell. (b) The RMSE is calculated between the estimated PSI and benchmarks of an event among all cells. The P-values are calculated by the Wilcoxon test. N.S.: P -value >0.05 , *: P -value <0.05 , **: P -value <0.01 , ***: P -value <0.001 .

6. Due to technical limitations, many AS events, particularly those at the 5' ends of genes, are often missed in scRNA-seq data. How does the model perform when imputing AS events that are detected in bulk RNA-seq but are missing in scRNA-seq data? Specifically, how accurate is the imputation for events that are purely inferred from bulk data and not observed in the single-cell context?

Reply: We thank the Reviewer for this valuable comment. Generally, SCSES only focus on estimating the PSI values of events detected in the integrated pseudo bulk sample of scRNA-seq datasets. Due to technical limitations, some events may completely lose all supporting junction read in individual cells, even if they can be detected in the pseudo bulk sample. To assess the performance of SCSES on these events, for each cell from four cell line datasets, we selected events without any supporting junction read for evaluation. As shown in **Response Fig. 16a**, the median RMSE calculated for each cell within a single cell line decreased from approximately 0.89 to 0.25 across the four cell lines.

Furthermore, the events, which are purely inferred from bulk data and not observed in the single-cell context, lack any supporting junction reads in the entire scRNA-seq dataset. These events can not be detected from the pseudo bulk file by SCESC framework. To address the reviewer's concern, we identified splicing events that are purely inferred from bulk data and not observed in the single-cell context across four cell lines. These events were artificially added to the event list detected by SCSES from the single-cell datasets of four cell lines. We then estimated the splicing levels of these additional events at the single-cell level and evaluated the performance of SCSES on them accordingly. As demonstrated in **Response Fig. 16b**, the median RMSE for each event decreased from approximately 0.9 to 0.3 across four cell lines, indicating that SCSES has some capacity to handle these events.

Response Figure 16. (a) Performance evaluation of SCSES on splicing events detected in the pseudo bulk sample but showing no junction read evidence at single-cell level across four cell lines. The RMSE comparing benchmarks and raw PSI values (first column) or SCSES estimated PSI values (columns 2-4), calculated for each cell from four cell lines. (b) Performance evaluation of SCSES on splicing events that are purely inferred from bulk data and lack any supporting reads in the entire scRNA-seq dataset. The RMSE comparing benchmarks and raw PSI values (first column) or SCSES estimated PSI values (columns 2-4), calculated for each event from four cell lines. The P-values are calculated by the Wilcoxon test. N.S.: P -value >0.05 , *: P -value <0.05 , **: P -value <0.01 , ***: P -value <0.001 .

7. The Number of Junction Reads for PSI Estimation. For the raw PSI matrix, are the number of junction reads taken into account during PSI estimation? The accuracy of PSI estimation is highly dependent on the number of reads. For example, if we assume that read assignment to inclusion and exclusion follows a binomial distribution (where n is the number of reads and p is the true PSI), then with 10 reads and a true PSI of 0.5, the 95% confidence interval for the estimated PSI would be [0.2, 0.8]. This demonstrates the significant variability in PSI estimation with low read counts. Did the authors incorporate this variability into their model, particularly when imputing PSI for non-dropout events, where the junction matrix is not used (Strategy 1)? Clarification on this aspect would greatly enhance the understanding of the model's reliability and robustness.

Reply: We thank the Reviewer for this valuable comment. To integrate the junction read counts into our model, we modified the Strategy 1 as follows:

- (1) Calculate cell distance matrix Dis based on RBP expression, junction read counts or raw PSI values;
- (2) Determine the K value for each cell by dynamic K algorithm, and obtain the KNN distance matrix Dis_{KNN} ;
- (3) Transfer the KNN distance matrix to KNN similarity matrix S_{cell} , and perform the network diffusion to obtain diffused similarity network \hat{S}_{cell} ;
- (4) For each event-cell pair, calculate the width of 95% Wilson confidence interval of PSI value, and obtain the confidence interval width matrix CIL . Specifically, assume the read count distribution follows a

binomial distribution (as Reviewer indicated), for an event-cell pair with I inclusion junction read counts and E exclusion junction read counts, the Wilson confidence interval is:

$$\left[\frac{\hat{p} + \frac{z_{\alpha/2}^2}{2n}}{1 + \frac{z_{\alpha/2}^2}{n}} - \frac{z_{\alpha/2}}{1 + \frac{z_{\alpha/2}^2}{n}} \sqrt{\frac{\hat{p}(1-\hat{p})}{n} + \frac{z_{\alpha/2}^2}{4n^2}}, \frac{\hat{p} + \frac{z_{\alpha/2}^2}{2n}}{1 + \frac{z_{\alpha/2}^2}{n}} + \frac{z_{\alpha/2}}{1 + \frac{z_{\alpha/2}^2}{n}} \sqrt{\frac{\hat{p}(1-\hat{p})}{n} + \frac{z_{\alpha/2}^2}{4n^2}} \right],$$

where $\hat{p} = \frac{I}{I+E}$, $n = I + E$, $z_{\alpha/2} \approx 1.96$ is the 97.5% quantile of the standard normal distribution. Then

the width of 95% Wilson confidence interval of PSI values is $\frac{2z_{\alpha/2}}{1 + \frac{z_{\alpha/2}^2}{n}} \sqrt{\frac{\hat{p}(1-\hat{p})}{n} + \frac{z_{\alpha/2}^2}{4n^2}}$,

- (5) Modify the cell similarity for each event-cell pair by integrating the original cell similarity and the PSI confidence score of each event in neighbor cells. Specifically, for cell c and event e , the original cell similarity vector of cell c with neighbor cells is $\hat{S}_{cell}(c, \cdot)$ and the width vector of PSI 95% confidence interval of event e is $CIL(e, \cdot)$. The PSI confidence score is defined as the reciprocal of the confidence interval width. Then the modified cell similarity of cell c for event e is $S_{modify}^{c,e} = \hat{S}_{cell}(c, \cdot) \odot \frac{1}{CIL(e, \cdot)}$,

where \odot represents the element-wise product of two vectors. Finally, the sum of $S_{modify}^{c,e}$ is scaled to 1.

- (6) Imputation with the modified cell similarity for each event. Specifically, for cell c and event e , the raw PSI values in all cells is defined as $PSI_{raw}(e, \cdot)$, then the imputed PSI is calculated as $PSI_{imp}(e, c) =$

$$\sum_{k \in Neighbor(c)} PSI_{raw}(e, k) * S_{modify}^{c,e}(k).$$

We compared the performance of SCSES before and after Strategy 1 modification with respect to the PSI accuracy on cell line datasets. Our results suggested that both approaches showed comparable performance in terms of Spearman correlation coefficients (SCC) between imputed versus reference PSI values among cells and RMSE among events across all four cell line datasets (**Response Fig. 17**).

Response Figure 17. Raincloud plots showing the performance comparison between raw and revised model. (a) The SCC value refers to the correlation between the estimated PSI values and benchmarks across all events in each cell of the four cell lines. (b) The RMSE between estimated PSI values and benchmarks is calculated across all cells from four cell lines for each event. The P-values are calculated by the Wilcoxon test. (c) Bar plot of NMI across three test datasets. For each dataset, down-sampling is replicated three times. N.S.: P -value >0.05 , *: P -value <0.05 , **: P -value <0.01 , ***: P -value <0.001 .

Even though modified Strategy 1 did not significantly improve the performance of SCSES, but it is a great idea to take the number of junction reads into account during PSI estimation of Strategy 1. The PSI values with higher confidence can provide more accurate splicing information for the PSI estimation. But it's still a challenge to find a proper way to integrate the confidence into our model, which will be a new direction in our next investigation.

In light of review's valuable suggestion, we have added a discussion about this point in the revised manuscript:

"In the current SCSES Strategy 1, raw PSI values are used directly. However, the reliability of a PSI value largely depends on its supporting junction read counts. PSI values with higher reliability provide more accurate information for imputing splicing profiles in similar cells. Therefore, incorporating PSI confidence scores into SCSES could enhance the accuracy of the estimated PSI values." (Discussion Section, Page 18, Line 485-489)

8. Clustering Parameters. The same set of parameters was applied to both gene expression and splicing profiles to derive cell clusters. It would be valuable to explore whether tuning the parameters specifically for the gene expression profile could yield cell clusters similar to those identified using the splicing profile, such as SC1-SC4 in the multiple myeloma dataset. Notably, SC1-SC4 exhibit differential expression in certain genes, suggesting that adjustments to clustering parameters might enable the recovery of SC1-SC4 clusters even when using the gene expression profile alone. In such a scenario, the uniqueness of splicing-based clustering in detecting these subgroups would need to be re-evaluated. It would be beneficial for the authors to perform additional clustering experiments with parameter optimization tailored separately for gene expression and splicing profiles. This could provide stronger evidence to support their claim that splicing-based clustering uncovers novel cell subgroups that cannot be detected through gene expression clustering alone.

Reply: We thank the Reviewer for this valuable suggestion. To address the reviewer's concerns, we conducted parameter optimization experiments and performed comparisons between gene expression-based and RNA splicing-based clustering.

The SC1-SC4 clusters were identified by Louvain algorithm following the Seurat pipeline. To determine the optimal clustering parameters, we systematically evaluated multiple parameter combinations, including six different numbers of high-variable features (1,000, 2,000, 3,000, 5,000, 8,000, and 10,000) and nine distinct clustering resolutions (0.1, 0.3, 0.5, 0.8, 1, 2, 3, 4, and 5). Clustering quality was measured using cluster modularity³², a metric that quantifies the strength of community structure within the network. Cell clusters with high modularity have dense connections between the cells within clusters but sparse connections between cells in different clusters, indicating a high-quality clustering result. For RNA splicing-based clustering, we achieved maximal modularity using 5,000 high-variable splicing events and a resolution of 1 (**Response Fig. 18, right panel**). Gene expression-based clustering showed optimal performance with 3,000 high-variable genes and a resolution of 1 (**Response Fig. 18, left panel**).

Response Figure 18. Line plots showing the change in cluster modularity with respect to clustering resolution for different numbers of high-variable features in expression-based clustering (**left panel**) and PSI-based clustering (**right panel**).

To assess the capacity of gene expression-based clustering in recapitulating the previously identified SC1–SC4 subgroups from RNA splicing-based analysis, we conducted a comparison of the clustering results using the same parameter combinations as in our optimization analysis above. To measure the consistence between the cell subgroups in our initial submission (SC1-SC4) and gene expression-based clustering results, we calculated the normalized mutual information (NMI), ranging from 0 to 1 that measures the degree of agreement between two clustering assignments, where higher values indicate a high degree of clustering similarity. All parameter combinations can separate cells sampling from initial to recurrent tumor. However, none of parameter combinations can recapitulate the SC1-SC4 clusters in the initial submission, where SC2 is the novel cell groups identified by SCSES-estimated splicing levels. These results validate that splicing-based clustering uncovers novel cell subgroups that cannot be detected through gene expression clustering alone (**Response Fig 19 – Fig 24**).

Response Figure 19. UMAP plots showing the cell clustering based on expression of top 1000 high-variable genes with respect to different resolutions. Colors in the first panel represents the SC1-SC4 clusters from initial submission, which is shown as reference. Colors in other panels represent cell clusters with respect to different resolutions. Bone marrow indicates the primary diagnosis samples. Ascites indicates the metastasis samples.

Response Figure 20. UMAP plots showing the cell clustering based on expression of top 2000 high-variable genes with respect to different resolutions. Colors in the first panel represents the SC1-SC4 clusters from initial submission, which is shown as reference. Colors in other panels represent cell clusters with respect to different resolutions. Bone marrow indicates the primary diagnosis samples. Ascites indicates the metastasis samples.

Response Figure 21. UMAP plots showing the cell clustering based on expression of top 3000 high-variable genes with respect to different resolutions. Colors in the first panel represents the SC1-SC4 clusters from initial submission, which is shown as reference. Colors in other panels represent cell clusters with respect to different resolutions. Bone marrow indicates the primary diagnosis samples. Ascites indicates the metastasis samples.

Response Figure 22. UMAP plots showing the cell clustering based on expression of top 5000 high-variable genes with respect to different resolutions. Colors in the first panel represents the SC1-SC4 clusters from initial submission, which is shown as reference. Colors in other panels represent cell clusters with respect to different resolutions. Bone marrow indicates the primary diagnosis samples. Ascites indicates the metastasis samples.

Response Figure 23. UMAP plots showing the cell clustering based on expression of top 8000 high-variable genes with respect to different resolutions. Colors in the first panel represents the SC1-SC4 clusters from initial submission, which is shown as reference. Colors in other panels represent cell clusters with respect to different resolutions. Bone marrow indicates the primary diagnosis samples. Ascites indicates the metastasis samples.

Response Figure 24. UMAP plots showing the cell clustering based on expression of top 10000 high-variable genes with respect to different resolutions. Colors in the first panel represents the SC1-SC4 clusters from initial submission, which is shown as reference. Colors in other panels represent cell clusters with respect to different resolutions. Bone marrow indicates the primary diagnosis samples. Ascites indicates the metastasis samples.

9. Tool Evaluation and Benchmarking. Please consider testing Psix in scRNA-seq of multiple myeloma (Extended Figure 12C) and hESC (Extended Figure 15D), as it appears to perform well and may provide additional insights or validation.

Reply: We thank the reviewer for their valuable comment. We used Psix to estimate the PSI values in multiple myeloma dataset and hESC dataset and generated UMAP plots based on estimated PSI values. As shown in **Response Fig. 25**, the splicing profiles inferred by Psix were able to separate primary diagnosis cells (SC1 and SC2) from relapse cells (SC3 and SC4), but failed to resolve subclusters within SC1–SC4. Similarly, in the hESC dataset, the splicing profile inferred by Psix showed limited capacity to identify the SC3 subgroup. Moreover, Psix completely failed to distinguish between SC1 and SC2—a novel discovery uniquely captured by SCSES.

Response Figure 25. UMAP plots showing the cell projections of MM cells (a) and developing 36h cells (b) using different AS inferring algorithms. Colors represent the cell clusters defined by SCSES.

In light of Reviewer's suggestion, we have added **Extended Data Fig. 22c** and **Extended Data Fig. 27d** into the revised Extended Data file, respectively.

Extended Data Figure 22c. UMAP plots showing the cell projections of MM cells using different AS inferring algorithms. Colors represent the cell clusters defined by SCSES.

Extended Data Figure 27d. UMAP plots showing the cell projections of developing 36h cells using different AS inferring algorithms. Colors represent the cell clusters defined by SCSES.

10. Formula Clarification for SCC. The formula presented in Line 627 appears to compute the Pearson correlation coefficient rather than the Spearman correlation coefficient. As these two measures assess different types of relationships—Pearson for linear relationships and Spearman for monotonic relationships—it is crucial to ensure the correct term is used to avoid confusion. If the formula is indeed calculating the Pearson correlation coefficient, please revise the description accordingly. Alternatively, if the intention was to compute the Spearman correlation coefficient, kindly provide the correct formula and verify whether the calculations throughout the manuscript reflect this adjustment. Clarifying this point will enhance the accuracy and credibility of the reported results.

Reply: We thank the reviewer for their valuable comment, and we sincerely apologize for our carelessness. It should be Spearman correlation coefficient rather than Pearson correlation coefficient in this part. We have corrected the description of the definition formula in the revised manuscript as follows:

“The SCC between the imputed PSI and benchmark PSI in cell c is defined as:

$$SCC_c = 1 - \frac{6 \sum_{i=1}^n (IPR_c(i) - BPR_c(i))^2}{n(n^2 - 1)}$$

where $IPR_c(i)$ and $BPR_c(i)$ are the imputed PSI rank and benchmark PSI rank of event i in cell c , respectively, n is the event count.” (Methods Section, Page 27, Line 711-715)

11. Alternative Last Exon. The alternative last exons of Lamp2 are detected by SCSES (Figure 6F), yet the

methods section specifies that only five classical splicing events (SE, A5SS, A3SS, MXE, and RI) are analyzable by SCSES. This raises the question of whether SCSES is capable of detecting alternative last exons as well. Furthermore, does SCSES also support the detection of alternative first exons? If SCSES indeed identifies these additional splicing events, it would be helpful to clarify this capability in the manuscript, including any modifications to the underlying model or algorithms that enable this functionality. Additionally, if SCSES does not currently analyze these events, it would be valuable to explain how the detection of the Lamp2 alternative last exon was achieved and whether this was an exception or a general feature of the method. Providing such clarifications would improve the transparency and completeness of the methodological framework.

Reply: We thank the Reviewer for this valuable comment and apologize for the unclear description regarding alternative last exon events. SCSES is currently designed to estimate the splicing levels of five classical splicing events (SE, A5SS, A3SS, MXE, and RI), and does not support alternative first/last exon events (ALE/AFE) in this version. In the inDrop-seq dataset, due to the technical limitation of 3' bias, we were able to detect only a limited number of classical splicing events. Specifically, only 2,635 events passed quality control in the inDrop-seq dataset, compared to over 10,000 events in the multiple myeloma and embryonic stem cell development datasets, respectively. We observed that, in addition to A3SS/A5SS events, MAJIQ is also capable of detecting alternative last exon events, which can be identified in 3'-biased sequencing datasets. To expand the splicing events pool in the inDrop-seq dataset, we attempted to incorporate these events into our analysis. As a result, we identified 343 ALE events using MAJIQ in this dataset. To adapt these ALE events to the SCSES framework, we extracted the ALE sequence features based on the structure of ALE (**Extended Data Fig. 41, Supplementary Table 11**). Without changing other procedures, we estimated the splicing levels of these ALE events and performed the analysis as shown in the initial submission.

Our results with the inDrop-seq dataset demonstrate that SCSES can estimate the splicing levels of novel event types with predefined event sequence features. However, the analysis of ALE in SCSES is still in early stage. The current version of the released package supports the estimation of only five classical event types. We believe that further development to support ALE, as well as AFE, could broaden the scope of SCSES in future versions.

Extended Data Figure 41. The event structure (a) and PSI definition (b) of ALE events.

Event Types	Category	Description
ALE	Length	Length of region C1, C2, A, and I
		length ratio of A/C1, A/C2, A/I
	Motif	Regions near 3'/5' splicing site of region A and I, 16nt upstream to 4nt downstream for 3' region, 4nt upstream to 6nt downstream for 5' region
	Conservation	Conservation scores of regions near 3'/5' of region A and I
	k-mer	1-3mer for region C1, C2, and regions near 3'/5' splicing site 1-4mer for region A
Adenine ratio	Adenine ratio in region A and I	

Supplementary Table 11. Event features of ALE events to calculate event similarity network.

In light of Reviewer's suggestion, we have added the **Extended Data Fig. 41** and **Supplementary Table 11** into the revised Extended Data file and Supplementary Tables file, respectively. We also revised the manuscript and Supplementary Note as follows:

“To expand the splicing events pool in the inDrop-seq dataset, we added the alternative last exon events detected by MAJIQ into our analysis (**Supplementary Note**). Totally, 2978 valid AS events were detected and quantified.” (Results Section, Page 15, Line 405-408)

“To adapt these ALE events to the SCSES framework, we extracted the ALE sequence features based on the structure of ALE (**Extended Data Fig. 41, Supplementary Table 11**). Without changing other procedures, we estimated the splicing levels of these ALE events.” (Supplementary Note Section, Page 9, Line 195-198)

Minor Comments:

1. Typographical Errors

Please address the following typos:

- "SCSEC" (Line 988) should be corrected to "SCSES".
- "SCSECI" (Line 429) should be corrected to "SCSES".
- Correct "Spearman correlation confidence" (Lines 135-136) to "Spearman correlation coefficient".

Reply: We thank the Reviewer for this valuable comment and apologize for our carelessness. We have revised the manuscript as follows:

“SCSES introduces three imputation strategies for different event-cell groups.” (Figure1 Caption, Page 44, Line 1119-1120)

“Furthermore, despite the good performance of SCSES on droplet-based data, it is still difficult to estimate the intensities of splicing events that are away from the 3'-end of transcripts.” (Discussion Section, Page 19, Line 502-504)

“In all cases, SCSES outperformed BRIE1, Expedition and rMATS, exhibiting an extreme increase in median SCC values across cells, ranging from 0.05~0.6, and a reduction in median RMSE by more than 0.05 across events” (Results Section, Page 7, Line 158-161)

2. Line 103 “four different strategies“

Update the text to "three different strategies" for correctness.

Reply: We thank the Reviewer for this valuable comment. We have revised the manuscript as follows:

“Based on our practice, we recommend three different data imputation strategies, matched to four types of biological scenarios, which are defined by the abundance of alternative junctions in both the target cell and its neighboring cells (**Fig. 1c, Extended Data Fig. 6**).” (Results Section, Page 5, Line 112-115)

3. Hyperparameter Explanations

Please provide details on how specific hyperparameters were determined, such as the restart probability in the Random Walk with Restart (RWR) algorithm and the k value for diffused similarity matrix. Including these details would greatly enhance the reproducibility and transparency of the methodology.

Reply: We thank the Reviewer for this valuable comment. The hyperparameters were determined through evaluation of multiple candidates, where their performance was assessed based on PSI estimation accuracy and clustering accuracy. To evaluate the accuracy of PSI estimation, we calculated the Spearman correlation coefficients (SCC) between the PSI values estimated by SCSES and the benchmark PSI values (derived

from bulk RNA-seq of the matched cell type) across all events within each cell in four single-cell datasets of cell lines. To evaluate the accuracy of cell clustering, we calculated the normalized mutual information (NMI) between the K-means clustering results derived from the estimated PSI profiles and the cell type annotations provided in the original publications of the test datasets across down-sampled datasets of iPSC, hEE and nPSC.

For the restart probability (λ), we tested six candidates (0, 0.1, 0.2, 0.4, 0.6, 0.8). As shown in **Extended Data Fig. 35**, the PSI estimation accuracy remained stable when λ was larger than 0.1, while the clustering accuracy achieved better results when λ was set to 0.2. After comprehensive evaluation across multiple datasets, we set $\lambda = 0.2$ as our default parameter.

For the K value for the diffused similarity matrix, the K values of cell similarities are automatically determined by the dynamic K algorithm for each cell. We further tested the effect of different K values of event similarities by evaluating six candidates (5, 10, 15, 20, 30, 50). As shown in **Extended Data Fig. 34**, the K values of event similarities had no obvious influence on the PSI estimation accuracy, while the clustering accuracies exhibited moderate fluctuations. After comprehensive evaluation across multiple datasets, we set K=10 as our default parameter.

For the convergence thresholds (decay) of random walk, we tested seven candidates (Inf, 0.5, 0.1, 0.05, 0.01, 0.001, 0.0001), where Inf means no diffusion procedure was applied. As shown in **Extended Data Fig. 36**, the PSI estimation accuracy increased as the decay decreased, and became relatively stable after 0.1. The clustering accuracies showed no consistent patterns across the three datasets. After comprehensive evaluation across multiple datasets, we set decay=0.05 as our default parameter.

It is worth noting that all these hyperparameters are customizable in the SCSES package. Users can test with different hyperparameter combinations to select the most suitable ones according to their specific situation.

In light of the reviewer's suggestion, we have added the corresponding results into the revised Extended Data file (Supplementary Note Section, Page 11, Line 260-292), and related figures have been added into the revised Extended Data file as **Extended Data Fig. 34-36**. The revised context in revised manuscript is as follows:

“SCSES keeps distances of nearest events with fixed k_{event} (10 by default, **Extended Data Fig. 34, Supplementary Note**)” (Methods Section, Page 22, Line 581-582)

“where λ is the restart probability (0.2 by default, **Extended Data Fig. 35, Supplementary Note**).” (Methods Section, Page 23, Line 595-596)

“The walk is stopped when $\Delta(t)$ is less than a certain threshold, which is 0.05 in SCSES by default (**Extended Data Fig. 36, Supplementary Note**).” (Methods Section, Page 23, Line 601-603)

Extended Data Figure 35. Performance evaluation of random walk algorithm under different convergence thresholds (Decay). **(a)** SCC among cells in four cell line datasets. The points indicate median SCC in all cells across cell lines. **(b)** NMI in down-sampling datasets. Note: "Inf" indicates no diffusion procedure was applied. The points indicate median NMI from three independent replicates.

Extended Data Figure 34. Performance evaluation of event similarity computation across different K values. **(a)** SCC among cells in four cell line datasets. The points indicate median SCC in all cells across cell lines. **(b)** NMI in down-sampling datasets. The points indicate median NMI from three independent replicates.

Extended Data Figure 36. Performance evaluation of random walk algorithm under different convergence thresholds (Decay). (a) SCC among cells in four cell line datasets. The points indicate median SCC in all cells across cell lines. (b) NMI in down-sampling datasets. Note: "Inf" indicates no diffusion procedure was applied. The points indicate median NMI from three independent replicates.

4. Line 548–549: Imputation Strategy Iterations

The manuscript states: "In SCSES, each imputation strategy is executed multiple times until the difference between two runs is less than 0.05." Could you clarify what is meant by "the difference" in this context? For example, does it refer to the difference in imputed PSI values, convergence of a loss function, or another metric? Providing this information would improve the clarity of the methods description.

Reply: We thank the Reviewer for this valuable comment. It means the imputation procedure utilizes an iterative approach, continuing until the change between the imputed PSI matrices at time t and time $t-1$ meets a convergence threshold. This threshold is defined using the same loss function employed in network diffusion. After each iteration, we calculate the change in the PSI matrix by

$$\Delta(t) = \frac{SSE(\mathbf{PSI}^t, \mathbf{PSI}^{t-1})}{SST(\mathbf{PSI}^t, \mathbf{PSI}^{t-1})}$$

where SSE is the sum of squares error and SST is the total sum of squares. The imputation procedure terminates when $\Delta(t)$ falls below a predefined convergence threshold (0.05, by default).

To clarify the description, we have revised the manuscript as follows:

"For each imputation strategy, the imputation procedure is performed iteratively until the change between the imputed PSI matrices of two consecutive steps meets a convergence threshold. The loss function is identical to that used in network diffusion, with a default threshold value of 0.05." (Methods Section, Page 24, Line 627-630)

5. The concept of information sharing through data diffusion has been reported in multiple publications on scRNA-seq analysis, primarily in the context of gene expression quantification rather than splicing quantification. The authors can cite these publications to enhance the validity of their work.

Reply: We thank the Reviewer for this valuable comment. We have added the relative citations in the revised manuscript as follows:

“Here, we present Single Cell Splicing ESTimation algorithm (SCSES), a network diffusion-based imputation method designed to accurately recover splicing changes across main types of splicing events at the single-cell level. SCSES is inspired by the data diffusion technique, which is widely applied in scRNA-seq data analysis, such as MAGIC³¹, DTFLOW³² and PHATE³³” (Introduction Section, Page 4, Line 71-75)

Reference

- Chu, L. F. *et al.* Single-cell RNA-seq reveals novel regulators of human embryonic stem cell differentiation to definitive endoderm. *Genome Biol* **17**, 173, doi:10.1186/s13059-016-1033-x (2016).
- Yuan, F., Hankey, W., Wagner, E. J., Li, W. & Wang, Q. Alternative polyadenylation of mRNA and its role in cancer. *Genes Dis* **8**, 61-72, doi:10.1016/j.gendis.2019.10.011 (2021).
- Hou, W., Ji, Z., Ji, H. & Hicks, S. C. A systematic evaluation of single-cell RNA-sequencing imputation methods. *Genome Biol* **21**, 218, doi:10.1186/s13059-020-02132-x (2020).
- Huang, Y. & Sanguinetti, G. BRIE: transcriptome-wide splicing quantification in single cells. *Genome Biol* **18**, 123, doi:10.1186/s13059-017-1248-5 (2017).
- Jang, J. S. *et al.* Molecular signatures of multiple myeloma progression through single cell RNA-Seq. *Blood Cancer J* **9**, 2, doi:10.1038/s41408-018-0160-x (2019).
- Andreatta, M. & Carmona, S. J. UCell: Robust and scalable single-cell gene signature scoring. *Comput Struct Biotechnol J* **19**, 3796-3798, doi:10.1016/j.csbj.2021.06.043 (2021).
- Needle, M. N. *et al.* The Multiple Myeloma Research Foundation (MMRF) CoMMpassSM Study: A Longitudinal Study in Newly-Diagnosed Multiple Myeloma Patients to Assess Genomic Profiles, Immunophenotypes and Clinical Outcomes. *Blood* **120**, 3980, doi:<https://doi.org/10.1182/blood.V120.21.3980.3980> (2012).
- Mulligan, G. *et al.* Gene expression profiling and correlation with outcome in clinical trials of the proteasome inhibitor bortezomib. *Blood* **109**, 3177-3188, doi:10.1182/blood-2006-09-044974 (2007).
- Shi, L. *et al.* The MicroArray Quality Control (MAQC)-II study of common practices for the development and validation of microarray-based predictive models. *Nat Biotechnol* **28**, 827-838, doi:10.1038/nbt.1665 (2010).
- Hanzelmann, S., Castelo, R. & Guinney, J. GSEA: gene set variation analysis for microarray and RNA-seq data. *BMC Bioinformatics* **14**, 7, doi:10.1186/1471-2105-14-7 (2013).
- Liao, Y., Smyth, G. K. & Shi, W. featureCounts: an efficient general purpose program for assigning sequence reads to genomic features. *Bioinformatics* **30**, 923-930, doi:10.1093/bioinformatics/btt656 (2014).
- Sebestyen, E. *et al.* Large-scale analysis of genome and transcriptome alterations in multiple tumors unveils novel cancer-relevant splicing networks. *Genome Res* **26**, 732-744, doi:10.1101/gr.199935.115 (2016).
- Li, J. *et al.* SFMetaDB: a comprehensive annotation of mouse RNA splicing factor RNA-Seq datasets. *Database (Oxford)* **2017**, doi:10.1093/database/bax071 (2017).
- Cook, K. B., Kazan, H., Zuberi, K., Morris, Q. & Hughes, T. R. RBPDB: a database of RNA-binding specificities. *Nucleic Acids Res* **39**, D301-308, doi:10.1093/nar/gkq1069 (2011).
- Bradley, R. K. & Anczukow, O. RNA splicing dysregulation and the hallmarks of cancer. *Nat Rev Cancer* **23**, 135-155, doi:10.1038/s41568-022-00541-7 (2023).
- Buen Abad Najar, C. F., Yosef, N. & Lareau, L. F. Coverage-dependent bias creates the appearance of binary splicing in single cells. *Elife* **9**, doi:10.7554/eLife.54603 (2020).
- Akter, M. & Ding, B. Modeling Movement Disorders via Generation of hiPSC-Derived Motor Neurons. *Cells* **11**, doi:10.3390/cells11233796 (2022).

- Akter, M., Cui, H., Sepehrimanesh, M., Hosain, M. A. & Ding, B. Generation of highly pure motor neurons from human induced pluripotent stem cells. *STAR Protoc* **3**, 101223, doi:10.1016/j.xpro.2022.101223 (2022).
- Sepehrimanesh, M. & Ding, B. Generation and optimization of highly pure motor neurons from human induced pluripotent stem cells via lentiviral delivery of transcription factors. *Am J Physiol Cell Physiol* **319**, C771-C780, doi:10.1152/ajpcell.00279.2020 (2020).
- Gulati, G. S. *et al.* Single-cell transcriptional diversity is a hallmark of developmental potential. *Science* **367**, 405-411, doi:10.1126/science.aax0249 (2020).
- Cao, J. *et al.* The single-cell transcriptional landscape of mammalian organogenesis. *Nature* **566**, 496-502, doi:10.1038/s41586-019-0969-x (2019).
- Nicklas, S. *et al.* The RNA helicase DDX6 regulates cell-fate specification in neural stem cells via miRNAs. *Nucleic Acids Res* **43**, 2638-2654, doi:10.1093/nar/gkv138 (2015).
- Shen, S. *et al.* MATS: a Bayesian framework for flexible detection of differential alternative splicing from RNA-Seq data. *Nucleic Acids Res* **40**, e61, doi:10.1093/nar/gkr1291 (2012).
- Leek, J. T., Johnson, W. E., Parker, H. S., Jaffe, A. E. & Storey, J. D. The sva package for removing batch effects and other unwanted variation in high-throughput experiments. *Bioinformatics* **28**, 882-883, doi:10.1093/bioinformatics/bts034 (2012).
- Qiao, L. *et al.* LAMP2A, LAMP2B and LAMP2C: similar structures, divergent roles. *Autophagy* **19**, 2837-2852, doi:10.1080/15548627.2023.2235196 (2023).
- Gough, N. R., Hatem, C. L. & Fambrough, D. M. The family of LAMP-2 proteins arises by alternative splicing from a single gene: characterization of the avian LAMP-2 gene and identification of mammalian homologs of LAMP-2b and LAMP-2c. *DNA Cell Biol* **14**, 863-867, doi:10.1089/dna.1995.14.863 (1995).
- Zhang, Y. *et al.* Alternative polyadenylation: methods, mechanism, function, and role in cancer. *J Exp Clin Cancer Res* **40**, 51, doi:10.1186/s13046-021-01852-7 (2021).
- Di Giandomartino, D. C., Nishida, K. & Manley, J. L. Mechanisms and consequences of alternative polyadenylation. *Mol Cell* **43**, 853-866, doi:10.1016/j.molcel.2011.08.017 (2011).
- Batra, R. *et al.* Loss of MBNL leads to disruption of developmentally regulated alternative polyadenylation in RNA-mediated disease. *Mol Cell* **56**, 311-322, doi:10.1016/j.molcel.2014.08.027 (2014).
- Sala Frigerio, C. *et al.* The Major Risk Factors for Alzheimer's Disease: Age, Sex, and Genes Modulate the Microglia Response to Abeta Plaques. *Cell Rep* **27**, 1293-1306 e1296, doi:10.1016/j.celrep.2019.03.099 (2019).
- Zhang, Y. *et al.* Definitive Endodermal Cells Supply an in vitro Source of Mesenchymal Stem/Stromal Cells. *Commun Biol* **6**, 476, doi:10.1038/s42003-023-04810-5 (2023).
- Newman, M. E. Modularity and community structure in networks. *Proc Natl Acad Sci U S A* **103**, 8577-8582, doi:10.1073/pnas.0601602103 (2006).

Point-by-point response to Reviewers

We thank the reviewers for their constructive and detailed evaluation of our manuscript. Their insightful comments have helped to improve our work. We have addressed each point raised in our revised manuscript as follows:

For a better reading of our responses, we marked the words in different colors as follows:

Reviewer comments are marked in **black font**

Responses are marked in **blue font**

New descriptions added to the revised manuscript are marked in **red font**

The location of added descriptions in the revised manuscript is marked in **green font**

Reviewer #1 (Remarks to the Author):

I thank the authors for conducting a comprehensive revision of the manuscript. However, I still have some concerns:

Reply: We thank the reviewer for acknowledging our comprehensive revision of the manuscript.

1. One primary concern is that, in spite of the substantial revisions requested, the authors chose not to modify any panels within the main figures. This absence of visual integration significantly detracts from the revision's perceived strength, inadvertently communicating that the extensive additional analyses, benchmarks, and results are not sufficiently critical or robust to warrant inclusion in the central presentation of their work.

Reply: We thank the reviewer for this valuable comment. In light of the Reviewer's suggestion, we modified the main figures by incorporating key results from two rounds of revision as follows:

(1). In **Figure 2**, we have added the new evaluations of SCSES on single cell datasets with paired short-read and long-read sequencing (new results from below Comment 2 and Comment 3) as new **Figure 2d**.

(2). In **Figure 3**, we have replaced panels **3e–g** with heatmaps displaying the differentially spliced events between MN-C1 and MN-C2, along with the corresponding gene expression profiles. Additionally, we added plots illustrating the distinct splicing patterns of *VPS29* between MN-C1 and MN-C2.

(3). In **Figure 4**, we have added the plots showing the survival differences between patients grouped by the expression levels of MM34-SC2 marker genes in three independent cohorts (MMRF, GSE9782 and GSE24080) as **Figure 4k-4l**.

In light of the reviewer's comments, we have revised the figure legend in the manuscript and added the following description:

[revised manuscript text omitted]

- 2. Benchmark in real scenarios. The authors included one additional benchmark from 174 iPSC-derived
cells, sequenced with C1 + SMARTer-seq. I would argue this is still a scenario where the heterogeneity
is limited with respect, for example, to a real tissue. Furthermore, the dataset is from 2016/2017 and
limited in the number of cells with respect to standard single-cell experiments. In summary, the additional
benchmark dataset fails to be an ideal representative of a single-cell dataset that could be analyzed with
the proposed tool.

**Reply:** We thank the reviewer for this valuable comment. We fully agree that benchmarking in real tissue
samples is more biological meaningful, though more challenging for method benchmarking. The main
difficulty lies in establishing a reliable ground truth for splicing quantification in complex tissues. Due to cellular
heterogeneity, a reliable benchmark would ideally require paired sequencing the same individual cell by using
two protocols—one with a specific sequencing method for alternative splicing detection and quantification to
serve as a reference, and the other using a commonly adopted single cell sequencing technique for method
evaluation.

To assess the performance of SCSES in biologically relevant tissue with sufficient cell numbers and inherent
heterogeneity, and in light of Reviewer#1 comment 3, we removed the results of iPSC in last round of revision,
and used two datasets in which each cell was profiled using paired short-read and long-read sequencing
technologies. Specifically, these two datasets are:

- ● Ovarian cancer sample: This dataset includes 2,174 cells derived from an ovarian cancer patient¹,
consisting of 8 cell types, and each cell is sequenced by both scTalLoR-seq (long-read) and 10x
Genomics (short-read).
- ● Human hippocampus sample: This dataset includes 8,681 cells², consisting of 16 cell types, and each
sequenced by ScISOr-Seq2 (long-read) and 10x Genomics (short-read).

The evaluation procedure is shown in **Response Fig. 1**. Specifically, we estimated PSI values from the short-
read data using SCSES and other methods, and used the PSI values derived from the long-read data as the
ground truth. As the short-read data were generated using a UMI-based protocol (which most compared tools
do not natively support), we applied a deduplication strategy: for each UMI, we retained one representative
read per barcode. To maximize the retention of splicing-relevant reads, we prioritized junction reads identified
by the presence of an 'N' operation in the BAM CIGAR string.

Response Fig. 1: The scheme for evaluating SCSES performance in paired short-read and long-read datasets

The splicing events detected by different methods were significantly varied, and most methods only considered exon skipping events (SE). To ensure a fair comparison, we used only the SE events identified by both SCSES and each individual method under comparison. High-confidence event-cell pairs from the long-read data were used as the benchmark; these pairs were defined as splicing events supported by more than 10 long-read molecules in a given cell. Reference PSI values were computed based on the ratio of long-read molecules including the alternative exon versus the total number of long-read molecules including/excluding the exon.

We then calculated the root mean squared error (RMSE) between the estimated PSI values by each method on short-read data and the benchmark PSI values calculated from long-read data. As shown in **Fig. 2d** and **Extended Data Fig. 14d**, SCSES consistently achieved the lowest RMSE among all compared methods. Notably, RMSE was reduced by more than 13% in the ovarian cancer dataset and by approximately 16% in the human hippocampus dataset. These results demonstrate that SCSES can accurately recover splicing levels in real, heterogeneous biological tissues.

Fig. 2d: Bar plots comparing the performance of different methods on Ovarian Cancer dataset. The RMSE is calculated between the estimated PSI values by different methods and benchmark values averaged on all cell-event pairs.

Extended Data Figure 1d: Bar plots comparing the performance of different methods on Human Hippocampus dataset. The RMSE is calculated between the estimated PSI values by different methods and benchmark values averaged on all cell-event pairs.

In light of Reviewer’s suggestion, we have added the corresponding results in the revised manuscript:

“To assess the performance of SCSES in real tissues, we applied SCSES and other algorithms to single-cell datasets from an ovarian cancer sample³⁸ and the human hippocampus³⁹, in which each cell was sequenced using paired short-read and long-read technologies (**Methods, Supplementary Note**). The PSI values of high-quality event-cell pairs derived from long-read data were used as benchmarks to calculate the RMSE of PSI estimates from each method. SCSES consistently achieved the lowest RMSE across all comparisons. Notably, RMSE was reduced by over 13% in the ovarian cancer dataset and by approximately 16% in the human hippocampus dataset by SCSES (**Fig. 2d, Extended Data Fig. 14d**).” (Results Section, Page 8, Line 169-177)

“To assess the performance of SCSES in the real heterogeneous single-cell environment, we collected two datasets where paired short-read and long-read sequencing libraries were constructed from individual cells, one of which was derived from high-grade serous ovarian carcinoma tissues³⁸ (sample P1, 8 cell types), and the other was from adult human hippocampus³⁹ (sample f1, 16 cell types).” (Methods Section, Page 26, Line 703-707)

“In paired long-read and short-read datasets, to ensure a fair comparison, we used only the SE events identified by both SCSES and each individual method under comparison. High-confidence event–cell pairs from the long-read data were used as the benchmark; these pairs were defined as splicing events supported by more than 10 long-read molecules in a given cell. The reference PSI values were computed based on the ratio of long-read molecules including the alternative exon versus the total number of long-read molecules including/excluding the exon.” (Methods Section, Page 28, Line 759-765)

“For long-read data processing, SiCeLoRe⁵ (v2.1) was employed for cell barcode and UMI assignment, followed by genome mapping using minimap2⁶ (v2.17) against the reference genome. Short-read data were processed using the 10x Genomics Cell Ranger⁷ (v9.0.1) for genome alignment. Only cells with matched barcodes between long-read and short-read datasets were retained for downstream evaluation. UMI deduplication was performed using an in-house Python script. To preserve the maximum number of splicing-associated reads, junction reads, which were identified by the 'N' tag in the BAM CIGAR column, were prioritized during the deduplication process.” (Extended Data File, Page 4, Line 76-83)

3. The authors' rationale for employing cell lines or in-vitro systems and bulk RNA-seq as ground truth for their benchmark presents a fundamental contradiction. While they rightly assert the superior utility of single-cell approaches for resolving cellular heterogeneity, they simultaneously claim that this very heterogeneity precludes the testing of alternative splicing (AS) methods on real single-cell data. Consequently, they advocate for benchmarking on datasets amenable to bulk analysis, and indeed utilize bulk RNA-seq as the ground truth. This creates a paradox: a method specifically developed for analyzing AS in single-cell data is benchmarked exclusively against scenarios where its purported advantage (single-cell resolution) offers no real added value. This approach severely limits the generalizability and practical applicability of the developed method to genuine biological contexts, a critical concern also highlighted by Reviewer 3. To overcome this inherent limitation and truly validate the method's utility, a crucial next step would be to incorporate long-read single-cell RNA-seq data into the benchmark,

leveraging the long-read detected isoforms as the definitive ground truth. This would directly assess the
method's performance in the heterogeneous single-cell environment it aims to address.

**Reply:** We thank the reviewer for this valuable comment. We fully acknowledge the contradiction pointed out
in our previous benchmarking strategy, where we used cell lines and bulk RNA-seq data as ground truth while
emphasizing the importance of single-cell resolution for addressing cellular heterogeneity.

We would like to clarify two key points regarding our original rationale:

- 1. **Use of cell line data:** Our intent in using cell line or in vitro data was solely to evaluate the accuracy
of PSI estimation under controlled conditions. These datasets provide a simplified system where
cellular heterogeneity is minimized, allowing a more direct comparison between estimated and
reference PSI values.
- 2. **Use of bulk RNA-seq as a benchmark:** We agree that bulk RNA-seq data cannot capture the cellular
heterogeneity intrinsic to real tissues. However, establishing a suitable benchmark of splicing changes
at the single-cell level remains extremely challenging. Ideally, one solution would involve paired
sequencing the same cell using two protocols—one with a specific sequencing method for alternative
splicing detection and quantification to serve as a reference, and the other using a commonly adopted
single cell sequencing technique for method evaluation. In practice, the number of high-quality events
or event–cell pairs that meet these criteria is limited, making them unsuitable for high-throughput
benchmarking of different methods. For example, only 266 high-quality event–cell pairs were detected
from long-read data. Therefore, we initially chose to use bulk data from relatively homogeneous cell
populations as an alternative for evaluating the accuracy of PSI estimation.

As the same reply to above comment 2, in response to the reviewer's suggestion and to overcome the
limitations of our original approach, we have incorporated benchmarking based on paired long-read and
short-read single-cell RNA-seq data, which directly addresses the heterogeneity present in real biological
samples.

Specifically, we used two datasets:

- 1. **An ovarian cancer dataset** with 2,174 cells, each profiled by both scTailoR-seq (long-read) and 10x
Genomics (short-read).
- 2. **A human hippocampus dataset** with 8,681 cells, each sequenced by ScISOr-Seq2 (long-read) and
10x Genomics (short-read).

We used the PSI values derived from long-read data as the ground truth and compared them with estimated
PSI by SCSES and other methods using short-read data. This approach provides a direct and realistic
assessment of method performance in the context of true cellular heterogeneity.

As shown in **Fig. 2d** and **Extended Data Fig. 14d** and replied to above comment 2, SCSES consistently
achieved the lowest RMSE among all compared methods. Notably, RMSE was reduced by more than 13%
in the ovarian cancer dataset and by approximately 16% in the human hippocampus dataset by SCSES.
These results demonstrate that SCSES can accurately recover splicing levels in real, heterogeneous
biological tissues.

- 4. In different scenarios, SCSES appears to exhibit similar performance to Psix. What specific advantages,
if any, can be claimed when asserting that Psix also achieves comparable performance? I would suggest
clarifying and strengthening the evidence showing that SCSES outperforms Psix. As it stands, the
examples provided, including the real single-cell scenario, do not convincingly demonstrate a significant
advantage, which in turn reduces the impact and justification for SCSES. For example, for the iPSC
dataset, the author claims that "SCSES divided the motor neurons (MNs) into two more subclusters (MN-

C1, MN-C2), a distinction that was not as clearly delineated by other methods or by gene expression values". However, in Figure 3c it is evident that Psix also identifies these two subclusters, yielding results that are comparable to those of SCSES (cluster MN-C2 can also be observed by SCASL).

Reply: We thank the reviewer for this valuable comment. We agree that Psix achieves comparable performance to SCSES in certain scenarios, such as Psix can detected MN-C1 and MN-C2 in iPSC dataset. However, we would like to clarify that SCSES offers improved accuracy and broader applicability across a variety of biological datasets.

Firstly, Psix is limited to estimating PSI for exon skipping (SE) events. In contrast, SCSES supports five canonical splicing event types (SE, A3SS, A5SS, IR, MXE) as well as non-canonical events such as alternative last exon (ALE) splicing, providing broader applicability of splicing event types.

Even for the SE events, SCSES can also achieve higher PSI accuracy. In cell line datasets, SCSES can increase the median value of the Spearman correlation coefficients (SCC) between the estimated and benchmark PSI values across all events within each cell by 3%~76%, and decrease the median of the root mean squared error (RMSE) between estimated and the benchmark PSI values across all cells by 8%~42% (**Response Figure 2a**). In simulation datasets, the performance of SCSES is significantly outperformed Psix in identifying planted splicing events and recovering PSI values (**Response Figure 2b**). In real tissue dataset with paired long-read and short-read datasets, SCSES can reduce the RMSE by 53.8% in ovarian cancer dataset and 72.2% in hippocampus dataset (**Response Figure 2c**).

For differentially spliced events (DSE) detection, AUC values for DSE detection of SCSES outperformed Psix in 4/6 comparisons, and have similar performance with Psix in the other 2 comparisons (**Response Figure 2d**). However, SCSES increased the Pearson correlation coefficient (PCC) between estimated splicing changes (Δ PSI) and reference Δ PSI values by 9.7%~42.9%, suggesting more reliably quantification of splicing differences (**Response Figure 2e**).

For cell type identification, compared to Psix, SCSES increased the normalized mutual information (NMI) between PSI-based cell clusters and known reference cell types by 12.6%~49.3% in all three test datasets (**Response Figure 2f**).

Response Figure 2. Performance comparison of Psix and SCSES. (a) Bar plots showing the median Spearman correlation coefficients (SCC) between the estimated and benchmark PSI values across all

events within each cell, and the median root mean squared error (RMSE) between estimated and benchmark PSI values across all cells for each splicing event on cell line datasets. (b) Bar plots showing the accuracy scores in high-quality and low-quality simulation datasets. (c) Bar plots showing the median root mean squared error (RMSE) between estimated and benchmark PSI values of all high-confidence event–cell pairs on paired short-read and long-read datasets. (d) Bar plots showing the area under the curve (AUC) values from DSE detection experiments on cell lines datasets. (e) Bar plots showing the Pearson correlation coefficients (PCC) of Δ PSI values between the estimated results and the benchmark on cell lines datasets. (f) Bar plots showing the normalized mutual information (NMI) between PSI-based cell clusters and known reference cell types.

Moreover, SCSES can identify novel subclusters with biological relevance in multiple myeloma and human embryo datasets, which are not able to be detected by Psix (Response Figure 3, Response Figure 4).

Response Figure 3: UMAP plots showing the cell projections of MM cells using PSI values from SCSES (left) and Psix (right).

Response Figure 4: UMAP plots showing the cell projections of the 36h cells of the human embryo datasets using PSI values from SCSES (left) and Psix (right).

In summary, although Psix is a valuable and high-performing method, SCSES provides consistently improved accuracy, robustness, and resolution across a wider range of splicing event types and biological scenarios — particularly in more complex or heterogeneous datasets. We believe these results reinforce the practical value and broader impact of SCSES.

In light of Reviewer’s suggestion, we revised the manuscript as follows:

“Interestingly, low dimensional reduction analysis showed that SCSES divided the motor neurons (MNs) into two more subclusters (MN-C1, MN-C2), a distinction that was also captured by the Psix and SCASL but was not as clearly delineated by other methods or by gene expression values (Fig. 3b, 3c, Extended Data Fig. 18a, b).” (Result Section, Page 9, Line 225-228)

**Reviewer #1 (Remarks on code availability):**

Please see the attached file to see issues about the code.

**Reply:** We thank the reviewer for this careful check on our code. In response to the bugs and issues raised,
we have revised our code and document accordingly.

1. I successfully installed SCSES using the Docker container, but I would suggest improving the
documentation, for example, by clearly stating that the exported port must be available and not already
in use by another application on the system. Including a complete example would also be very helpful.

**Reply:** We added an illustration to explain that the port bound to Docker should not be in use by the host.
Additionally, we included documentation on how to check for used ports on different operating systems. The
modified description are as follows:

Step 4. Create Docker Container

After building the image, create a Docker container with the following command:

```
docker run -d -p [exported port]:8787 -e PASSWORD=[user password] -v [local directory]:/data --name tes
```

[exported port] : An **unused** port on the host machine to access the container. To show all **used** ports on the host machine, input the following command in linux terminal `netstat -tuwpan 2|awk '{print $4}'|cut -d ":" -f 2|sort|uniq -c` OR `Get-NetTCPConnection | Where-Object { $_.State -eq "Listen" } | Select-Object -ExpandProperty LocalPort|Sort-Object | Group-Object | Select-Object -Property Count, Name` in Windows Powershell.

[user password] : A user-defined password for logging into the RStudio server.

[local directory] : A local directory mapped to the container for data storage and sharing.

Here is an example. In this case, We map host port `1234` to container port `8787`, set the RStudio Server login password to `william`, and mount the host directory `/d/SCSES/` (Windows system) to the container directory `/data`. This allows data located in `/d/SCSES` on the host to be accessed inside the Docker container at `/data`.

```
docker run -d -p 1234:8787 -e PASSWORD=william -v /d/SCSES:/data --name test scses
```

2. At first, I mistakenly assumed the login username would match my personal username. It wasn't clear
that the username to use is "rstudio". I suggest clarifying this point. Including a screenshot of the login
page would also be very helpful in guiding users through the process more quickly.

**Reply:** We emphasized in the GitHub README.md that the username for the RStudio Server is "rstudio",
and added the login screenshot. The modified description is as follows:

Step 5. Access RStudio Server

Now, you can access the RStudio server by opening a web browser and navigating to `[host IP]:[exported port]`.

If you run the above example locally (not remote server), you can access by `http://localhost:1234` or `http://127.0.0.1:1234`. The username to log in Rstudio server is `rstudio` and the password is use-defined in the `docker run` command (`william` in the example).

In this pre-configd RStudio server environment, SCSES and all its dependencies are correctly installed and ready for use.

Please refer to Getting started to start the first experience with SCSES.

3. It's also unclear how to properly use the createConfigshiny(host, port) function. The documentation
doesn't explain whether host and port are required. Initially, I ran the function specifying my host and the
port but it didn't work. Then, I ran it without arguments (createConfigshiny()). While the function executed
without errors, no web page opened. After reviewing the source code, I found that the launch.browser
argument defaults to FALSE, which explains why the Shiny app didn't open automatically. Please
consider documenting this behavior clearly and include an example showing how to use the function to
launch the app in a browser. Additionally, I suggest including more detailed explanations and practical
examples for the configuration file. This would help users better understand how to customize the
settings.

**Reply:** Additional details about the usage of the "createConfigshiny" function have been included.

5. Configuration File

SCSES requires a json-based configuration file to set all parameters in the algorithm. Here is a demo config file of the configure file. For a detailed explanation of the configuration file, please refer to the ConfigurationGuide.txt.

SCSES provides a shiny app to help you to generate the configure file. You can start the app by `createConfigshiny` function.

For non-docker users of SCSES, the full command should be:

```
library(SCSES)
createConfigshiny(host, port, launch.browser=FALSE)
```

You should set the following parameters:

- `host` : the server's IP address, for local access, you can set as "localhost" or "127.0.0.1"
- `port` : The TCP port that the application should listen on. If the port is not specified, and the shiny.port option is set (with options(shiny.port = XX)), then that port will be used. Otherwise, use a random port between 3000:8000, excluding ports that are blocked by Google Chrome for being considered unsafe: 3659, 4045, 5060, 5061, 6000, 6566, 6665:6669 and 6697. Up to twenty random ports will be tried.
- `launch.browser` : if launch the app in the default web browser automatically, default is FALSE. Setting `launch.browser = TRUE` may cause errors in headless environments (servers without GUI) or when no default browser is configured

After running `createConfigshiny`, you will see a URL appear in the console. Copy this URL and paste it into your web browser to access the application.

For docker users of SCSES, you can use the following command, and the web page will be opened automatically:

```
library(SCSES)
createConfigshiny(host = "localhost", launch.browser=TRUE)
```

The web page allows you to specify parameters used in SCSES, such as the BAM file path, the working directory, etc. The hint of each parameter can be found by hovering the mouse over the widget.

After setting all parameters, you can click `Create Config` button and a json file will be generated in the `work_path` you provided if all parameters are set correctly.

Note: The `test dataset` in this Tutorial includes a limited number of cells and events to ensure faster completion of the Tutorial. Therefore, the default parameters in `createConfigshiny` are not suitable. Please use the `createDemoConfigshiny` function instead, which provides default values optimized for the test dataset.

For non-docker users:

```
# For non-docker users
library(SCSES)
createDemoConfigshiny(host = "localhost", launch.browser=FALSE)
```

For docker users:

```
# For docker users
library(SCSES)
createDemoConfigshiny(host = "localhost", launch.browser=TRUE)
```

We also updated the Shiny UI, where parameter hints are now available by hovering the mouse over each
widget, such as

4. The example files are quite large and may be overwhelming for new users. I recommend providing a
simplified, minimal example to make it easier to understand the basics and get started quickly.

**Reply:** We updated the test dataset, which now contains only the aligned reads on chromosome 1. As a
result, the total size has been reduced from 3.6 GB to 1.36 GB, where 238MB are bam files of cells, and
others are all necessary tool files, such as reference genome (.fa), gene annotation files (.gtf, and .gff3), and
sequence conservation files (.bw).

5. Running get10XEXPmatrix() I got the following error: I tried to manually install the Seurat package, and
the installation completed successfully. However, when attempting to load the library, I encountered the
following error: The issue appears to be caused by the missing system library libglpk.so.40 in the
Docker container. To resolve the issue, I added the following line to the SCSES.dockerfile to install the
required system dependencies: apt-get install -y libglpk40 libglpk-dev After updating the Dockerfile, I
rebuilt the Docker image and recreated the container. Once these changes were applied, the
libglpk.so.40 error no longer occurred, and the issue was resolved.

**Reply:** We added the dependency on the Seurat package in the DESCRIPTION file of SCSES. Additionally,
we included the libglpk library as a dependency when building the Docker image.

6. In Step 5, "Constructs Similarity Networks," some parameters are mentioned as adjustable either through
the configuration file or directly via function arguments. However, the default values for these parameters
are not provided. Please include the default settings to improve clarity and reproducibility.

**Reply:** Default parameters for "Construct Similarity Networks" and other steps are added in the help page of
corresponding functions.

getCellSimilarity (SCSES) R Documentation

Construct KNN Graph for Single Cells

Description

This function constructs k-nearest neighbor (KNN) graphs for single cells based on multiple data modalities including RNA-binding protein (RBP) expression, raw junction read counts, and PSI values. It computes cell-to-cell distances using specified distance metrics and applies random walk-based similarity refinement to capture higher-order relationships in the cellular neighborhood structure. The resulting similarity matrices are used for downstream imputation.

Usage

```
getCellSimilarity(
  paras
  rds_path = NULL,
  output_path = NULL,
  feature_num = paras$task$input$feature_num,
  rbp = paras$task$input$rbp,
  cell_similarity_data = paras$task$input$cell_similarity_data,
  distance_method = paras$task$input$KNN$cell$distance_method,
  alpha_cell = paras$task$input$KNN$cell$alpha,
  decay_cell = paras$task$input$KNN$cell$decay,
  kcell_max = paras$task$input$KNN$cell$kmax,
  kcell_min = paras$task$input$KNN$cell$kmin,
  cell.select = NULL
)
```

Arguments

paras A list object containing SCSES configuration parameters, typically loaded using `readSCSESconfig(paras_file)`.

rds_path Character string specifying the path to processed RDS data files. This directory should contain the preprocessed single-cell data matrices (`exprds`, `rc.rds`, `psirds`) generated by upstream processing functions. Default: extracted from `paras$basic$work_path/rds_processed/`.

output_path Character string specifying the output directory for cell similarity results. Default: `work_path/imputation/cell_similarity/`.

feature_num Integer specifying the number of highly variable features to select. Default: 1000 (extracted from `paras$task$input$feature_num`).

rbp Character string or file path specifying RNA-binding protein (RBP) information. Can be either:

- File path to RBP annotation file
- Character string containing RBP gene symbols
- `NULL` to use default RBP set from configuration

Default: `paras$task$input$rbp`.

cell_similarity_data Character string specifying features for cell similarity calculation, separated by semicolons. Valid options include:

- "EXP_RBP": TPM/normalized UMI counts for RNA-binding proteins (source: `rds_processed/exprds`)
- "RC": Raw junction read counts for splicing events (source: `rds_processed/rc.rds`)
- "PSI": PSI values for splicing events (source: `rds_processed/psi.rds`)

Example: "EXP_RBP;RC;PSI"

distance_method Character string specifying the distance metric for cell-to-cell similarity calculation. Supported methods:

- "euclidean"
- "cosine"

Default: "euclidean".

alpha_cell Numeric value between 0 and 1 specifying the random walk probability (1-restart probability). Default: 0.8 (from `paras$task$inputKNNcell$alpha`).

decay_cell Numeric threshold of change in the similarity matrix. The algorithm stops when the change falls below this threshold between consecutive iterations. Default: 0.05 (from `paras$task$inputKNNcell$decay`).

kcell_max Integer specifying the maximum number of nearest neighbors to consider for each cell. Default: 50 (from `paras$task$inputKNNcell$kmax`).

kcell_min Integer specifying the minimum number of nearest neighbors to maintain for each cell. Default: 5 (from `paras$task$inputKNNcell$kmin`).

cell.select Character vector specifying cell identifiers to include in the analysis. Cell names should match the single-cell BAM file names excluding the `.bam` suffix. If `NULL`, all available cells are processed. Default: `NULL` (process all cells).

Value

Character string specifying the path to the cell similarity output directory (`work_path/imputation/cell_similarity/`). The function saves two main output files:

- `cell_similars.rds`: List containing similarity matrices for different features
- `dyn.cell.rds`: List containing the number of neighbors determined for each cell

[Package SCSES version 1.0.0 Index]

7. I encountered the following error while running `getEventSimilarity()`. I then tried with different values for the `kevent` parameter, but consistently encountered the same error. When I reduced `kevent` to 1, it resulted in a different error: In the end, I wasn't able to successfully complete the tutorial from GitHub.

Reply: Due to test dataset includes fewer cells and events to ensure faster execution of the tutorial, the default parameters in "createConfigshiny" are not suitable for test data. We added a new function "createDemoConfigshiny" to provides default values optimized for the test dataset. We added the relative description in the Github documents:

Step 0. Get the cofigure file

You can create the configuration file using the Shiny app. To start the app, run `createConfigshiny` function. Details can be found in here

Note1: The **test dataset** includes a limited number of cells and chromosomes to ensure faster completion of the Tutorial. Therefore, the default parameters in `createConfigshiny` are **not suitable**. Please use `createDemoConfigshiny` function instead, which provides default values optimized for the test dataset.

Note2: An example of parameter configuration for the test dataset is also included in the downloaded files as `cell_line.json`. However, users should modify the file and program paths according to their own system environments.

8. The GitHub repository is currently not very clear and not easy to follow. I recommend improving it by adding more detailed explanations, practical examples, and a clearer, better-organized structure to help users navigate and understand the setup more effectively.

Reply: We fixed all other detected bugs and modified the tutorial for better understanding.

**Reviewer #2 (Remarks to the Author):**

The author has thoroughly addressed all of my questions and concerns, providing detailed explanations and
clarifications where necessary. I am now satisfied with the revisions and there are no outstanding issues.
Based on the improvements made, I recommend that this paper be accepted for publication.

**Reply:** We truly appreciate the reviewer's thoughtful comments and kind recommendation. We are glad that
our revisions and clarifications have resolved all concerns, and we are grateful for your support of the
manuscript's publication.

**Reviewer #3 (Remarks to the Author):**

The authors have addressed some of the original concerns raised in the initial review. However, some have
not been adequately addressed. These points still require clarification.

**Reply:** We thank the reviewer for the constructive feedback and for pointing out the remaining concerns.

1. Rationale for using cell line datasets

To validate splicing heterogeneity within a cell line, the authors projected cells into a UMAP space using raw
PSI values from high-quality events (Response Figure 1). However, a seemingly random distribution in the
UMAP plot does not necessarily imply a lack of heterogeneity, as the spatial distribution is highly influenced
by parameters such as "n.neighbors" in the Seurat pipeline. A more direct measure of heterogeneity would
be the distribution of PSI variances across cells for all events.

Additionally, the criteria used to define high-quality events are not fully described. For example:

- How was the minimum junction read count of 20 determined?

- What quantitative threshold defines a "huge" difference between upstream and downstream splice site
coverage?

- Are the authors only examining exon skipping (SE) events, or are other splicing types included?

Besides, in Response Figure 2a, the meaning of "Raw" PSI is unclear. Does it refer to PSI values for individual
cells, a pseudo-bulk sample, or averaged values across cells? This should be clarified in the text or figure
legend.

**Reply:** We thank the Reviewer for this valuable comment. To address Reviewer's concern, we made following
clarification:

(1). The heterogeneity of splicing events across cell line cells

Following the reviewer's suggestion, we calculated the PSI variances of high-quality splicing events (including
all five event types) within each cell line. **Response Figure 5a** presents the distribution of PSI variances
across four cell line scRNA-seq datasets and a biologically heterogeneous reference dataset (iPSC). As
shown in **Response Figure 5b**, the Kolmogorov–Smirnov (KS) test reveals that PSI variance distributions in
the cell line datasets are significantly lower than those in the reference dataset, indicating limited cell-to-cell
heterogeneity within each cell line. Additionally, we believe it is not appropriate to assess the variances of
non-high-quality events across cells. The PSI values of these events are inherently unreliable in single cells,
and any observed variance may result from technical noise, such as sequencing artifacts, rather than true
biological variability. Therefore, such variances cannot be used to accurately reflect cellular heterogeneity.

Response Figure 5. (a) Density plots showing the distribution of PSI variances in single cell datasets of cell lines and a biologically heterogeneous dataset. (b) Empirical cumulative distribution function curves comparing the distribution of PSI variances in single cell datasets of cell lines and a biologically heterogeneous dataset. The gray curve corresponds to the biologically heterogeneous dataset, serving as a reference for comparison. The Kolmogorov-Smirnov (KS) test was performed to evaluate statistical differences between distributions.

(2). The criteria used to define high-quality events

The minimum junction count of 20 was a widely used threshold for selecting reliable splicing events in bulk and single-cell RNA-seq data analysis. For example, Mertes, C. et al used splicing sites supported by at least 20 reads to fit their model in FRASER³, and Xiang, X. et al removed the AS modules with less than 20 junction reads in SCASL algorithm⁴. Furthermore, we tested a range of thresholds from 10 to 30 in step of 5. As shown in **Response Figure 6**, the variance distributions of the selected splicing events showed minimal differences across these thresholds.

Response Figure 6. Empirical cumulative distribution function curves showing the distribution of PSI variance in single-cell datasets of cell lines for high-quality events with more than 10, 15, 20, 25, and 30 junction reads supporting either inclusion or exclusion splicing events.

The quantitative threshold used to define a "huge" difference between upstream and downstream splice site coverage was a fold change greater than 10 between the upstream and downstream junction read counts,

which was referred from Expedition algorithm⁵.

In Response Letter of the first-round review, we plot UMAP plot with high-quality events of all splicing types,
not only the SE events.

(3). The meaning of “Raw” PSI in (**Response Figure 2 in the 1st-round response letter**) indicates the very
original PSI values derived from the cells without any imputation and correction by SCSES or other methods.
In light of this comment, we have clarified this in the figure legend.

Response Figure 2 in the 1st-round response letter. Density plots showing the high splicing consistency between bulk and single cell datasets of cell lines. **(a)** Distribution of PSI differences between bulk datasets (Ref) and single cell dataset of corresponding event-cell pairs (Raw), where Raw PSI indicates the original PSI values derived directly from each cell without any imputation and correction by SCSES or other methods. **(b)** Distribution of Pearson correlation coefficients (PCC) between PSI values from bulk data and corresponding single cells.

2. Splicing events in cell cycle genes

The authors excluded splicing events in genes annotated with cell cycle-related functions in an attempt to
remove potential confounding effects. However, this approach is problematic. Cell cycle-regulated splicing
events are not necessarily limited to genes labeled as "cell cycle" genes in gene ontology. A more appropriate
strategy would be to identify and exclude splicing events that show cell cycle-dependent regulation, for
instance by analyzing bulk RNA-seq data from synchronized cells at different stages of the cell cycle.

**Reply:** We appreciate the Reviewer for this valuable suggestion. Following the reviewer's suggestion, we
revised our strategy to more directly identify and exclude splicing events that exhibit cell cycle-dependent
regulation, regardless of gene annotation. Specifically, we assessed the correlation between PSI values and
cell cycle phase using bulk RNA-seq data from cell cycle-synchronized samples.

We searched the NCBI database and found several datasets from synchronized cell lines, including HCT116
(GSE123958), HeLa (GSE81485), U2OS (GSE143275), MDA-MB-231 (GSE216497), IRM5/75 (GSE97774),
MCF-7 (GSE94479), REP-1 (GSE116131), and NB4 (PRJEB7566). Unfortunately, synchronized RNA-seq
datasets for some of our evaluation cell lines (e.g., HCC1954, HepG2, HL-60) were not available.

To identify cell cycle–dependent splicing events, we processed each dataset as follows:

- If the dataset included annotated cell cycle stage information, we directly used it. Otherwise, we inferred the cell cycle stage of each sample using the ‘CellCycleScoring’ function from the ‘Seurat’ package.
- We then performed differential splicing analysis between samples from different cell cycle phases to identify differentially spliced events (DSEs). Splicing events with an absolute PSI change ($|\Delta\text{PSI}|$) greater than 0.2 and a P -value <0.05 between any two phases were identified as cell cycle–dependent DSEs. In so doing, we have identified 1656 splicing events that potentially associated with cell cycle.
- All DSEs identified in any of the above cell lines were considered cell-cycle–dependent and excluded from downstream evaluation.

By removing these events, we obtained a set of cell cycle–independent splicing events shared across multiple cell types for performance evaluation. The results of this refined analysis have been incorporated into the revised manuscript as **Fig. 2a-b, d-e** and **Extended Data Fig. 11-16**. Overall, the results and relative performance across methods remained largely consistent, demonstrating that SCSES maintains robust performance even after removing potentially confounding splicing events linked to cell cycle regulation.

Figure2. SCSES recovers the splicing levels in individual cells. a, b. Raincloud plots showing the performance comparison between SCSES-RC and other algorithms in real scRNA-seq data of HCT116. (a) The SCC value refers to the correlation between the estimated PSI values and benchmarks of all events in each cell. (b) The RMSE is calculated between the estimated PSI and benchmarks of an event among all cells. The events detected by both SCSES and the compared algorithm are considered for each comparison. The boxes indicate median (center), Q25, and Q75 (bounds of box), the smallest value within 1.5 times interquartile range below Q25 and the largest value within 1.5 times interquartile range above Q75 (whiskers). The P -values are calculated by the Wilcoxon test. *** for P -value<0.001. c. Bar plots showing the accuracy scores of different algorithms in HCT116 synthetic dataset. The accuracy score is defined as the product of correlation between inference and benchmarks and AS events recall rate. Error bars represent the standard error of the mean from five independent replicates. d. Bar plots comparing the performance of different methods on Ovarian Cancer dataset. The RMSE is calculated between the estimated PSI values by different methods and benchmark values averaged on all cell-event pairs. e. Comparisons of the detected DSEs between SCSES-RC with other algorithms in the real datasets. Bar plot shows the SCC of ΔPSI in DSEs from the benchmark in each comparison group. Colors represent different algorithms. The inner circle represents the difference of AUC for DSEs identification between SCSES and the compared algorithm. On the circle, *: $AUC_{SCSES} - AUC_{ref} > 0.1$, -: $AUC_{SCSES} - AUC_{ref} < 0$. f. Scatter plot showing the inclusion level of *VPS29* exon 2 (left panel) and *NUMB* exon 12 (right panel) between 4 cell lines. g. Read coverage showing the inclusion of *VPS29* exon 2 (left panel) and *NUMB* exon 12 (right panel) on the bulk RNA-seq of four cell lines. Alternative exons are highlighted in red.

In light of Reviewer's suggestion, we revised the manuscript as follows:

"In all cases, SCSES outperformed BRIE1, Expedition and rMATS, exhibiting an extreme increase in median SCC values across cells, ranging from 0.1~0.6, and a reduction in median RMSE by more than 19.4% across events (Fig. 2a, 2b, Extended Data Fig. 11-13)" (Results Section, Page 7, Line 158-162)

"(5) removing splicing events exhibiting cell-cycle dependence. Specifically, we curated eight cell cycle-synchronized RNA-seq datasets (GSE123958, GSE81485, GSE143275, GSE216497, GSE97774, GSE94479, GSE116131 and PRJEB7566) from GEO and European Nucleotide Archive (ENA). Among these datasets, five contained cell cycle phase annotations in their original publications, while the 'CellCycleScoring' function from the 'Seurat' package was used to assign cell cycle phases for the remaining three datasets (Supplementary Table 11). Differential splicing analysis was then conducted through pairwise comparisons between all cell cycle phases. Cell cycle-regulated splicing events were defined based on the following criteria: $\Delta PSI > 0.2$ and $P < 0.05$ (Wilcoxon test), with consistent direction of splicing changes across all samples for each phase pair. Subsequently, AS events associated with cell cycle were excluded from evaluation." (Methods Section, Page 27, Line 726-737)

3. Cell similarity based on RBP transcript expression

The authors state that "splicing-based similarities used in our initial submission, specifically similarities derived from gene expression of RBPs, have better performance compared to the whole-transcriptome or random transcriptomic similarities." However, Response Figure 15 suggests that the improvement is minimal (less than 0.05 difference in Spearman correlation coefficient), and in some cases—such as HL-60 cells—there is virtually no improvement. This raises the question of whether transcript-level RBP expression is truly informative for PSI imputation. Please clarify the rationale for relying on RBP expression and discuss the limitations or possible improvements in this approach.

Reply: We appreciate the Reviewer for this valuable comment. In the SCSES algorithm, we initially chose to define cell splicing similarity based on the expression of RNA-binding proteins (RBPs) rather than the whole

transcriptome, because RBPs play a central role in regulating alternative splicing. They bind directly to
specific RNA sequences and control key steps in RNA splicing process, such as splice site selection and
spliceosome assembly. Therefore, we hypothesized that RBP expression profiles would better capture
splicing-specific characteristics of each cell and could guide PSI imputation more effectively.

However, as the Reviewer correctly pointed out, **Response Figure 15** shows that the improvement in
performance—measured by the Spearman correlation between estimated and ground-truth PSI values—is
modest in many cases (typically <0.05), and nearly absent in certain datasets such as HL-60. We
acknowledge that in such scenarios, RBP expression may not offer a significant advantage over whole-
transcriptome or random gene subsets. This limitation may stem from the fact that RBP expression is only
one of many layers involved in splicing regulation. For instance, post-transcriptional and post-translational
modifications of RBPs, RNA modifications (e.g., m⁶A), and splicing-associated noncoding RNAs also play
important roles. Moreover, transcript-level RBP expression may only indirectly reflect the functional splicing
activity in a given cell. To address this, SCSES is designed to flexibly allow other definitions of splicing-based
cell similarity, including those derived from junction read counts or PSI values, which may better capture cell-
specific splicing states in practice.

As discussed in the manuscript, integrating additional regulatory signals into our modeling framework—
including splicing-associated RNA modifications, post-translational regulation of splicing factors, and
noncoding RNAs—may further improve the depiction of splicing status at the single-cell level.

Response Figure 15 in 1st-round response letter. (a, b) Raincloud plots showing the performance comparison between alternative similarity measures in real scRNA-seq data of cell lines. (a) The SCC value refers to the correlation between the estimated PSI values and benchmarks of all events in each cell. (b) The RMSE is calculated between the estimated PSI and benchmarks of an event among all cells. The P-values are calculated by the Wilcoxon test. N.S.: P -value >0.05 , *: P -value <0.05 , **: P -value <0.01 , ***: P -value <0.001 .

In light of Reviewer's suggestion, we modified the manuscript in the Discussion section:

"In SCSES, we used the gene expressions of RBPs, raw PSI, or raw junction read counts to calculate cell

splicing similarity in current version. RBPs are crucial regulators of RNA splicing process, which bind directly
to specific RNA sequences and control key steps such as splice site selection and spliceosome assembly.
However, transcript-level RBP expression may only indirectly reflect the actual functional splicing activity
within individual cells. In this context, junction read counts and PSI values provide a more direct
representation of cellular splicing states, while their quantitative accuracy is constrained by the inherent
sparsity of junction reads in single-cell data. Moreover, RBP expression is only one of many layers involved
in splicing regulation, other factors, such as post-transcriptional and post-translational modifications of RBPs,
RNA modifications (e.g., m6A), and splicing-associated noncoding RNAs, can also affect the splicing process.
Therefore, integrating these diverse regulatory factors into our modeling framework would improve the
depiction of cellular splicing heterogeneity.” (Discussion Section, Page 19, Line 499-511)

4. Events detected in bulk but not in scRNA-seq data

The authors state that “some events may completely lose all supporting junction reads in individual cells,
even if they can be detected in the pseudo-bulk sample.” This explanation is not convincing. If an event is
detected in a pseudo-bulk sample—which aggregates reads across all cells—one would expect that at least
some individual cells contribute reads supporting that event. Please explain under what circumstances an
event could appear in the pseudo-bulk but be entirely absent in all single cells.

In Response Figure 16, the distinction between panels (a) and (b) is unclear. Specifically, what does “in the
entire scRNA-seq dataset” mean in this context? Please provide clarification.

**Reply:** We appreciate the Reviewer for this comment, and we apologize for our unclear description. We
would like to clarify that our meaning in “some events may completely lose all supporting junction reads in
individual cells, even if they can be detected in the pseudo-bulk sample” is that even if some events are
detected in the pseudo-bulk sample, the events may be disappeared in a certain group of single cells, rather
than in all single cells.

To be honest, we were a little confused about the Major Concern 6 in the first-round review, which is about
“the model perform when imputing AS events that are detected in bulk RNA-seq but are missing in scRNA-
seq data? Specifically, how accurate is the imputation for events that are purely inferred from bulk data and
not observed in the single-cell context?”. We thought that the Reviewer may have two potential questions:
the first one was that the performance in cell-event pairs, where the event was detected in the pseudo-bulk
sample of single-cell data while lost all supporting junction reads in some of the individual cells (**Response**
**Figure 16a in the 1st-round response letter**); the other one was that the performance in the events that
were detected in bulk data used for benchmarks and not identified “in the entire scRNA-seq dataset”
(**Response Figure 16b in the 1st-round response letter**). The meaning of “in the entire scRNA-seq dataset”
in the second situation is “in all single cells”; in another word, the events were missed in pseudo-bulk sample.
For the second questions, to evaluate SCSES performance, we identified splicing events that were present
in the bulk RNA-seq benchmark datasets but absent from our pseudo-bulk detection results from the scRNA-
seq data, then artificially added these bulk-specific events to our scRNA-seq event list and applied the
SCSES algorithm to quantify their splicing levels across individual cells. The results for the first questions
were shown in **Response Figure 16a in the 1st-round response letter**, and the results for the second
question were shown in **Response Figure 16b in the 1st-round response letter**.

Response Figure 16 in the 1st-round response letter. (a) Performance evaluation of SCSES on splicing events detected in the pseudo-bulk sample but showing no junction read evidence at single-cell level across four cell lines. The RMSE comparing benchmarks and raw PSI values (first column) or SCSES estimated PSI values (columns 2-4), calculated for each cell from four cell lines. (b) Performance evaluation of SCSES on splicing events that are purely inferred from bulk data and lack any junction reads in the entire scRNA-seq dataset. The RMSE comparing benchmarks and raw PSI values (first column) or SCSES estimated PSI values (columns 2-4), calculated for each event from four cell lines. The P-values are calculated by the Wilcoxon test. N.S.: P -value >0.05 , *: P -value <0.05 , **: P -value <0.01 , ***: P -value <0.001 .

5. Extended Data Figure 35 Legend

The legend for Extended Data Figure 35 incorrectly describes the content. It should refer to the restart probability parameter. Please revise the legend accordingly.

Reply: We thank the Reviewer for pointing this out and apologize for the oversight. Upon checking, we found that the legend for **Extended Data Figure 35** was correctly updated in the Revised Extended Data File, but was mistakenly mislabeled (legends for **Extended Data Figure 34** and **Extended Data Figure 35** are incorrectly switched) in the Response Letter submitted during the first-round revision. To prevent similar inconsistencies, we have now thoroughly re-checked all figure legends across the revised manuscript and the revised supplementary materials to ensure accuracy and consistency.

**Reference**

- 1 Byrne, A. *et al.* Single-cell long-read targeted sequencing reveals transcriptional variation in ovarian
cancer. *Nat Commun* **15**, 6916 (2024). <https://doi.org:10.1038/s41467-024-51252-6>
- 2 Joglekar, A. *et al.* Single-cell long-read sequencing-based mapping reveals specialized splicing
patterns in developing and adult mouse and human brain. *Nat Neurosci* **27**, 1051-1063 (2024).
<https://doi.org:10.1038/s41593-024-01616-4>
- 3 Mertes, C. *et al.* Detection of aberrant splicing events in RNA-seq data using FRASER. *Nat Commun*
**12**, 529 (2021). <https://doi.org:10.1038/s41467-020-20573-7>
- 4 Xiang, X., He, Y., Zhang, Z. & Yang, X. Interrogations of single-cell RNA splicing landscapes with
SCASL define new cell identities with physiological relevance. *Nat Commun* **15**, 2164 (2024).
<https://doi.org:10.1038/s41467-024-46480-9>
- 5 Song, Y. *et al.* Single-Cell Alternative Splicing Analysis with Expedition Reveals Splicing Dynamics
during Neuron Differentiation. *Mol Cell* **67**, 148-161 e145 (2017).
<https://doi.org:10.1016/j.molcel.2017.06.003>

Point-by-point response to Reviewers

We thank the reviewers for their constructive and detailed evaluation of our manuscript. Their insightful comments have helped to improve our work. We have addressed each point raised in our revised manuscript as follows:

For a better reading of our responses, we marked the words in different colors as follows:

Reviewer comments are marked in **black font**

Responses are marked in **blue font**

New descriptions added to the revised manuscript are marked in **red font**

The location of added descriptions in the revised manuscript is marked in **green font**

Reviewer #1 (Remarks to the Author):

We thank the authors for answering our previous points, in particular by considering two additional single-cell benchmarking datasets, one using long-read data and one using full-coverage short-read data as ground truth. These additions potentially provide stronger evidence of the tool's robustness and performance. However, to enable a complete understanding of the results of these benchmarks, it would be important to consider the following points:

Reply: We thank the reviewer for acknowledging our efforts in evaluation of SCSES performance.

1. The use of a pairwise design, in which SCSES is compared to only one tool in each round, represents a non-standard choice and makes the results more difficult to evaluate, particularly in the absence of information on how the AS events differed across each comparison. Moreover, the performances of SCSES change in each comparison, because a different set of AS events is considered each time. A solution could be to use the same set of events, detected by all tools, for the benchmark. Alternatively, the number of AS events detected by each method, along with the number of AS events present in the ground truth, should be included. For each comparison, the number of AS events considered and the number discarded because not detected by the tools should be provided. A discussion on the differences in AS detection observed between the methods with respect to the ground truth should also be included.

Reply: We thank the reviewer for this valuable comment. In all our experiments, we found that different tools exhibited varying sensitivities in detecting splicing events, which is a well-known limitation of current splicing detection algorithms. As a result, the overlap of detected events across tools differed significantly across datasets. In some dataset, the number of common events by all tools is very small, making a direct evaluation based on the shared set less meaningful. Therefore, using the same set of events detected by all tools is not an appropriate approach. For this reason, we adopted a pairwise comparison strategy, which ensures that each comparison is made on the set of events detectable by both tools, thereby providing a fairer evaluation. Following the reviewer's suggestion, we summarized the numbers of AS events detected by different methods in the short-read data of the Ovarian Cancer and Human Hippocampus datasets, as shown in **Response Figure 1a and 1c**, respectively. For these two datasets, the ground truth was defined as the set of high-quality event–cell pairs identified in the paired long-read data, with the event set determined by taking the union of events detected by all five methods. Based on this definition, the numbers of considered event–cell pairs and those discarded because they were not detected by certain tools are presented in **Response Figure 1b and 1d**, respectively. The **Response Figure 1** is added into the Supplementary Figures as **Extended Data Fig. 14e-h**. In addition, we have included in the revised manuscript a discussion on the

43 differences in AS detection across the methods relative to the ground truth as:

44 “Considering the substantial differences in detected events by different methods (Supplementary Fig. 14e–

45 h), we used only the common high-quality event–cell pairs between SCSES and each compared method for

the evaluation” (Results Section, Line 176-178)

Furthermore, it is well recognized that different splicing event detection tools often produce markedly different

outputs, which can lead to conflicting interpretations (Supplementary Fig. 14e–h, 16a). To address this

limitation, SCSES has been designed as an open framework that allows users to incorporate event lists

generated by their preferred detection tools. (Discussion Section, Line 491-496)

Response Figure 1: (a) Venn diagram showing the event counts identified by different methods from the short-read data of the Ovarian Cancer dataset. (b) Venn diagram showing the high-confidence event–cell pair counts from the long-read data used as a benchmark in the Ovarian Cancer dataset. (c) Venn diagram showing the event counts identified by different methods from the short-read data of the Human Hippocampus dataset. (d) Venn diagram showing the high-confidence event–cell pair counts from the long-read data used as a benchmark in the Human Hippocampus dataset.

2. In the manuscript, it is stated: “The event counts for each comparison and each cell are listed in

Supplementary Table 2.” However, the table contains only about 200 rows, making it unclear whether the

benchmark was conducted at the individual-cell level, as currently stated in the figure captions. At the cell

level, there should be thousands of rows, corresponding to the number of cells of the datasets, not only

200. Furthermore, providing details on which genes and which specific events were included in the

analysis would improve the clarity and completeness of the reported results.

**Reply:** We thank the reviewer for this valuable comment and apologize for the unclear description.

Supplementary Table 2 reports the event counts for each comparison at the individual-cell level in the cell

line datasets, which include cells from five cell lines with a total of 235 cells, thereby explaining why the table
contains only about 200 rows. The counts of event-cells pairs used for Ovarian Cancer and Human
Hippocampus datasets were shown in **Response Figure 1**, respectively. To further improve clarity, we have
added the event information (including the event coordination and located gene name) in **Supplementary**
**Data 3-4**.

**Reviewer #1 (Remarks on code availability):**

The GitHub page has been clarified and is now easier to follow without errors. However, I recommend adding
a note for users who are not familiar with Docker: specifically, that the working directory must have appropriate
write permissions for the Docker container to function correctly. In my case, I was working on a remote server
without sudo access, and I needed to run `chmod -R 777` on the working directory in order for the container
to write files properly.

**Reply:** We thank the reviewer for this valuable reminder. We have added a note to the GitHub page to remind
users to ensure that the working directory has the appropriate write permissions, and included instructions
on how to change the directory permissions if necessary.

Step 4. Create Docker Container

After building the image, create a Docker container with the following command:

```
docker run -d -p [exported port]:8787 -e PASSWORD=[user password] -v [local directory]:/data --name tes
```

[exported port] : An **unused** port on the host machine to access the container. To show all **used** ports on the host machine, input the following command in linux terminal `netstat -tuwanp 2|awk '{print $4}'|cut -d ":" -f 2|sort|uniq -c` OR `Get-NetTCPConnection | Where-Object { $_.State -eq "Listen" } | Select-Object -ExpandProperty LocalPort|Sort-Object | Group-Object | Select-Object -Property Count, Name` in Windows Powershell.

[user password] : A user-defined password for logging into the RStudio server.

[local directory] : A local directory mapped to the container for data storage and sharing.

Please ensure that the [local directory] is writable, as SCSES will generate and save multiple intermediate files during execution. You can modify the directory permissions with `chmod -R 777 [local directory]`. If this command does not resolve the issue, please contact your system administrator for assistance.

75

76

Remarks on code availability:

- 1) I successfully installed SCSES using the Docker container, but I would suggest improving the documentation, for example, by clearly stating that the exported port must be available and not already in use by another application on the system. Including a complete example would also be very helpful.
- 2) At first, I mistakenly assumed the login username would match my personal username. It wasn't clear that the username to use is "**rstudio**". I suggest clarifying this point. Including a screenshot of the login page would also be very helpful in guiding users through the process more quickly.
- 3) It's also unclear how to properly use the `createConfigshiny(host, port)` function. The documentation doesn't explain whether host and port are required. Initially, I ran the function specifying my host and the port but it didn't work. Then, I ran it without arguments (`createConfigshiny()`). While the function executed without errors, no web page opened. After reviewing the source code, I found that the `launch.browser` argument defaults to `FALSE`, which explains why the Shiny app didn't open automatically. Please consider documenting this behavior clearly and include an example showing how to use the function to launch the app in a browser. Additionally, I suggest including more detailed explanations and practical examples for the configuration file. This would help users better understand how to customize the settings.
- 4) The example files are quite large and may be overwhelming for new users. I recommend providing a simplified, minimal example to make it easier to understand the basics and get started quickly.
- 5) Running `get10XEXPmatrix()` I got the following error:

```
> rds.path = get10XEXPmatrix(paras, expr_path, sample_name)
```

```
Error in get10XEXPmatrix(paras, expr_path, sample_name) :  
  Please install Seurat to read files
```

I tried to manually install the Seurat package, and the installation completed successfully. However, when attempting to load the library, I encountered the following error:

```
> library(Seurat)
```

```
Loading required package: SeuratObject  
Loading required package: sp  
  
Attaching package: 'SeuratObject'  
  
The following objects are masked from 'package:base':  
  
  intersect, t
```

```
Error: package or namespace load failed for 'Seurat' in dyn.load(file, DLLpath = DLLpath, ...):  
unable to load shared object '/usr/local/lib/R/site-library/igraph/libs/igraph.so':  
libglpk.so.40: cannot open shared object file: No such file or directory
```

Show Traceback
Rerun with Debug

The issue appears to be caused by the missing system library `libglpk.so.40` in the Docker container. To resolve the issue, I added the following line to the `SCSES.dockerfile` to install the required system dependencies:

```
apt-get install -y libglpk40 libglpk-dev
```

After updating the Dockerfile, I rebuilt the Docker image and recreated the container. Once these changes were applied, the `libglpk.so.40` error no longer occurred, and the issue was resolved.

6) In Step 5, "Constructs Similarity Networks," some parameters are mentioned as adjustable either through the configuration file or directly via function arguments. However, the default values for these parameters are not provided. Please include the default settings to improve clarity and reproducibility.

7) I encountered the following error while running `getEventSimilarity()`

```
[1] "[2025-05-30 17:48:04.349496] Calculate A5SS event Similarity"
[1] "[2025-05-30 17:48:14.50837] Calculate A5SS event Similarity Finished"
[1] "[2025-05-30 17:48:16.480354] Calculate MXE event Similarity"

Error in getEventSimilarity(paras) :
  The number of events is less than K event!

In addition: Warning messages:
1: replacing previous import 'hdf5r::h5version' by 'rhdf5::h5version' when loading 'SCSES'
2: replacing previous import 'hdf5r::h5const' by 'rhdf5::h5const' when loading 'SCSES'
3: replacing previous import 'fs::path' by 'rtracklayer::path' when loading 'SCSES'
4: replacing previous import 'hdf5r::values' by 'rtracklayer::values' when loading 'SCSES'
5: previous export 'hg19' is being replaced
```

I then tried with different values for the `kevent` parameter, but consistently encountered the same error. When I reduced `kevent` to 1, it resulted in a different error:

```
[1] "[2025-05-30 18:36:38.065269] SE event encoding..."
22/22 [=====] - 0s 2ms/step
[1] "[2025-05-30 18:36:49.067445] SE event encoding Finish."
[1] "[2025-05-30 18:36:49.068103] step5 Calculate splicing regulation distance and Combine distance"
[1] "384 rbps are used to calculate splicing regulation information"
[1] "Save data"
[1] "Save data Finished"
[1] "[2025-05-30 18:37:03.889684] step6 Calculate combined event similarity ====="
[1] "[2025-05-30 18:37:05.896843] Calculate A3SS event Similarity"

Error in if (length(pr) == 0L && i == 0L) i <- integer(0L) :
  missing value where TRUE/FALSE needed
```

In the end, I wasn't able to successfully complete the tutorial from GitHub.

The GitHub repository is currently not very clear and not easy to follow. I recommend improving it by adding more detailed explanations, practical examples, and a clearer, better-organized structure to help users navigate and understand the setup more effectively.